# Approximating $f$-Divergences with Rank Statistics

**Viktor Stein** [1] [*]   **José Manuel de Frutos** [2] [*]

## Abstract

We introduce a rank-statistic approximation of $f$-divergences that avoids explicit density-ratio estimation by working directly with the distribution of ranks. For a resolution parameter $K$, we map the mismatch between two univariate distributions $\mu$ and $\nu$ to a rank histogram on $\{0, \ldots, K\}$ and measure its deviation from uniformity via a discrete $f$-divergence, yielding a rank-statistic divergence estimator. We prove that the resulting estimator of the divergence is monotone in $K$, is always a lower bound of the true $f$-divergence, and we establish quantitative convergence rates for $K \to \infty$ under mild regularity of the quantile-domain density ratio. To handle high-dimensional data, we define the sliced rank-statistic $f$-divergence by averaging the univariate construction over random projections, and we provide convergence results for the sliced limit as well. We also derive finite-sample deviation bounds along with asymptotic normality results for the estimator. Finally, we empirically validate the approach by benchmarking against neural baselines and illustrating its use as a learning objective in generative modeling experiments.

## 1. Introduction

Quantifying discrepancies between probability distributions is fundamental in statistics and machine learning. A prominent and widely used class of such measures is $f$-divergences, defined in (1), which include the Kullback-Leibler divergence, total variation, Hellinger, and $\chi^2$-type

---

[*]Equal contribution  [1]Department of Mathematics, Technical University of Munich & Munich Center for Machine Learning, Germany. The majority of the work was conducted while at the Institute of Mathematics at the Technical University of Berlin, Germany & the Berlin Mathematical School. [2]Department of Signal Theory and Communications, Universidad Carlos III, Madrid, Spain. Correspondence to: Viktor Stein <viktor.stein@tum.de>, José Manuel de Frutos <jofrutos@ing.uc3m.es>.

*Proceedings of the 43$^{rd}$ International Conference on Machine Learning*, Seoul, South Korea. PMLR 306, 2026. Copyright 2026 by the author(s).

divergences (Ali & Silvey, 1966; Csiszár, 1967). They arise throughout the field, from hypothesis testing and model comparison to variational objectives for implicit generative modeling (Goodfellow et al., 2014; Nowozin et al., 2016). However, reliably estimating $f$-divergences from samples is challenging: most formulations depend on the density ratio $\frac{d\mu}{d\nu}$, so approaches that first estimate the densities (or their ratio) and then substitute can suffer from severe statistical error in moderate-to-high dimensions (Moon & Hero, 2014; Rubenstein et al., 2019).

A common workaround is to estimate $f$-divergences via *variational* formulations, which recast divergence estimation as (regularized) risk minimization and, in many cases, as a classification-style objective (Nguyen et al., 2010; Nowozin et al., 2016; Ruderman et al., 2012). For KL in particular, the Donsker–Varadhan representation gives another variational objective, which is used by neural mutual-information estimators such as MINE (Belghazi et al., 2018). Related principles include noise-contrastive estimation for unnormalized models, which learns by discriminating data from artificial noise samples (Gutmann & Hyvärinen, 2010). In practice, however, these approaches may require delicate function-class choices and optimization heuristics, and can inherit the instabilities associated with adversarial/variational training (Arjovsky et al., 2017; Gulrajani et al., 2017).

A different line of work mitigates high-dimensional difficulties by comparing *one-dimensional projections* of distributions and aggregating the resulting discrepancies. The sliced Wasserstein distance and its extensions are prominent examples in generative modeling, offering favorable computational scaling by reducing multivariate comparisons to repeated 1D problems (Kolouri et al., 2019; Wu et al., 2019). More generally, sliced probability divergences have been studied from statistical and topological viewpoints (Nadjahi et al., 2020). These ideas suggest that if one can build a robust and scalable 1D divergence estimator, then slicing can lift it to higher dimensions.

In this paper, we develop a *rank-statistic* approximation of $f$-divergences. The construction starts from a fixed reference measure $\nu$ and uses the (univariate) probability integral transform (PIT): if $X \sim \nu$, then $F_\nu(X)$ is uniform on $[0, 1]$ (Rosenblatt, 1952; Savage, 1952; Knothe, 1957). Unifor-

mity diagnostics based on PIT/rank histograms are standard tools in forecast calibration and reliability assessment (Hamill, 2001; Gneiting et al., 2007). We turn this principle into a general divergence construction: we discretize the PIT into a *rank histogram* with $K$ bins and measure its deviation from uniformity via an entropic function $f$. For a related construction to approximate the CDF of a probability density, see (Leblanc, 2012). The resulting rank-statistic divergence is bounded, depends only on *order information*, and admits simple estimators built from sorting and counting operations. We then extend this divergence to $\mathbb{R}^d$ by averaging over random 1D projections, yielding *sliced rank-statistic $f$-divergences* in the spirit of (Kolouri et al., 2019; Nadjahi et al., 2020; Beckmann et al., 2025).

## 1.1. Contributions

The main results of this paper are the following:

- We propose a rank-histogram approximation $\mathbf{D}_{f,\nu}^{(K)}(\mu)$ of the $f$-divergence, parameterized by a resolution $K$, which is an optimization-free estimator of the (sliced) $f$-divergence. This generalizes (de Frutos et al., 2024a;b; 2026) to different choices of entropy function $f$, enabling us to choose differentiable functions $f$ that better interact with automatic differentiation schedules used for learning tasks.

- We establish basic regularity properties, and show that $\mathbf{D}_{f,\nu}^{(K)}(\mu)$ is nondecreasing in $K$ and dominated by $\mathbf{D}_{f,\nu}(\mu)$. Under mild assumptions on the density ratio, we prove consistency as $K \to \infty$ and provide quantitative approximation rates. We also derive finite-sample deviation bounds for the univariate estimator and prove asymptotic normality.

- We define sliced rank-statistic $f$-divergences in $\mathbb{R}^d$ by averaging the univariate construction over random 1D projections, thereby inheriting the univariate $f$-divergence's key properties.

- We benchmark against classical and neural baselines on synthetic tasks, showing that rank-statistic $f$-divergences provide stable approximations of the target $f$-divergence that perform well in high dimensions and with few samples. Our generative transport experiments show that rank-statistic $f$-divergences can be used as sample-based transport objectives for implicit models, illustrated on two-dimensional toy problems and on image datasets.

**Notation** By $\mathbb{N}$ we denote the non-negative integers. For $K \in \mathbb{N}$ we set $[K] := \{0, \ldots, K\}$. The uniform distribution on $[K]$ is denoted by $U_K$. The quantile function of a univariate probability measure $\mu \in \mathcal{P}(\mathbb{R})$ is denoted by

$Q_\mu$ and its CDF by $R_\mu$. The expectation of a function $f$ under $\mu \in \mathcal{P}(\mathbb{R}^d)$ is denoted by $\mathbb{E}_\mu[f] := \int_{\mathbb{R}^d} f(x) \, \mathrm{d}\mu(x)$. The pushforward is denoted by $\#$. We denote by $\mathcal{C}^k$ the $k$-times continuously differentiable functions. We say that $h$ is $H$-Hölder continuous of order $\alpha$ and write $h \in \mathcal{C}^{0,\alpha}$ if $|h(x) - h(y)| \leq H|x - y|^\alpha$ holds for all $x, y$. The range of a function $h$ is denoted by $\mathrm{ran}(h)$. Given i.i.d. samples $X_1, \ldots, X_N \sim \mu$ and $Y_1, \ldots, Y_M \sim \nu$, we denote the corresponding empirical measures by $\hat{\mu}_N$, $\hat{\nu}_M$. Throughout, unhatted symbols denote population-level objects, whereas hatted symbols denote empirical/sample-based quantities.

# 2. One-Dimensional Rank-Based Approximation of $f$-Divergences

We begin with the one-dimensional setting and introduce a rank-based approximation of $f$-divergences. The construction relies on the probability integral transform and a discrete rank histogram that encodes the mismatch between a distribution $\mu$ and a target $\nu$.

In this section, let $\mu, \nu \in \mathcal{P}(\mathbb{R})$ be univariate probability measures and $\nu$ be atomless. Throughout, let $f \colon [0, \infty) \to \mathbb{R} \cup \{+\infty\}$ always be a convex, lower semicontinuous function with $f(1) = 0$ and $\lim_{t \to \infty} \frac{1}{t} f(t) > 0$. We then say that $f$ is an *entropy function*. Note that due to convexity, $f$ is Lipschitz on any compact set $C \subset (0, \infty)$.

The (continuous) $f$-divergence of $\mu$ with respect to $\nu$ is

$$
D_{f,\nu}(\mu) := \begin{cases} \int f\left(\frac{\mathrm{d}\mu}{\mathrm{d}\nu}(x)\right) \mathrm{d}\nu(x), & \text{if } \mu \ll \nu, \\ +\infty, & \text{otherwise.} \end{cases} \quad (1)
$$

Directly working with $\frac{\mathrm{d}\mu}{\mathrm{d}\nu}$ is often inconvenient. Instead, we approximate $D_{f,\nu}(\mu)$ using a rank statistic of $\mu$ relative to $\nu$.

**Definition 2.1.** Fix $K \in \mathbb{N}$. Let $Y \sim \mu$ and $(\tilde{Y}_j)_{j=1}^K \overset{\text{i.i.d.}}{\sim} \nu$, independent of $Y$. The *rank statistic of order $K$* of $\mu$ with respect to $\nu$ is

$$
A_{\mu|\nu}^{(K)} := \#\{j \in \{1, \ldots, K\} : \tilde{Y}_j \leq Y\} \in [K]. \quad (2)
$$

We denote by $Q_{\mu|\nu}^{(K)}$ the probability mass function (PMF) of $A_{\mu|\nu}^{(K)}$ on $[K]$.

When $\mu \ll \nu$, then $\mu = \nu$ if and only if the rank statistic is uniform, i.e., $Q_{\nu|\nu}^{(K)}(n) \equiv 1/(K+1)$, for all $K \in \mathbb{N}$, see Lemma B.1.

The PMF $Q_{\mu|\nu}^{(K)}$ can be seen as a discrete "rank histogram" of $\mu$ with respect to $\nu$. It records how often a draw from $\mu$ falls below the $K$ i.i.d. draws from $\nu$. Departures of this histogram from the uniform law signal discrepancies between $\mu$ and $\nu$.

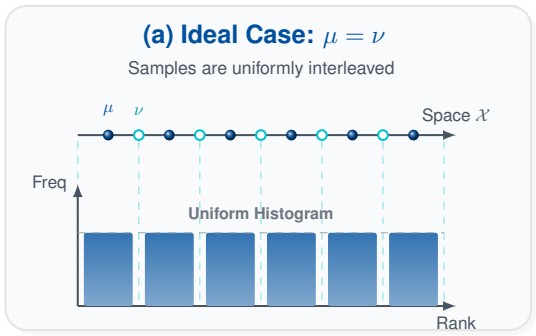
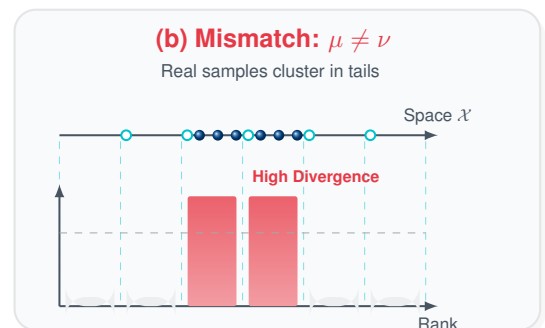

*Figure 1.* Conceptual illustration of the rank-statistic $f$-divergence. **(a)** When $\mu = \nu$, samples are uniformly interleaved, resulting in a uniform rank histogram. **(b)** With a mismatch, samples from $\mu$ cluster in specific rank bins, creating a non-uniform histogram that indicates divergence.

To quantify the discrepancy of the rank histogram from uniformity, we use a discrete $f$-divergence, see also Remark B.2.

**Definition 2.2.** Let $U_K$ denote the uniform distribution on $[K]$. The *rank-statistic $f$-divergence of order $K$ of $\mu$ with respect to $\nu$* is

$$D_{f,\nu}^{(K)}(\mu) \coloneqq \mathrm{D}_f\big(Q_{\mu|\nu}^{(K)} \,\|\, U_K\big)$$
$$= \frac{1}{K+1} \sum_{n=0}^{K} f\big((K+1)\, Q_{\mu|\nu}^{(K)}(n)\big), \quad (3)$$

where $\mathrm{D}_f(\cdot\|\cdot)$ is the discrete $f$-divergence on the finite alphabet $[K]$.

*Example* 2.1. For the entropy function $f_{\mathrm{TV}} \coloneqq |\cdot -1|$ of the total variation divergence, Definition 2.2 recovers the ISL discrepancy $d_K$ from (de Frutos et al., 2024a) on absolutely continuous measures, up to a prefactor: $D_{f_{\mathrm{TV}},\nu}^{(K)}(\mu) = (K+1)d_K(\mu,\nu)$.

Now, we collect basic properties of the rank-statistic $f$-divergence, in particular that $D_{f,\nu}^{(K)}$ is monotone in the rank resolution $K$ and that, similarly to $D_{f,\nu}$, the approximation $D_{f,\nu}^{(K)}$ inherits regularity properties from $f$. The second inequality below generalizes (de Frutos et al., 2024a, Thm. 2) (de Frutos et al., 2026, Thm. 2.2).

**Theorem 2.3.** *Let $\mu, \nu \in \mathcal{P}(\mathbb{R})$ and $K \in \mathbb{N}$. The map $D_{f,\nu}^{(K)}$ is convex, and if $R_\nu$ is continuous, then it is also weakly lower semicontinuous. Furthermore,*

$$D_{f,\nu}^{(K)}(\mu) \leq D_{f,\nu}^{(K+1)}(\mu) \leq D_{f,\nu}(\mu). \quad (4)$$

*Proof.* See Section B.2. $\qquad \square$

*Remark* 2.4 (Markov kernel interpretation of $D_f^{(K)}$). Let $(b_{n,K})_{n=0}^{K}$ be the Bernstein polynomials, see Section A. One can also prove (4) by noticing that the Markov kernel $\kappa \colon \mathbb{R} \times [K] \to [0,1], (y, \{n\}) \mapsto (b_{n,K} \circ R_\nu)(y)$ fulfills

$D_{f,\nu\kappa}(\mu\kappa) = D_{f,\nu}^{(K)}(\mu)$ and then use the data processing inequality for $f$-divergences (Lemma A.2). We also have $D_{f,T_\#\nu}^{(K)}(T_\#\mu) = D_{f,\nu}^{(K)}(\mu)$ for strictly increasing functions $T \colon \mathbb{R} \to \mathbb{R}$.

### 2.1. Approximation properties

We are interested in the behavior of the increasing sequence $\left\{D_{f,\nu}^{(K)}(\mu)\right\}_{K\in\mathbb{N}}$ as the resolution parameter $K$ grows. Increasing $K$ refines the rank histogram, so one expects the discrete quantity $D_{f,\nu}^{(K)}(\mu)$ to approach the continuous $f$-divergence $D_{f,\nu}(\mu)$. This is indeed the case under a mild regularity assumption on the rank density ratio $r$, whose regularity determines the convergence rates precisely in the way that it determines the convergence rate of the Bernstein approximation, see Section A.

**Theorem 2.5** (Convergence of the truncated divergence). *If $\mu \ll \nu$ and $r \coloneqq r_{\mu|\nu} \coloneqq \frac{\mathrm{d}\mu}{\mathrm{d}\nu} \circ Q_\nu \in \mathcal{C}([0,1])$ and $f$ is $L_f$-Lipschitz on $\mathrm{ran}(r)$, then for $K \to \infty$,*

$$D_{f,\nu}(\mu) - D_{f,\nu}^{(K)}(\mu) \begin{cases} \to 0, & \text{if } r \in \mathcal{C}([0,1]), \\ \in O(K^{-\frac{\alpha}{2}}), & \text{if } r \in C^{0,\alpha}([0,1]), \\ & \text{if } r \in \mathcal{C}^2([0,1]), \text{ or} \\ \in O(K^{-1}), & \text{if } r \text{ is Lipschitz and} \\ & f \in \mathcal{C}^2([0,\infty)). \end{cases}$$

*Proof.* Consider the piecewise-constant function

$$r_K(u) \coloneqq (K+1)Q_{\mu|\nu}^{(K)}(n)$$

for $u \in \left[\frac{n}{K+1}, \frac{n+1}{K+1}\right), n \in [K]$. Then,

$$D_{f,\nu}^{(K)}(\mu) = \frac{1}{K+1} \sum_{n=0}^{K} f\big((K+1)Q_{\mu|\nu}^{(K)}(n)\big)$$
$$= \int_0^1 f\big(r_K(u)\big)\, \mathrm{d}u.$$

In Section B.3, we prove that $r_K$ converges uniformly to $r$ and prove the rates. The result then follows from

$$D_{f,\nu}(\mu) - D_{f,\nu}^{(K)}(\mu) \leq \int_0^1 |f(r_K(u)) - f(r(u))| \, \mathrm{d}u$$
$$\leq L_f \|r_K - r\|_\infty. \qquad \square$$

We examine the applicability of Theorem 2.5 to standard $f$-divergences.

*Example* 2.2 (Applicability of Convergence Rates). Among the standard entropy functions considered here, $f_{\mathrm{TV}}$ is the only globally Lipschitz one (up to scalar prefactors).

- If $0 \notin \mathrm{ran}(r)$, then the $O(K^{-\frac{\alpha}{2}})$ rate is achieved for most divergences (including KL, Jensen–Shannon, squared Hellinger, and Jeffreys) since they are Lipschitz away from zero.

- The fast rate $O(K^{-1})$ is obtained if $f \in \mathcal{C}^2([0,\infty))$ which excludes KL and Hellinger and $|\cdot -1|^\alpha$, for $\alpha \in (1,2)$, but holds for the $\chi^2$-divergence (with $f_{\chi^2}(t) = \frac{1}{2}(t-1)^2$) and other polynomial ("Tsallis")-entropy functions, and the triangular discrimination generator $f(t) = \frac{(t-1)^2}{t+1}$ (Lindsay, 1994).

For a long list of choices of $f$, see (Stein et al., 2026, Tab. 1).

## 2.2. Empirical estimation and finite-sample bounds

We now turn to the empirical estimation of the rank-statistic $f$-divergence. Given sample sizes $N, M \in \mathbb{N}$, let $X_1, \ldots, X_N \overset{\text{i.i.d.}}{\sim} \mu$ and $Y_1, \ldots, Y_M \overset{\text{i.i.d.}}{\sim} \nu$. We denote the corresponding empirical measures by

$$\hat{\mu}_N := \frac{1}{N} \sum_{i=1}^N \delta_{X_i}, \qquad \hat{\nu}_M := \frac{1}{M} \sum_{j=1}^M \delta_{Y_j}. \quad (5)$$

The empirical estimator is obtained by applying the rank construction to $\hat{\mu}_N$ and $\hat{\nu}_M$. The exact plug-in rank law is $\bar{Q}_{N,M}^{(K)} := Q_{\hat{\mu}_N|\hat{\nu}_M}^{(K)}$. The corresponding plug-in estimator is

$$D_{f,\hat{\nu}_M}^{(K)}(\hat{\mu}_N) := \frac{1}{K+1} \sum_{n=0}^K f\left((K+1)\bar{Q}_{N,M}^{(K)}(n)\right).$$

Equivalently, one may approximate $\bar{Q}_{N,M}^{(K)}$ by Monte Carlo resampling $K$ reference points from $\hat{\nu}_M$.

The following results bound the finite-sample error between the empirical estimator and the true rank-statistic divergence for fixed $K$ and also give a high-probability concentration bound around the expectation of the empirical estimator.

**Theorem 2.6** (Univariate finite sample complexity and concentration bound). *Let $K \in \mathbb{N}$ be fixed, and let $\mu, \nu \in \mathcal{P}(\mathbb{R})$*

and $\hat{\mu}_N$ and $\hat{\nu}_M$ be their corresponding empirical measures with sample sizes $N$ and $M$. Suppose that $f$ is $L_f$-Lipschitz on $[0, K+1]$. The expected estimation error satisfies

$$\mathbb{E}\left[\left|D_{f,\hat{\nu}_M}^{(K)}(\hat{\mu}_N) - D_{f,\nu}^{(K)}(\mu)\right|\right]$$
$$\leq L_f(K+1)\sqrt{2\pi}\left(\frac{1}{\sqrt{N}} + \frac{1}{\sqrt{M}}\right).$$

*For any $\delta > 0$, with probability at least $1 - \delta$, we have*

$$\left|D_{f,\hat{\nu}_M}^{(K)}(\hat{\mu}_N) - \mathbb{E}\left[D_{f,\hat{\nu}_M}^{(K)}(\hat{\mu}_N)\right]\right|$$
$$\leq L_f(K+1)\sqrt{2\log(2/\delta)\left(\frac{1}{N} + \frac{1}{M}\right)}.$$

*Proof.* See Section B.6 and Section B.7. $\square$

## 3. Sliced Rank-Based $f$-Divergences in Higher Dimensions

We now extend the rank-statistic $f$-divergence from Definition 2.2 to the $d$-dimensional setting via *slicing*. The idea is to reduce the high-dimensional discrepancy between $\mu$ and $\nu$ to a collection of one-dimensional discrepancies along suitably chosen projections, in the spirit of sliced Wasserstein distances and related constructions. Throughout this section, we work with one-dimensional projections along unit directions on the sphere.

For $s \in \mathbb{S}^{d-1}$, let $\mu_s := s_\#\mu$ be the one-dimensional pushforward of $\mu$ by $x \mapsto s^\top x$. For fixed $K \in \mathbb{N}$, Definition 2.2 yields a univariate rank-statistic $f$-divergence $D_f^{(K)}(\mu_s \mid \nu_s)$ describing the mismatch between $\mu_s$ and $\nu_s$. We assume that $\nu_s$ is atomless for almost every $s \in \mathbb{S}^{d-1}$.

**Definition 3.1** (Sliced rank-statistic $f$-divergence). Let $\mu, \nu \in \mathcal{P}(\mathbb{R}^d)$ with $\mu \ll \nu$. Let $\sigma$ denote the uniform probability measure on $\mathbb{S}^{d-1}$. The *sliced rank-statistic $f$-divergence* of order $K$ and the *sliced $f$-divergence* are, respectively,

$$\mathbf{D}_{f,\nu}^{(K)}(\mu) := \int_{\mathbb{S}^{d-1}} D_{f,\nu_s}^{(K)}(\mu_s) \, \mathrm{d}\sigma(s),$$
$$\mathrm{SD}_{f,\nu}(\mu) := \int_{\mathbb{S}^{d-1}} D_{f,\nu_s}(\mu_s) \, \mathrm{d}\sigma(s).$$

Note that in one dimension, $\mathbf{D}_{f,\nu}^{(K)}$ coincides with $D_{f,\nu}^{(K)}$.

The next result states that the results from the previous section carry over to the sliced construction.

**Theorem 3.2.** *The map $\mathbf{D}_{f,\nu}^{(K)}$ is convex. Let $\mu, \nu \in \mathcal{P}(\mathbb{R}^d)$ with $\mu \ll \nu$. Then,*

$$\mathbf{D}_{f,\nu}^{(K)}(\mu) \leq \mathrm{SD}_{f,\nu}(\mu) \leq D_{f,\nu}(\mu), \qquad K \in \mathbb{N}. \quad (6)$$

If $r_{\mu_s|\nu_s} \in \mathcal{C}([0,1])$ for almost all $s \in \mathbb{S}^{d-1}$, then

$$\lim_{K \to \infty} \mathbf{D}_{f,\nu}^{(K)}(\mu) = \mathrm{SD}_{f,\nu}(\mu),$$

*Proof.* See Section B.4. □

Now, we examine the variance of the estimator $D_{f,\nu}^{(K)}$ when estimating its input by samples.

**Theorem 3.3** (Asymptotic normality, sliced one-sample case). *Fix $K \in \mathbb{N}$ and $\mu, \nu \in \mathcal{P}(\mathbb{R}^d)$ with $\mu \neq \nu$ and $D_{f,\nu}^{(K)}(\mu) > 0$, and let $\hat{\mu}_N$ be the empirical approximation from (5). If $f \in \mathcal{C}^1([0, K+1])$ and $f'$ is Lipschitz, then there exists a constant $\tau_K^2 \geq 0$ such that, in distribution,*

$$\sqrt{N}\left(\mathbf{D}_{f,\nu}^{(K)}(\hat{\mu}_N) - \mathbf{D}_{f,\nu}^{(K)}(\mu)\right) \xrightarrow[N \to \infty]{d} \mathcal{N}(0, \tau_K^2).$$

*Proof.* See Section B.8, where we also give a criterion ensuring $\tau_K^2 > 0$. □

# 4. Experiments

We evaluate the proposed rank-statistic $f$-divergence estimator across synthetic and high-dimensional settings. Our experiments quantify estimation accuracy, sensitivity to the resolution parameter $K$, and the benefits of the sliced extension. We also demonstrate its practical behavior when used as a fully sample-based objective in downstream learning on the CIFAR-10 dataset (Krizhevsky, 2009). We defer additional experiments to Section C.

## 4.1. Neural vs. rank-statistic divergence estimation across dimensions

We benchmark the proposed rank-statistic $f$-divergence estimator on the same synthetic setup as the neural KL-divergence estimator of Sreekumar et al. (2021), using their training protocol for all sample sizes (optimizer, architecture scaling, and training schedule), taking $\mu$ and $\nu$ as (suitably truncated) a standard Gaussian and a uniform distribution, respectively (details are deferred to Section C.1).

In contrast to the neural baseline, our rank-statistic estimator involves no iterative optimization: once the samples are fixed, it is fully determined by the rank resolution $K$.

Since both $\mu$ and $\nu$ factorize over coordinates (independent Gaussian components and a product-box truncation),

$$\mathrm{KL}(\mu\|\nu) = \sum_{j=1}^d \mathrm{KL}(\mu_j\|\nu_j).$$

Accordingly, an axis-corrected rank estimator is used: compute the 1D degree-$K$ terms $D_{f_{\mathrm{KL}},\nu_j}^{(K)}(\mu_j)$ and sum them

up:

$$D_{\mathrm{KL,axis}}^{(K)}(\mu\|\nu) := \sum_{j=1}^d D_{f_{\mathrm{KL}},\nu_j}^{(K)}(\mu_j).$$

This leverages the fact that the coordinate axes already capture the discrepancy, without averaging over random projections.

**Evaluation and plots.** For each dimension, Figure 2 reports the estimated $\mathrm{KL}(\mu\|\nu)$ as a function of the sample size $n$, for both the neural baseline and the rank-statistic estimator. The ground-truth $\mathrm{KL}(\mu\|\nu)$ (dashed horizontal line) is computed analytically (implementation details are deferred to Section C.1).

Across all $d \in \{2, 5, 10\}$, the rank-statistic estimator tracks the analytic reference closely and becomes increasingly stable as $n$ grows, with the uncertainty band contracting rapidly; in particular, it is already accurate in the smaller-$n$ regime (most noticeably for $d = 5$ and $d = 10$), where it provides a useful signal before the neural baseline has stabilized. The neural estimator exhibits larger variability and more noticeable deviations from the reference, especially in higher dimensions, suggesting that neural $f$-divergence estimation requires a larger sample budget to become reliable in this setting. Overall, these results indicate that the rank-statistic approach is competitive on this benchmark, often offering smoother and more data-efficient estimates while avoiding iterative training.

## 4.2. Univariate empirical convergence and the influence of the resolution parameter $K$

This subsection benchmarks the one-dimensional rank-statistic estimator against standard $f$-divergences in settings where accurate reference values are available, and studies how the finite resolution parameter $K$ controls the approximation gap. We focus on three widely used discrepancies, Kullback-Leibler (KL), Jensen-Shannon (JS) (Lin, 1991), and the squared Hellinger divergence, and consider four representative mismatch families: (i) Gaussian mean shifts: $\mathcal{N}(0,1)$ vs. $\mathcal{N}(\Delta, 1)$ with $\Delta \in \{0, 0.5, 1, 2\}$ (JS and KL); (ii) Gaussian scale changes: $\mathcal{N}(0,1)$ vs. $\mathcal{N}(0, \sigma)$ with $\sigma \in \{1, 1.2, 1.5, 2\}$ (KL and squared Hellinger); (iii) a symmetric Gaussian mixture: $\frac{1}{2}\mathcal{N}(-\Delta, 1) + \frac{1}{2}\mathcal{N}(+\Delta, 1)$ vs. $\mathcal{N}(0,1)$ (JS); and (iv) a tail-mismatch case of $\mathrm{Laplace}(0,1)$ vs. $\mathcal{N}(0,1)$ (JS).

Reference values use closed forms when available (Gaussian-Gaussian KL and squared Hellinger), and otherwise a high-accuracy one-dimensional numerical/Monte Carlo reference; full details are deferred to Appendix C.2. Unless stated otherwise, the number of samples is $n_\mu = n_\nu = 10,000$ and results are averaged over 10 seeds. Table 1 reports the ratio $D_{f,\nu}^{(K)}(\mu)/D_{f,\nu}(\mu)$ for $K \in$

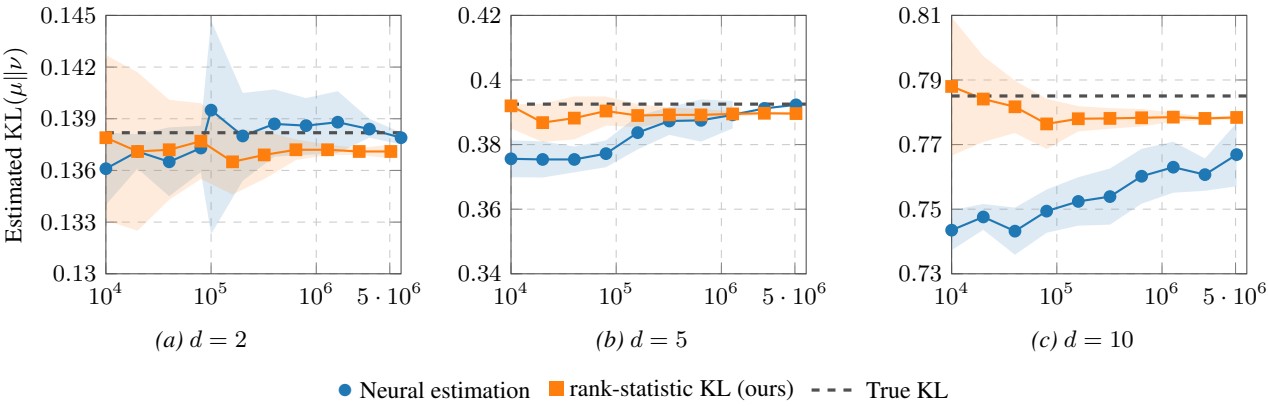

*(a) d = 2*   *(b) d = 5*   *(c) d = 10*

● Neural estimation   ■ rank-statistic KL (ours)   - - - True KL

*Figure 2.* Convergence of Kullback-Leibler divergence estimates for increasing sample size $n$, averaged over 10 independent runs. Shaded bands denote the $\pm 1$ standard deviation interval. Results are shown for $K = 64$ across all samples.

$\{32, 64, 128, 256, 512\}$. For the mean shift experiments, the Jensen-Shannon (JS) divergence outperforms the KL, while for the scale change experiments, the KL outperforms the squared Hellinger divergence. Additional figures and sweeps over $(n, K)$ are provided in Section C.2.

### 4.3. Sliced rank-statistic $f$-divergences: Empirical convergence

We consider $d$-dimensional benchmarks using the sliced estimator $D_{f,\nu}^{(K)}(\mu)$, obtained by averaging the one-dimensional rank divergence over $L$ random projections. Unless stated otherwise, $K = 64$, $L = 128$, $n_\mu = n_\nu = 10,000$, and results are reported as mean±std over $R = 10$ runs. Implementation details are deferred to Appendix C.3.

Three settings are considered: (i) Gaussian-Gaussian pairs, (ii) covariance mismatches (isotropic and anisotropic), and (iii) non-Gaussian pairs. For (i), KL and squared Hellinger have closed-form references, while JS is approximated by a moment-matched Gaussian proxy. For (iii), reference values are obtained by Monte Carlo evaluation of the divergence formula using closed-form log-densities.

Figure 3 reports the ratio $d\, D_{f,\nu}^{(K)}(\mu)/D_{f,\nu}(\mu)$ for mean-shift benchmarks across several dimensions. Overall, the ratio stays close to one with moderate variability, indicating that the simple $d$-scaling provides a reasonable normalization in these settings. Deviations become more noticeable in higher dimension, especially for JS and KL, suggesting that a fixed number of projections $L$ can lead to mild under/over-estimation as $d$ grows, while squared Hellinger remains comparatively stable. Additional benchmarks and ablations are reported in Appendix C.3.

**Practical choice of $K$ and $L$.** The rank resolution $K$ controls the one-dimensional approximation–estimation trade-off. Small values of $K$ lead to a coarse but statistically

stable rank histogram, while larger values provide a finer approximation at the cost of increased finite-sample variability. This mirrors the classical binning trade-off in calibration-error estimation: coarse binning can average out structured departures from uniformity, whereas overly fine binning leaves too few samples per bin and increases statistical noise (Roelofs et al., 2022). In practice, this motivates using moderate values of $K$, or progressive schedules that start from a coarse resolution and increase $K$ as the coarse rank histogram stabilizes during training.

The number of projections $L$ controls the Monte Carlo approximation of the spherical average. For fixed $K$, this projection error behaves like a standard Monte Carlo error of order $O(L^{-1/2})$ under bounded directional discrepancies, as in general sliced probability divergences (Nadjahi et al., 2020). Moderate values of $L$ can be effective when the discrepancy is diffuse across directions, whereas localized or highly anisotropic discrepancies may require substantially more projections. Similar projection-complexity effects have also been observed in sliced Wasserstein generative models and sliced-Wasserstein flows (Wu et al., 2019; Liutkus et al., 2019). In such cases, structured or variance-reduced directions, such as orthogonal or quasi-Monte Carlo projections, may be preferable.

### 4.4. Generative Transport Dynamics for Rank $f$-Divergences

A useful way to turn a discrepancy into a learning principle is to interpret it as an energy and derive an update rule that transports samples in data space toward a target distribution. In our setting, the energy is the (sliced) rank $f$-divergence, and we implement its minimization through particle transport dynamics based on one-dimensional quantile matching.

Given particles $\{x_i\}_{i=1}^N \sim \mu$ and reference samples $\{y_j\}_{j=1}^M \sim \nu$, we draw directions $s_1, \ldots, s_L \in \mathbb{S}^{d-1}$ and

| Family | Scen. | Param. | Ratio $D_{f,\nu}^{(K)}(\mu)/D_{f,\nu}(\mu)$ for $K =$ | | | | |
|---|---|---|---|---|---|---|---|
| | | | 32 | 64 | 128 | 256 | 512 |
| Mean shift | JS | $\Delta = 0.5$ | $0.933 \pm 0.040$ | $0.968 \pm 0.041$ | $0.989 \pm 0.042$ | $\mathbf{1.003 \pm 0.042}$ | $1.013 \pm 0.042$ |
| | JS | $\Delta = 1.0$ | $0.928 \pm 0.033$ | $0.961 \pm 0.034$ | $0.981 \pm 0.035$ | $0.992 \pm 0.035$ | $\mathbf{0.999 \pm 0.035}$ |
| | JS | $\Delta = 2.0$ | $0.930 \pm 0.008$ | $0.962 \pm 0.008$ | $0.981 \pm 0.009$ | $0.991 \pm 0.009$ | $\mathbf{0.997 \pm 0.009}$ |
| | KL | $\Delta = 0.5$ | $0.946 \pm 0.060$ | $0.987 \pm 0.063$ | $\mathbf{1.013 \pm 0.065}$ | $1.030 \pm 0.066$ | $1.044 \pm 0.068$ |
| | KL | $\Delta = 1.0$ | $0.880 \pm 0.024$ | $0.924 \pm 0.025$ | $0.952 \pm 0.026$ | $0.969 \pm 0.027$ | $\mathbf{0.980 \pm 0.027}$ |
| | KL | $\Delta = 2.0$ | $0.775 \pm 0.010$ | $0.844 \pm 0.012$ | $0.895 \pm 0.013$ | $0.933 \pm 0.015$ | $\mathbf{0.959 \pm 0.016}$ |
| Scale change | KL | $\sigma = 1.2$ | $0.743 \pm 0.063$ | $0.841 \pm 0.070$ | $0.908 \pm 0.072$ | $0.954 \pm 0.072$ | $\mathbf{0.991 \pm 0.072}$ |
| | KL | $\sigma = 1.5$ | $0.779 \pm 0.027$ | $0.872 \pm 0.029$ | $0.927 \pm 0.030$ | $0.958 \pm 0.031$ | $\mathbf{0.977 \pm 0.031}$ |
| | KL | $\sigma = 2.0$ | $0.803 \pm 0.018$ | $0.898 \pm 0.020$ | $0.953 \pm 0.021$ | $0.982 \pm 0.022$ | $\mathbf{0.998 \pm 0.022}$ |
| | Hell$^2$ | $\sigma = 1.2$ | $0.741 \pm 0.077$ | $0.853 \pm 0.089$ | $0.931 \pm 0.098$ | $0.986 \pm 0.106$ | $\mathbf{1.029 \pm 0.111}$ |
| | Hell$^2$ | $\sigma = 1.5$ | $0.735 \pm 0.035$ | $0.842 \pm 0.039$ | $0.908 \pm 0.041$ | $0.948 \pm 0.042$ | $\mathbf{0.973 \pm 0.042}$ |
| | Hell$^2$ | $\sigma = 2.0$ | $0.744 \pm 0.014$ | $0.858 \pm 0.014$ | $0.926 \pm 0.014$ | $0.965 \pm 0.013$ | $\mathbf{0.987 \pm 0.012}$ |
| Multimodal | JS | $\Delta = 0.5$ | $0.746 \pm 0.157$ | $0.849 \pm 0.176$ | $0.926 \pm 0.189$ | $\mathbf{0.994 \pm 0.196}$ | $1.068 \pm 0.199$ |
| | JS | $\Delta = 1.0$ | $0.769 \pm 0.038$ | $0.849 \pm 0.040$ | $0.898 \pm 0.041$ | $0.929 \pm 0.041$ | $\mathbf{0.948 \pm 0.042}$ |
| | JS | $\Delta = 2.0$ | $0.846 \pm 0.019$ | $0.912 \pm 0.020$ | $0.950 \pm 0.021$ | $0.972 \pm 0.021$ | $\mathbf{0.985 \pm 0.021}$ |
| Heavy tails | JS | – | $0.488 \pm 0.028$ | $0.651 \pm 0.036$ | $0.778 \pm 0.041$ | $0.869 \pm 0.044$ | $\mathbf{0.933 \pm 0.046}$ |

*Table 1.* 1D divergence estimation benchmarks (10 runs). We report the ratio estimate/reference (mean $\pm$ std) for various $K$ values. Boldface highlights, for each row, the $K$ whose mean ratio is closest to 1 (i.e., the most accurate approximation). Ratios above one arise from finite-sample Monte Carlo/numerical-reference error and do not contradict the population lower-bound property (4).

form the one-dimensional projections

$$x_i^{(\ell)} := \langle x_i, s_\ell \rangle, \qquad y_j^{(\ell)} := \langle y_j, s_\ell \rangle.$$

For each slice $\ell$, let $\nu^{(\ell)} := (s_\ell)_\# \nu$ denote the projected reference law, and let $\widehat{\nu}^{(\ell)} := \frac{1}{M} \sum_{j=1}^M \delta_{y_j^{(\ell)}}$ be its empirical counterpart. We compute the initial soft rank coordinates in slice $\ell$, where the superscript $(\ell)$ indexes the projection direction and the subscript 0 denotes the initial rank-space state:

$$v_{0,i}^{(\ell)} \approx \widehat{R}_{\nu^{(\ell)}, \tau}\big(x_i^{(\ell)}\big) \in [0, 1], \qquad i = 1, \dots, N.$$

Equivalently, we write $\mathbf{v}_0^{(\ell)} = \big(v_{0,1}^{(\ell)}, \dots, v_{0,N}^{(\ell)}\big) \in [0,1]^N$. Here, $\widehat{R}_{\nu^{(\ell)}, \tau}$ denotes the smoothed empirical CDF constructed from the projected reference samples $\{y_j^{(\ell)}\}_{j=1}^M$, obtained by replacing the indicator functions in the empirical CDF by logistic soft indicators, as in standard differentiable rank approximations.

To refine these rank coordinates, we associate with any vector $\mathbf{v} = (v_1, \dots, v_N) \in [0,1]^N$ a discrete PMF on $\{0, \dots, K\}$ via a Bernstein-smoothed histogram, which can be viewed as a smooth sample-based analogue of $Q_{\mu|\nu}^{(K)}$:

$$\widehat{Q}^{(K)}(\mathbf{v})(n) := \frac{1}{N} \sum_{i=1}^N b_{n,K}(v_i), \qquad n = 0, \dots, K. \tag{7}$$

We then measure its deviation from the discrete uniform law

$U_K$, as in Definition 2.2:

$$\mathrm{D}_f\big(\widehat{Q}^{(K)}(\mathbf{v}) \,\|\, U_K\big) = \frac{1}{K+1} \sum_{n=0}^K f\Big((K+1)\widehat{Q}^{(K)}(\mathbf{v})(n)\Big). \tag{8}$$

The rank-space refinement in slice $\ell$ is then defined by the proximal step

$$\mathbf{v}_1^{(\ell)} \in \underset{\mathbf{v} \in [0,1]^N}{\arg\min} \left\{ \mathrm{D}_f\big(\widehat{Q}^{(K)}(\mathbf{v}) \,\|\, U_K\big) + \frac{1}{2\eta} \big\|\mathbf{v} - \mathbf{v}_0^{(\ell)}\big\|_2^2 \right\}, \tag{9}$$

where $\eta > 0$ controls the trust in the current rank coordinates. In practice, (9) can be approximated by deterministic updates (e.g. SGD) or by Langevin-type inner samplers such as ULA or MALA.

The updated rank coordinates are mapped back to the projection axis through the empirical quantile map of the reference slice:

$$z_i^{(\ell)} \approx Q_{\widehat{\nu}^{(\ell)}}\big(v_{1,i}^{(\ell)}\big), \qquad \Delta_i^{(\ell)} := z_i^{(\ell)} - x_i^{(\ell)}.$$

Here, $Q_{\widehat{\nu}^{(\ell)}}$ is evaluated by sorting the projected reference samples and linearly interpolating between adjacent order statistics, following the sliced-Wasserstein flow construction (Liutkus et al., 2019, Sec. 3.3 and Alg. 1). The scalar correction $\Delta_i^{(\ell)}$ is therefore a one-dimensional monotone transport correction in slice $\ell$. Finally, we lift these corrections back

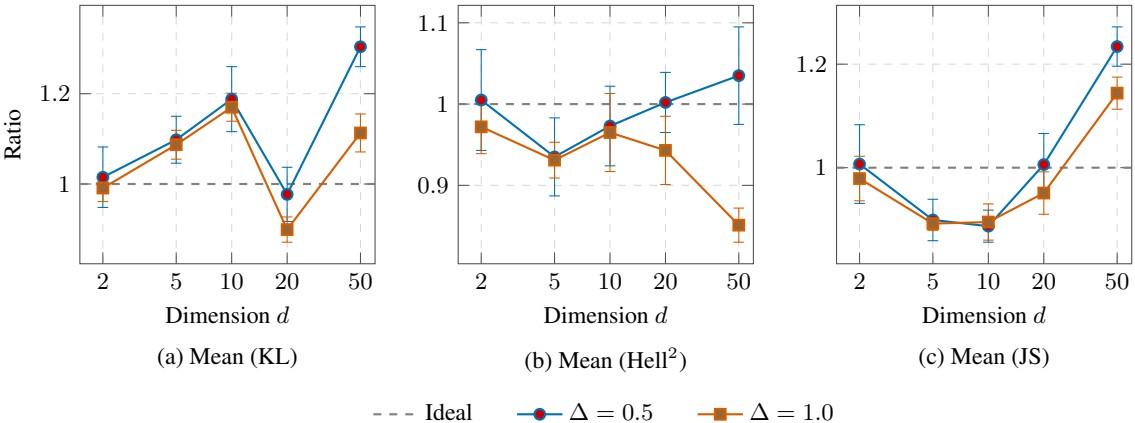

Figure 3. Comparison of mean shift metrics across dimensions for KL, Hellinger, and JS divergences.

to $\mathbb{R}^d$ by aggregating over slices:

$$x_i \leftarrow x_i + \varepsilon \frac{d}{L} \sum_{\ell=1}^{L} \Delta_i^{(\ell)} s_\ell, \qquad (10)$$

with step size $\varepsilon > 0$ and, optionally, per-slice clipping for stability. Iterating (10) yields a practical transport dynamics that moves the particle cloud toward $\nu$ while being driven by a bounded, rank-based energy. The full pseudocode is given in Appendix C.4.

*Remark* 4.1. This particle algorithm resembles an explicit time discretization of a Wasserstein gradient flow of a Moreau envelope of $D_{f,\nu}^{(K)}$ (similar to (Stein et al., 2026)), the difference being that here the Moreau envelope is taken in quantile space.

### 4.4.1. TWO-DIMENSIONAL TOY EXAMPLES

We illustrate the induced particle dynamics on four 2D toy targets: (i) a checkerboard distribution, (ii) a noisy ring, (iii) a two-spirals dataset, and (iv) a two-component Gaussian mixture (two blobs). In each case, we draw $M$ reference samples from the target $\nu$ and initialize $N$ particles from an isotropic Gaussian. We then iterate (10) for a fixed number of outer steps and report snapshots at $t \in \{0, 1, 5, 10, 20, 40, 100, 200, 400\}$.

Figure 4 uses an SGD approximation of the rank-proximal refinement (9) with the KL generator, $L = 10$ projection directions, and trust-region parameter $\eta = 0.5$. We use a moderate outer step size by starting from $\varepsilon = 0.20$ and linearly annealing it to $0.15$ over training; in parallel we anneal the rank smoothing temperature (cf. Algorithm 1) from $\tau = 0.30$ to $0.10$ and increase the rank resolution from $K = 80$ to $K = 128$.

Qualitatively, the dynamics rapidly match the target geometry across very different structures. On the checkerboard, particles populate multiple disconnected cells without de-

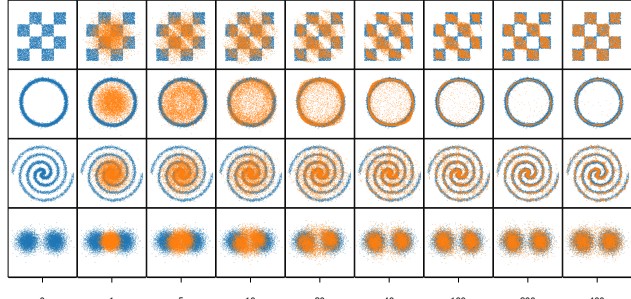

Figure 4. Rank-Proximal Transport on 2D toy targets. Using SGD to minimize the rank-statistic KL with $L = 10$ random projections, particles (orange) evolve from a Gaussian start ($t = 0$) to match the target support (blue).

generating to a single region; on the ring, they expand radially and then redistribute along the angular directions; on spirals, the cloud progressively aligns with the nonlinear manifold; and on the two-blobs mixture, it splits and concentrates around both modes.

### 4.4.2. CIFAR-10 EXPERIMENTS

We next illustrate the induced particle dynamics on *CIFAR-10* (Krizhevsky, 2009) using a center-outward rank-proximal transport (CO-RPT) update (see details of the algorithm in Appendix C.5.1). Although CIFAR-10 is natively $32 \times 32$, we use a high-quality bicubic upsampled $64 \times 64$ version with antialiasing, and treat each particle as an RGB image flattened to $\mathbb{R}^{3 \cdot 4096}$. We use $M$ real CIFAR-10 training images as reference samples from the target distribution $\nu$. Starting from $N$ i.i.d. Gaussian particles, we iterate the center-outward transport update for a fixed number of outer steps (see Section C.5 for full schedules).

In our runs, we use the Jensen–Shannon generator with trust parameter $\eta = 0.5$ and a small number of inner prox steps per iteration. Since the transport is performed directly in a

high-dimensional pixel space, we use a moderate outer step size, linearly annealing $\varepsilon$ from $0.16$ to $0.10$ over training. In parallel, we anneal the rank-smoothing temperature from $\tau = 0.30$ to $\tau = 0.07$ and increase the rank resolution from $K = 64$ to $K = 160$. To avoid overly large updates in the ambient pixel space, we clip the per-particle correction with a cap of $0.30$. We additionally use a small angular blending parameter $\beta_{\text{angle}} = 0.01$.

Qualitatively, the dynamics progressively transforms the initial noise cloud into structured images that match low- and mid-level statistics of CIFAR-10, such as global color balance, coarse spatial layout, and local texture; see Figure 5. Further implementation details, and qualitative results on MNIST, are deferred to Section C.5.

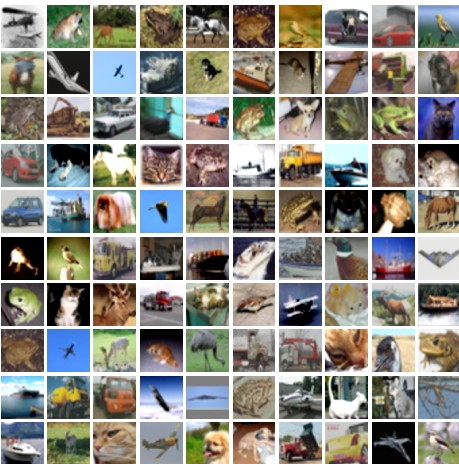

*Figure 5.* CO-RPT samples on CIFAR-10 ($64 \times 64$) after $T = 20{,}000$ outer steps.

As an additional pretraining experiment, Section C.6 shows that rank-statistic pretraining improves precision/recall trade-offs for DCGAN on MNIST; Table 5 reports that TV/KL pretraining improves precision after DCGAN fine-tuning, while JS and Hellinger emphasize recall.

## 5. Conclusions, Future Work, and Limitations

We proposed *rank-statistic* approximations of $f$-divergences that replace density-ratio estimation with simple rank counting in a discrete histogram. The resulting surrogate $D_{f,\nu}^{(K)}(\mu)$ has a clean variational structure (convexity and weak lower semicontinuity), is nondecreasing in the resolution $K$, and remains dominated by the target divergence; under mild regularity of the density ratio we proved consistency as $K \to \infty$ with quantitative rates, and established finite-sample deviation guarantees for practical estimators. For multivariate data, we introduced sliced rank-statistic divergences by averaging the univariate construction over random 1D projections, inheriting its key properties, and we validated the approach empirically on synthetic benchmarks and as a

sample-based transport and pretraining objective for implicit generative modeling.

We generalized the rank-statistic approximation of the TV-divergence from (de Frutos et al., 2026). Since the TV-divergence is the only $f$-divergence which is also an integral probability metric (IPM) (Müller, 1997), it would be interesting to see if IPMs like maximum mean discrepancy (Borgwardt et al., 2006) or the Wasserstein-1 metric can be approximated by rank statistics as well. It could also be promising to replace the Bernstein polynomials by another family, like B-splines or general non-linear filters. Investigating the geodesic convexity properties of $D_{f,\nu}^{(K)}$ in the Wasserstein geometry could yield convergence rates of the generative transport dynamics. Lastly, it would be interesting to identify a joint regime for $(K, N) \to \infty$ yielding the best convergence rate. We leave these questions for future work.

Our results also highlight limitations: projection complexity in high dimensions when discrepancies are strongly anisotropic or concentrated in dependencies that are hard to detect from 1D views, and the reliance on random directions for capturing such effects efficiently. Future work includes variance-reduced and structured projection schemes (e.g., orthogonal or quasi-Monte Carlo directions), improved anisotropy calibration beyond simple $d\times$ normalizations, tighter dimension-dependent guarantees, and scaling the objective inside modern large-scale generative pipelines. We expand the discussion of limitations and future directions in Appendix D.

## Impact Statement

This paper presents work whose goal is to advance the field of Machine Learning, in particular through rank-based approximations of $f$-divergences and sample-based objectives for generative modeling. The proposed methods are generic and are not designed for identification, surveillance, or applications involving sensitive personal data.

Our empirical evaluation focuses on standard natural-image benchmarks and avoids datasets centered on identifiable facial images. This choice keeps the experiments focused on the methodological properties of the proposed objective while reducing privacy-related concerns. We do not identify any additional societal consequences that require specific highlighting beyond these general considerations.

## Acknowledgements

This project has received funding from the European Research Council (ERC) under the European Union's Horizon Europe research and innovation programme (grant agreement No. 101198055, project acronym NEITALG). This

work has also been partially supported by the Office of Naval Research (award N00014-22-1-2647) and Spain's Agencia Estatal de Investigación (ref. PID2024-158181NB-I00 NISA and PID2021-123182OB-I00 EPiCENTER) funded by MCIN/AEI/10.13039/501100011033 and by "ERDF A way of making Europe."

Views and opinions expressed are, however, those of the author(s) only and do not necessarily reflect those of the European Union, the European Research Council Executive Agency, the U.S. Office of Naval Research, or the Spanish Agencia Estatal de Investigación. Neither the European Union nor any of the aforementioned granting authorities can be held responsible for them.

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

# Appendix

In this appendix, we first recall well-known results about Bernstein polynomials and $f$-divergences in Section A. In Section B we prove the theorems from the main text, and in Section C we provide supplementary explanations and experiments. Finally, in Section D, we elaborate on limitations and future work.

| Symbol | Meaning |
|--------|---------|
| $\mathbb{N}$ | Set of non-negative integers. |
| $[K]$ | Discrete set $\{0, \dots, K\}$. |
| $\mathcal{P}(\mathbb{R}^d)$ | Set of probability measures on $\mathbb{R}^d$. |
| $\mu, \nu$ | Probability measures to be compared; typically $\nu$ is the reference distribution. |
| $\hat{\mu}_N, \hat{\nu}_M$ | Empirical measures built from $N$ samples from $\mu$ and $M$ samples from $\nu$. |
| $Q_\mu, R_\mu$ | Quantile function and cumulative distribution function of a univariate measure $\mu$. |
| $T_{\#}\mu$ | Pushforward of $\mu$ by a map $T$. |
| $\mu\kappa$ | Pushforward of $\mu \in \mathcal{P}(\mathbb{R})$ by a Markov kernel $\kappa\colon [K] \times \mathbb{R} \to [0,1]$, i.e. $(\mu\kappa)(n) = \int_\mathbb{R} \kappa(n,x)\,\mathrm{d}\mu(x)$ for $n \in [K]$. |
| $f$ | Entropy function/generator of an $f$-divergence. |
| $D_{f,\nu}(\mu)$ | Continuous $f$-divergence of $\mu$ with respect to $\nu$. |
| $K$ | Rank resolution parameter. |
| $U_K$ | Uniform distribution on $[K]$. |
| $A_{\mu|\nu}^{(K)}$ | Rank statistic of order $K$ of $\mu$ with respect to $\nu$. |
| $Q_{\mu|\nu}^{(K)}$ | Probability mass function of $A_{\mu|\nu}^{(K)}$, interpreted as a rank histogram. |
| $D_{f,\nu}^{(K)}(\mu)$ | Rank-statistic approximation of $D_{f,\nu}(\mu)$ at resolution $K$. |
| $r_{\mu|\nu}$ | Quantile-domain density ratio $r_{\mu|\nu} = \frac{d\mu}{d\nu} \circ Q_\nu$, when $\mu \ll \nu$. |
| $r_K$ | Piecewise-constant approximation of $r_{\mu|\nu}$ induced by the rank histogram. |
| $b_{n,K}$ | Bernstein basis polynomial of degree $K$. |
| $s \in \mathbb{S}^{d-1}$ | One-dimensional projection direction. |
| $\mu_s, \nu_s$ | One-dimensional projected measures along direction $s$. |
| $\sigma$ | Uniform probability measure on the sphere $\mathbb{S}^{d-1}$. |
| $\mathbf{SD}_{f,\nu}(\mu)$ | Sliced $f$-divergence obtained by averaging one-dimensional divergences over projections. |
| $\mathbf{D}_{f,\nu}^{(K)}(\mu)$ | Sliced rank-statistic $f$-divergence of order $K$. |
| $L$ | Number of random projection directions used in empirical sliced estimators. |
| $\tau$ | Smoothing temperature used in differentiable rank/CDF approximations. |

*Table 2.* Frequently used notation.

## A. Well-known results

Here, we recall results about Bernstein polynomials and $f$-divergences.

Bernstein polynomials were introduced in (Bernstein, 1912) to prove the Weierstraß approximation theorem in a simple way.

**Lemma A.1** (Properties of Bernstein polynomials). *The Bernstein polynomials $b_{n,K}\colon [0,1] \to [0,\infty)$, $u \mapsto \binom{K}{n} u^n (1 - u)^{K-n}$ have the following properties.*

1. *We have $\sum_{n=0}^K b_{n,K}(s) = 1$ for all $s \in [0,1]$, and $\int_0^1 b_{n,K}(s)\,\mathrm{d}s = \frac{1}{K+1}$ for $n \in [K]$.*

2. *For $K \in \mathbb{N}$ and $n \in [K-1]$ we have*

$$b_{n,K-1} = \frac{K-n}{K}b_{n,K} + \frac{n+1}{K}b_{n+1,K}.$$

3. *The function $(K+1)b_{n,K}$ is the probability density function of the $\mathrm{Beta}(n+1, K-n+1)$ distribution, whose mean and variance are $\frac{n+1}{K+2}$ and $\frac{(n+1)(K-n+1)}{(K+2)^2(K+3)}$, respectively.*

4. *Let $f \in \mathcal{C}([0,1])$. For the Bernstein operator $B_K \colon \mathcal{C}([0,1];\mathbb{R}) \to \Pi_K$, $f \mapsto \sum_{n=0}^{K} f\left(\frac{n}{K}\right) b_{n,K}$ we have $\|B_K[f] - f\|_\infty \to 0$ for $K \to \infty$.*

5. *If $g$ is Lipschitz, then $\|B_K[g] - g\|_\infty \in O(K^{-\frac{1}{2}})$ and if $g \in \mathcal{C}^2([0,1])$, then $\|B_K[g] - g\|_\infty \in O(K^{-1})$.*

*Proof.* See (Lorentz, 1986, Chp. 1). □

**Lemma A.2** (Data processing inequality for discrete $f$-divergences)**.** *For finite sets $X$ and $Y$ and a matrix $W \in [0,1]^{\#Y \times \#X}$ fulfilling $W \mathbb{1}_{\#Y} = \mathbb{1}_{\#X}$ and $P, Q \in \mathcal{P}(X)$ we have*

$$\mathrm{D}_f(W^\mathsf{T} P \mid W^\mathsf{T} Q) \le \mathrm{D}_f(P \mid Q).$$

*Proof.* See (Polyanskiy & Wu, 2025, Subsec. 7.2). □

# B. Proofs of theorems

First, we repeat some important properties of the rank statistics (de Frutos et al., 2024a, App. A) (de Frutos et al., 2026, Thm. 4.1).

## B.1. Properties of the rank statistic

**Lemma B.1** (Properties of the rank statistic)**.** *Let $\mu, \nu \in \mathcal{P}(\mathbb{R})$ with $\mu \ll \nu$ and $K \in \mathbb{N}_{>0}$, and let $(b_{n,K})_{n \in [K]}$ be the Bernstein polynomials. Suppose that $\nu$ is atomless.*

1. *We have*

$$Q_{\mu|\nu}^{(K)}(n) = \int_{\mathbb{R}} b_{n,K}(R_\nu(y)) \,\mathrm{d}\mu(y) = \int_0^1 b_{n,K}(s) \left(\frac{\mathrm{d}\mu}{\mathrm{d}\nu} \circ Q_\nu\right)(s) \,\mathrm{d}s, \qquad \forall n \in [K], \tag{11}$$

*where $R_\nu$ and $Q_\nu$ are the cumulative distribution function (CDF) and the quantile function of $\nu$, respectively.*

2. *Furthermore, we have $\mu = \nu$ if and only if for all $K \in \mathbb{N}$ we have $A_{\mu|\nu}^{(K)} \sim U_K$, i.e., $Q_{\mu|\nu}^{(K)}(n) = \frac{1}{K+1}$ for all $n \in [K]$.*

*Proof.* 1. Given $y \sim \mu$, the random variable $A_{\mu|\nu}^{(K)} \mid y$ follows a $\mathrm{Bin}(K, R_\nu(y))$ distribution, whose probability mass function is $[K] \ni n \mapsto b_{n,K}(R_\nu(y))$. By the law of total probability, we thus have

$$Q_{\mu|\nu}^{(K)}(n) = \mathbb{P}(A_{\mu|\nu}^{(K)} = n) = \int_{\mathbb{R}} \mathbb{P}\left(A_{\mu|\nu}^{(K)} = n \mid y\right) \,\mathrm{d}\mu(y) = \int_{\mathbb{R}} (b_{n,K} \circ R_\nu)(y) \,\mathrm{d}\mu(y)$$

as in (Elvira et al., 2021, Eq. (B.2)). The second equation follows from the change of variables formula for the pushforward measure.

2. If $\mu = \nu$ and $K \in \mathbb{N}$, then

$$Q_{\mu|\nu}^{(K)}(n) = \int_0^1 b_{n,K}(s) \left(\frac{\mathrm{d}\mu}{\mathrm{d}\nu} \circ Q_\nu\right)(s) \,\mathrm{d}s = \int_0^1 b_{n,K}(s) \,\mathrm{d}s = \frac{1}{K+1}, \qquad \forall n \in [K].$$

The converse direction follows as in the proof of (de Frutos et al., 2024a, Thm. 3), the assumption that $\mu$ and $\nu$ admit densities is not needed. □

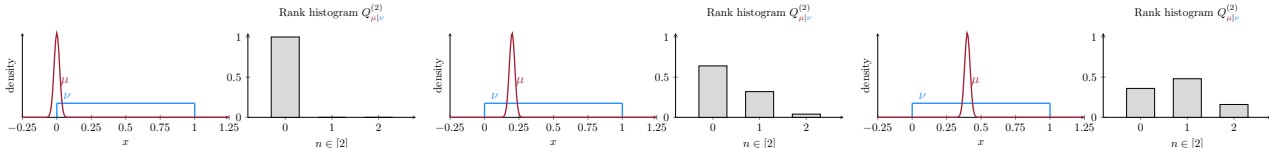

*Figure 6.* Illustration of the rank histogram $Q_{\mu|\nu}^{(K)}$ for $K = 2$, $\nu \sim U([0,1])$ and $\mu$ being a Gaussian with varying mean.

In the next remark, we illustrate why measuring the deviation of the rank histogram $Q_{\mu|\nu}^{(K)}$ using an $f$-divergence is meaningful.

*Remark* B.2 (Suitability of $f$-divergences). For $\mu, \nu \in \mathcal{P}(\mathbb{R})$ with $\mu \ll \nu$, we have

$$D_{f,\nu}(\mu) = \int_{\mathbb{R}} f\left(\frac{\mathrm{d}\mu}{\mathrm{d}\nu}(x)\right) \mathrm{d}[Q_\nu \# \lambda_{(0,1)}](x) = \int_0^1 f\left(\left(\frac{\mathrm{d}\mu}{\mathrm{d}\nu} \circ Q_\nu\right)(s)\right) \mathrm{d}s = D_{f,\lambda_{(0,1)}}\left(\frac{\mathrm{d}\mu}{\mathrm{d}\nu} \circ Q_\nu \cdot \lambda_{(0,1)}\right),$$

where $\lambda_{(0,1)}$ is the Lebesgue measure on $(0, 1)$. Hence, the $f$-divergence between $\mu$ and $\nu$ can be rewritten as the $f$-divergence between two densities on the (bounded) unit interval. The same holds for the $\alpha$-Rényi divergences, and we leave the exploration of rank-statistic approximations of Rényi divergences for future work.

This clean reformulation is not possible for other discrepancies, like integral probability metrics or Wasserstein distances.

## B.2. Proof of Theorem 2.3

**Theorem 2.3.** *Let $\mu, \nu \in \mathcal{P}(\mathbb{R})$ and $K \in \mathbb{N}$. The map $D_{f,\nu}^{(K)}$ is convex, and if $R_\nu$ is continuous, then it is also weakly lower semicontinuous. Furthermore,*

$$D_{f,\nu}^{(K)}(\mu) \leq D_{f,\nu}^{(K+1)}(\mu) \leq D_{f,\nu}(\mu). \tag{4}$$

*Proof.* 1. First, we show that $D_{f,\nu}^{(K)} \leq D_{f,\nu}$.

Let $U_{n,K} \sim \mathrm{Beta}(n+1, K-n+1)$ and $r := \frac{\mathrm{d}\mu}{\mathrm{d}\nu} \circ Q_\nu$. Then,

$$c_n^{(K)} := (K+1)\, Q_{\mu|\nu}^{(K)}(n) = \int_0^1 (K+1)b_{n,K}(u)\, r(u)\, \mathrm{d}u = \mathbb{E}\big[r(U_{n,K})\big],$$

By Jensen's inequality, we have

$$D_f^{(K)}(\mu \mid \nu) = \frac{1}{K+1} \sum_{n=0}^K f\big(c_n^{(K)}\big) \leq \frac{1}{K+1} \sum_{n=0}^K \mathbb{E}\big[f(r(U_{n,K}))\big]$$

$$= \frac{1}{K+1} \sum_{n=0}^K \int_0^1 f(r(u))(K+1)b_{n,K}(u)\, \mathrm{d}u = \int_0^1 f(r(u))\, \mathrm{d}u,$$

where in the last step we used $\frac{1}{K+1}\sum_{n=0}^K (K+1)b_{n,K} \equiv 1$.

2. Now, we prove the monotonicity with respect to $K$.

We want to use the data-processing inequality for discrete $f$-divergences (Lemma A.2) with $X := [K+1]$, $Y := [K]$, $P := Q_{\mu|\nu}^{(K+1)}$, and $Q := \mathcal{U}([K+1])$ and construct $W$ such that $W^\mathsf{T} P = Q_{\mu|\nu}^{(K+1)}$ and $W^\mathsf{T} Q = U_K$. Then,

$$D_{f,\nu}^{(K)}(\mu) = \mathrm{D}_f\left(Q_{\mu|\nu}^{(K)} \mid U_K\right) = \mathrm{D}_f\left(Q_{\mu|\nu}^{(K+1)} W \mid U_{K+1} W\right)$$

$$\leq \mathrm{D}_f\left(Q_{\mu|\nu}^{(K+1)} \,\|\, U_{K+1}\right) = D_{f,\nu}^{(K+1)}(\mu).$$

We set

$$W_{n,m} := \begin{cases} \frac{m}{K+1}, & \text{if } n = m-1, \\ \frac{K+1-m}{K+1}, & \text{if } n = m, \\ 0, & \text{otherwise.} \end{cases} \quad n \in [K],\ m \in [K+1].$$

Then,

$$W\, \mathbb{1}_{K+1} = \left(\sum_{n=0}^K W_{n,m}\right)_{m \in [K+1]} = (W_{m-1,m} + W_{m,m})_{m \in [K+1]} = \left(\frac{m}{K+1} + \frac{K+1-m}{K+1}\right)_{m \in [K+1]} = \mathbb{1}_{K+2}$$

and

$$W^\mathsf{T} Q = W^\mathsf{T} \mathcal{U}([K+1]) = \left( \sum_{m=0}^{K} W_{n,m} \frac{1}{K+2} \right)_{n \in [K]} = \left( \frac{1}{K+2} (W_{n,n} + W_{n,n+1}) \right)_{n \in [K]}$$

$$= \frac{1}{K+2} \left( \frac{K+1-n}{K+1} + \frac{n+1}{K+1} \right)_{n \in [K]} = \frac{1}{K+1} \mathbb{1}_{K+1} = U_K.$$

Next,

$$W^\mathsf{T} P = W^\mathsf{T} Q_{\mu|\nu}^{(K+1)} = \left( W_{n,n} Q_{\mu|\nu}^{(K+1)}(n) + W_{n,n+1} Q_{\mu|\nu}^{(K+1)}(n+1) \right)_{n \in [K]}$$

$$= \left( \frac{K+1-n}{K+1} \int_{\mathbb{R}} b_{n,K+1} \circ R_\nu \, \mathrm{d}\mu + \frac{n+1}{K+1} \int_{\mathbb{R}} b_{n,K+1} \circ R_\nu \, \mathrm{d}\mu \right)_{n \in [N]}$$

$$= \left( \int_{\mathbb{R}} b_{n,K} \circ R_\nu \, \mathrm{d}\mu \right)_{n \in [K]} = Q_{\mu|\nu}^{(K)}.$$

The identity about Bernstein polynomials we use in the last line expresses that starting from a count $n$ of "successes" among $K+1$ trials ($n$ of the $K+1$ samples that are drawn from $\nu$ are lower than $Y \sim \mu$), we delete one trial uniformly at random. With probability $n/(K+1)$ we delete a success and the new count is $n-1$, while with probability $(K+1-n)/(K+1)$ we delete a failure and the new count remains $n$.

3. By (11), $Q_{\mu|\nu}^{(K)}(n)$ is linear in $\mu$. Since $f$ is convex, the functional $D_{f,\nu}^{(K)}$ is convex as well.

4. If $Q_\nu$ is continuous, then by (11), $Q_{\mu|\nu}^{(K)}$ is weakly continuous, since the integrand is continuous. Hence, $D_{f,\nu}^{(K)}$ is weakly lower semicontinuous by the lower semicontinuity of $f$. $\qquad\square$

### B.3. Proof of Theorem 2.5

First, we prove the convergence if $r \in \mathcal{C}([0,1])$.

*Proof.* The only point not proved in the proof sketch in the main text is the uniform convergence of $r_K$ to $r$ for $K \to \infty$.

Indeed, for $u \in \left[ \frac{n}{K+1}, \frac{n+1}{K+1} \right)$ we have

$$|r_K(u) - r(u)| = \left| c_n^{(K)} - r(u) \right| \leq \left| c_n^{(K)} - r\left( \frac{n}{K} \right) \right| + \left| r\left( \frac{n}{K} \right) - r(u) \right|.$$

The second summand is bounded above as follows:

$$\left| r\left( \frac{n}{K} \right) - r(u) \right| \leq \sup \left\{ |r(x) - r(y)| : |x-y| \leq \frac{1}{K+1} \right\} =: \omega_r \left( \frac{1}{K+1} \right),$$

which is the *modulus of continuity* of $r$ at $\frac{1}{K+1}$. Hence,

$$\|r_K - r\|_\infty \leq \max_{n \in [K]} \left| c_n^{(K)} - r\left( \frac{n}{K} \right) \right| + \omega_r \left( \frac{1}{K+1} \right). \tag{12}$$

Since $r$ is uniformly continuous, the second summand vanishes for $K \to \infty$.

We now upper bound the first summand. Let $U_{n,K} \sim \mathrm{Beta}(n+1, K+1-n)$. The density of $U_{n,K}$ is $(K+1)b_{n,K}$ and its mean is $m_{n,K} := \frac{n+1}{K+2}$. For $n \in [K]$, we have

$$\left| \mathbb{E}[r(U_{n,K})] - r\left( \frac{n}{K} \right) \right| \leq \mathbb{E}\left[ |r(U_{n,K}) - r(m_{n,K})| \right] + \left| r(m_{n,K}) - r\left( \frac{n}{K} \right) \right|.$$

Furthermore,

$$\left| m_{n,K} - \frac{n}{K} \right| = \left| \frac{K-2n}{K(K+2)} \right| \leq \frac{1}{K+2} < \frac{1}{K} \to 0, \qquad K \to \infty, \tag{13}$$

We have

$$\mathbb{V}[U_{n,K}] = \frac{(n+1)(K-n+1)}{(K+2)^2(K+3)} \leq \frac{1}{4(K+3)} < \frac{1}{K}.$$ (14)

If the modulus of continuity $\omega_r$ is concave, then by Jensen's inequality, we have

$$\left|\mathbb{E}[r(U_{n,K})] - r\left(\frac{n}{K}\right)\right| \leq \mathbb{E}\left[\left|r(U_{n,K}) - r\left(\frac{n}{K}\right)\right|\right] \leq \mathbb{E}\left[\omega_r\left(U_{n,K} - \frac{n}{K}\right)\right]$$
$$\leq \omega_r\left(\mathbb{E}\left[\left|U_{n,K} - \frac{n}{K}\right|\right]\right) \leq \omega_r\left(\frac{1}{K+2} + \frac{1}{2\sqrt{K+3}}\right),$$

where we use Equations (13) and (14) in the last inequality. Hence, by (12),

$$\|r_K - r\|_\infty \leq \omega_r\left(\frac{1}{K+2} + \frac{1}{2\sqrt{K+3}}\right) + \omega_r\left(\frac{1}{K+1}\right) \xrightarrow{K\to\infty} 0.$$

$\square$

Now, let us prove the convergence rates.

**Lemma B.3.** *If $f$ is Lipschitz on* $\mathrm{ran}(r)$*, then*

$$D_{f,\nu}(\mu) - D_{f,\nu}^{(K)} \in \begin{cases} O(K^{-\frac{\alpha}{2}}), & \text{if } r \in C^{0,\alpha}([0,1]), \\ O(K^{-1}), & \text{if } r \in \mathcal{C}^2([0,1]), \text{or if } r \text{ is Lipschitz, } f \in \mathcal{C}^2([0,\infty)). \end{cases}$$

*Proof.*   1. If $r$ is $H_r$-Hölder continuous with exponent $\alpha \in (0,1]$, then $\omega_r(\delta) = H_r\delta^\alpha$, so the bound becomes

$$D_{f,\nu}(\mu) - D_{f,\nu}^{(K)}(\mu) \leq L_f H_r\left(\left(\frac{1}{K+2} + \frac{1}{2\sqrt{K+1}}\right)^\alpha + \left(\frac{1}{K+1}\right)^\alpha\right).$$

2. If $r \in \mathcal{C}^2([0,1])$. By Taylor's theorem, there exists a $\xi$ between $U_{n,K}$ and $\frac{n}{K}$ such that

$$r(U_{n,K}) = r\left(\frac{n}{K}\right) + r'\left(\frac{n}{K}\right)\left(U_{n,K} - \frac{n}{K}\right) + \frac{1}{2}r''(\xi)\left(U_{n,K} - \frac{n}{K}\right)^2,$$

so that

$$\left|c_K(n) - r\left(\frac{n}{K}\right)\right| \leq \|r'\|_\infty\left|m_{n,K} - \frac{n}{K}\right| + \frac{1}{2}\|r''\|_\infty \mathbb{E}\left[\left(U_{n,K} - \frac{n}{K}\right)^2\right],$$

so by the estimates from Equations (13) and (14) we obtain again

$$\max_{n\in[K]}\left|c_K(n) - r\left(\frac{n}{K}\right)\right| \leq \frac{\|r'\|_\infty}{K+2} + \frac{\|r''\|_\infty}{8(K+3)} + \frac{\|r''\|_\infty}{2(K+2)^2}.$$

Combined with $\omega_r\left(\frac{1}{K+1}\right) \leq \frac{\|r'\|_\infty}{K+1}$, we obtain $\|r_K - r\|_\infty \in O(K^{-1})$.

3. Now assume that $r$ is $L_r$–Lipschitz and $f \in C^2$ with $\|f''\|_\infty \leq M_f$. For brevity, set

$$Z_{n,K} := r(U_{n,K}), \qquad m_{n,K} := \mathbb{E}[Z_{n,K}].$$

By Taylor's theorem with remainder, for every $x \in \mathbb{R}$ there exists $\xi_x$ on the line segment between $x$ and $m_{n,K}$ such that

$$f(x) = f(m_{n,K}) + f'(m_{n,K})(x - m_{n,K}) + \frac{1}{2}f''(\xi_x)(x - m_{n,K})^2.$$

Hence,

$$\left|f(x) - f(m_{n,K}) - f'(m_{n,K})(x - m_{n,K})\right| \leq \frac{M_f}{2}(x - m_{n,K})^2.$$

Applying this Taylor expansion with $x = Z_{n,K}$, and using that $m_{n,K} = \mathbb{E}[Z_{n,K}]$ together with the bound $|f''| \leq M_f$ on the remainder, we obtain

$$
\begin{aligned}
J_{n,K} &= \mathbb{E}\left[f(Z_{n,K})\right] - f\left(\mathbb{E}[Z_{n,K}]\right) = \mathbb{E}\left[f(Z_{n,K}) - f(m_{n,K})\right] \\
&= \mathbb{E}\left[f(Z_{n,K}) - f(m_{n,K}) - f'(m_{n,K})(Z_{n,K} - m_{n,K}) + f'(m_{n,K})(Z_{n,K} - m_{n,K})\right] \\
&= \mathbb{E}\left[f(Z_{n,K}) - f(m_{n,K}) - f'(m_{n,K})(Z_{n,K} - m_{n,K})\right] + f'(m_{n,K})\,\mathbb{E}[Z_{n,K} - m_{n,K}] \\
&= \mathbb{E}\left[f(Z_{n,K}) - f(m_{n,K}) - f'(m_{n,K})(Z_{n,K} - m_{n,K})\right] \\
&\leq \mathbb{E}\left[\left|f(Z_{n,K}) - f(m_{n,K}) - f'(m_{n,K})(Z_{n,K} - m_{n,K})\right|\right] \\
&\leq \frac{M_f}{2}\,\mathbb{E}\left[(Z_{n,K} - m_{n,K})^2\right] = \frac{M_f}{2}\,\mathbb{V}(Z_{n,K}).
\end{aligned}
$$

$|r(u) - r(v)| \leq L_r |u - v|$. Using the characterization

$$
\mathbb{V}(Z_{n,K}) = \inf_{c \in \mathbb{R}} \mathbb{E}\left[(Z_{n,K} - c)^2\right],
$$

and taking $c = r(\mathbb{E}[U_{n,K}])$, we get

$$
\begin{aligned}
\mathbb{V}(Z_{n,K}) &= \inf_{c \in \mathbb{R}} \mathbb{E}\left[(r(U_{n,K}) - c)^2\right] \\
&\leq \mathbb{E}\left[(r(U_{n,K}) - r(\mathbb{E}[U_{n,K}]))^2\right] \leq L_r^2\,\mathbb{E}\left[(U_{n,K} - \mathbb{E}[U_{n,K}])^2\right] = L_r^2\,\mathbb{V}(U_{n,K}).
\end{aligned}
$$

Combining the two bounds yields

$$
J_{n,K} \leq \frac{M_f}{2}\,L_r^2\,\mathbb{V}(U_{n,K}).
$$

As in (14), for $U_{n,K} \sim \mathrm{Beta}(n+1, K-n+1)$ we have $\mathbb{V}(U_{n,K}) < \frac{1}{K}$. Hence

$$
J_{n,K} \leq \frac{M_f}{2}\,L_r^2\,\frac{1}{K} = \frac{M_f L_r^2}{2K}.
$$

Finally, averaging over $n$,

$$
0 \leq D_{f,\nu}(\mu) - D_{f,\nu}^{(K)}(\mu) = \frac{1}{K+1}\sum_{n=0}^{K} J_{n,K} \leq \frac{1}{K+1}\sum_{n=0}^{K} \frac{M_f L_r^2}{2K} = \frac{M_f L_r^2}{2K}. \qquad \square
$$

### B.4. Proof of Theorem 3.2

**Theorem 3.2.** *The map $\mathbf{D}_{f,\nu}^{(K)}$ is convex. Let $\mu, \nu \in \mathcal{P}(\mathbb{R}^d)$ with $\mu \ll \nu$. Then,*

$$
\mathbf{D}_{f,\nu}^{(K)}(\mu) \leq \mathrm{SD}_{f,\nu}(\mu) \leq D_{f,\nu}(\mu), \qquad K \in \mathbb{N}. \tag{6}
$$

*If $r_{\mu_s|\nu_s} \in \mathcal{C}([0,1])$ for almost all $s \in \mathbb{S}^{d-1}$, then*

$$
\lim_{K \to \infty} \mathbf{D}_{f,\nu}^{(K)}(\mu) = \mathrm{SD}_{f,\nu}(\mu),
$$

*Proof.* The convexity follows as in the proof of Theorem 2.3, because the pushforward is a linear operation. First, note that the absolute continuity $\mu \ll \nu$ implies that the projected measures satisfy $\mu_s \ll \nu_s$ for all $s \in \mathbb{S}^{d-1}$. By Theorem 2.5 we have the direction-wise convergence $\lim_{K \to \infty} D_{f,\nu_s}^{(K)}(\mu_s) = D_{f,\nu_s}(\mu_s)$. By Theorem 2.3 and by applying the data–processing inequality for $f$–divergences to the measurable map $x \mapsto s^\top x$, we have $D_{f,\nu_s}^{(K)}(\mu_s) \leq D_{f,\nu_s}(\mu_s) \leq D_{f,\nu}(\mu)$. Integrating over the sphere yields (6). Furthermore, we can thus apply the dominated convergence theorem and obtain

$$
\lim_{K \to \infty} \int_{\mathbb{S}^{d-1}} D_{f,\nu_s}^{(K)}(\mu_s)\,\mathrm{d}\sigma(s) = \int_{\mathbb{S}^{d-1}} \lim_{K \to \infty} D_{f,\nu_s}^{(K)}(\mu_s)\,\mathrm{d}\sigma(s) = \int_{\mathbb{S}^{d-1}} D_{f,\nu_s}(\mu_s)\,\mathrm{d}\sigma(s). \qquad \square
$$

### B.5. Proof of Theorem 2.6

**Lemma B.4** (Sampling $\mu$ only). *Fix $\nu \in \mathcal{P}(\mathbb{R})$ and $K \in \mathbb{N}$. Let $\mu \in \mathcal{P}(\mathbb{R})$ and let $\hat{\mu}_N$ be the empirical measure based on $N$ i.i.d. samples from $\mu$. Then, denoting $\mathbf{Q}_{\mu|\nu}^{(K)} := (Q_{\mu|\nu}^{(K)}(n))_{n \in [K]}$, we have*

$$\mathbb{E}\Big[\big\|\mathbf{Q}_{\hat{\mu}_N|\nu}^{(K)} - \mathbf{Q}_{\mu|\nu}^{(K)}\big\|_1\Big] \leq \frac{K+1}{2\sqrt{N}}.$$

*Proof.* Fix $n \in [K]$ and define, for $x \in \mathbb{R}$,

$$\varphi_n(x) := \mathbb{P}_{\tilde{Y}_1,\ldots,\tilde{Y}_K \sim \nu}\Big(\#\{j \in \{1,\ldots,K\} : \tilde{Y}_j \leq x\} = n\Big). \tag{15}$$

Then, $0 \leq \varphi_n(x) \leq 1$ for all $x$. By Equation (11),

$$Q_{\mu|\nu}^{(K)}(n) = \mathbb{P}\big(A_{\mu|\nu}^{(K)} = n\big) = \mathbb{E}_\mu[\varphi_n] \qquad \text{and} \qquad Q_{\hat{\mu}_N|\nu}^{(K)}(n) = \mathbb{E}_{\hat{\mu}_N}[\varphi_n] = \int_\mathbb{R} \varphi_n(x)\, \mathrm{d}\hat{\mu}_N(x) = \frac{1}{N}\sum_{i=1}^N \varphi_n(X_i).$$

Thus, for each $n$,

$$Q_{\hat{\mu}_N|\nu}^{(K)}(n) - Q_{\mu|\nu}^{(K)}(n) = \frac{1}{N}\sum_{i=1}^N \big(\varphi_n(X_i) - \mathbb{E}_\mu[\varphi_n]\big).$$

Since $X_1,\ldots,X_N \overset{\text{i.i.d.}}{\sim} \mu$ and $\varphi_n \colon \mathbb{R} \to [0,1]$ is a fixed deterministic function, the random variables $\varphi_n(X_1),\ldots,\varphi_n(X_N)$ are also i.i.d. and take values in $[0,1]$ with $\mathbb{V}(\varphi_n(X_1)) \leq \frac{1}{4}$. Set

$$Z_i := \varphi_n(X_i) - \mathbb{E}_\mu[\varphi_n], \qquad i \in \{1,\ldots,N\}.$$

Then $(Z_i)_{i=1}^N$ are i.i.d. and centered, so by independence

$$\mathbb{V}\big(Q_{\hat{\mu}_N|\nu}^{(K)}(n) - Q_{\mu|\nu}^{(K)}(n)\big) = \mathbb{V}\left(\frac{1}{N}\sum_{i=1}^N Z_i\right) = \frac{1}{N^2}\sum_{i=1}^N \mathbb{V}(Z_i) = \frac{1}{N}\mathbb{V}(Z_1) = \frac{1}{N}\mathbb{V}(\varphi_n(X_1)) \leq \frac{1}{4N}.$$

Applying Cauchy–Schwarz yields

$$\mathbb{E}\Big[\big\|\mathbf{Q}_{\hat{\mu}_N|\nu}^{(K)} - \mathbf{Q}_{\mu|\nu}^{(K)}\big\|_1\Big] \leq \sum_{n=0}^K \mathbb{E}\big[|Q_{\hat{\mu}_N|\nu}^{(K)}(n) - Q_{\mu|\nu}^{(K)}(n)|\big] \leq \sum_{n=0}^K \sqrt{\mathbb{V}\big(Q_{\hat{\mu}_N|\nu}^{(K)}(n) - Q_{\mu|\nu}^{(K)}(n)\big)} \leq \frac{K+1}{2\sqrt{N}}. \qquad \square$$

**Lemma B.5** (Sampling $\nu$ only). *Fix $K \in \mathbb{N}$. Let $\nu \in \mathcal{P}(\mathbb{R})$ and let $\hat{\nu}_M$ be the empirical measure based on $M$ i.i.d. samples from $\nu$. Then for any $\mu \in \mathcal{P}(\mathbb{R})$,*

$$\mathbb{E}\Big[\big\|\mathbf{Q}_{\mu|\hat{\nu}_M} - \mathbf{Q}_{\mu|\nu}\big\|_1\Big] \leq K\sqrt{\frac{2\pi}{M}},$$

*where $\mathbf{Q}$ is defined in Lemma B.4. In particular, the bound holds uniformly in $\mu$.*

*Proof.* Fix $\mu \in \mathcal{P}(\mathbb{R})$ and $K \in \mathbb{N}$. Let $R_\nu$ and $R_{\hat{\nu}_M}$ denote the CDFs of $\nu$ and $\hat{\nu}_M$. By (11),

$$Q_{\mu|\nu}^{(K)}(n) = \mathbb{E}_\mu\big[b_{n,K}(R_\nu)\big], \qquad Q_{\mu|\hat{\nu}_M}^{(K)}(n) = \mathbb{E}_\mu\big[b_{n,K}(R_{\hat{\nu}_M})\big].$$

By a simple coupling argument (alternatively combine (Kontorovich, 2026, Eq. (3))) with (Kontorovich, 2025, Eq. (4))),

$$\sum_{n=0}^K |b_{n,K}(t) - b_{n,K}(s)| = 2\, d_{\mathrm{TV}}\big(\mathrm{Bin}(K,s), \mathrm{Bin}(K,t)\big) \leq 2K\, |s - t|.$$

Now fix a realization of $\hat{\nu}_M$. For any $x \in \mathbb{R}$ we have

$$\sum_{n=0}^K \big|b_{n,K}(R_{\hat{\nu}_M}(x)) - b_{n,K}(R_\nu(x))\big| \leq 2K\, \big|R_{\hat{\nu}_M}(x) - R_\nu(x)\big|.$$

Hence,

$$
\left\|\mathbf{Q}_{\mu|\hat{\nu}_M} - \mathbf{Q}_{\mu|\nu}\right\|_1 = \sum_{n=0}^{K}\left|\mathbb{E}_\mu\left[b_{n,K}(R_{\hat{\nu}_M})\right] - \mathbb{E}_\mu\left[b_{n,K}(R_\nu)\right]\right| \le \mathbb{E}_\mu\left[\sum_{n=0}^{K}\left|b_{n,K}(R_{\hat{\nu}_M}) - b_{n,K}(R_\nu)\right|\right]
$$

$$
\le 2K\,\mathbb{E}_\mu\left[\left|R_{\hat{\nu}_M} - R_\nu\right|\right] \le 2K\sup_{x\in\mathbb{R}}\left|R_{\hat{\nu}_M}(x) - R_\nu(x)\right|.
$$

Taking expectations over $\hat{\nu}_M$ and applying the Dvoretzky–Kiefer–Wolfowitz (DKW) inequality (Massart, 1990),

$$
\mathbb{P}\left(\sup_{x\in\mathbb{R}}\left|R_{\hat{\nu}_M}(x) - R_\nu(x)\right| > t\right) \le 2e^{-2Mt^2}, \qquad t > 0,
$$

we obtain

$$
\mathbb{E}\left[\sup_{x\in\mathbb{R}}\left|R_{\hat{\nu}_M}(x) - R_\nu(x)\right|\right] = \int_0^\infty \mathbb{P}\left(\sup_{x\in\mathbb{R}}\left|R_{\hat{\nu}_M}(x) - R_\nu(x)\right| > t\right)\mathrm{d}t \le \int_0^\infty 2e^{-2Mt^2}\,\mathrm{d}t = \sqrt{\frac{\pi}{2M}}.
$$

Combining the two displays yields

$$
\mathbb{E}\left[\left\|\mathbf{Q}_{\mu|\hat{\nu}_M} - \mathbf{Q}_{\mu|\nu}\right\|_1\right] \le 2K\sqrt{\frac{\pi}{2M}}.
$$

The bound is uniform in $\mu$ because $\mu$ does not appear on the right-hand side. $\qquad\square$

### B.6. Proof of the univariate finite sample complexity bound Theorem 2.6

**Theorem 2.6** (Univariate finite sample complexity and concentration bound). *Let $K \in \mathbb{N}$ be fixed, and let $\mu, \nu \in \mathcal{P}(\mathbb{R})$ and $\hat{\mu}_N$ and $\hat{\nu}_M$ be their corresponding empirical measures with sample sizes $N$ and $M$. Suppose that $f$ is $L_f$-Lipschitz on $[0, K+1]$. The expected estimation error satisfies*

$$
\mathbb{E}\left[\left|D_{f,\hat{\nu}_M}^{(K)}(\hat{\mu}_N) - D_{f,\nu}^{(K)}(\mu)\right|\right]
$$

$$
\le L_f(K+1)\sqrt{2\pi}\left(\frac{1}{\sqrt{N}} + \frac{1}{\sqrt{M}}\right).
$$

*For any $\delta > 0$, with probability at least $1 - \delta$, we have*

$$
\left|D_{f,\hat{\nu}_M}^{(K)}(\hat{\mu}_N) - \mathbb{E}\left[D_{f,\hat{\nu}_M}^{(K)}(\hat{\mu}_N)\right]\right|
$$

$$
\le L_f(K+1)\sqrt{2\log(2/\delta)\left(\frac{1}{N} + \frac{1}{M}\right)}.
$$

*Proof of Theorem 2.6.* For $K \in \mathbb{N}$ we have

$$
\left|D_{f,\hat{\nu}_M}^{(K)}(\hat{\mu}_N) - D_{f,\nu}^{(K)}(\mu)\right| \le \frac{1}{K+1}\sum_{n=0}^{K}\left|f\left((K+1)Q_{\hat{\mu}_N|\hat{\nu}_M}^{(K)}(n)\right) - f\left((K+1)Q_{\mu|\nu}^{(K)}(n)\right)\right|
$$

$$
\le \frac{1}{K+1}\sum_{n=0}^{K}L_f(K+1)\left|Q_{\hat{\mu}_N|\hat{\nu}_M}^{(K)}(n) - Q_{\mu|\nu}^{(K)}(n)\right| = L_f\sum_{n=0}^{K}\left|Q_{\hat{\mu}_N|\hat{\nu}_M}^{(K)}(n) - Q_{\mu|\nu}^{(K)}(n)\right|,
$$

(16)

since for each $n \in [K]$, the quantities $(K+1)Q_{\hat{\mu}_N|\hat{\nu}_M}^{(K)}(n)$ and $(K+1)Q_{\mu|\nu}^{(K)}(n)$ lie in the interval $[0, K+1]$, and $f$ is $L_f$–Lipschitz on $[0, K+1]$. Taking expectations, we obtain

$$
\mathbb{E}\left[\left|D_f^{(K)}(\hat{\mu}_N \mid \hat{\nu}_M) - D_f^{(K)}(\mu \mid \nu)\right|\right] \le L_f\,\mathbb{E}\left[\sum_{n=0}^{K}\left|Q_{\hat{\mu}_N|\hat{\nu}_M}^{(K)}(n) - Q_{\mu|\nu}^{(K)}(n)\right|\right].
$$

(17)

We now decompose the rank-PMF error into two contributions: the error due to sampling $\mu$ and the error due to sampling $\nu$. By the triangle inequality,

$$
\sum_{n=0}^{K}\left|Q_{\hat{\mu}_N|\hat{\nu}_M}^{(K)}(n) - Q_{\mu|\nu}^{(K)}(n)\right| \le \sum_{n=0}^{K}\left|Q_{\hat{\mu}_N|\hat{\nu}_M}^{(K)}(n) - Q_{\hat{\mu}_N|\nu}^{(K)}(n)\right| + \sum_{n=0}^{K}\left|Q_{\hat{\mu}_N|\nu}^{(K)}(n) - Q_{\mu|\nu}^{(K)}(n)\right|.
$$

Taking expectations and applying Lemma B.5 (which holds uniformly with respect to the argument of $D_{f,\nu}^{(K)}$, so it can be used with the random $\hat{\mu}_N$) and Lemma B.4, we obtain

$$
\mathbb{E}\Big[\sum_{n=0}^{K}|Q_{\hat{\mu}_N|\hat{\nu}_M}^{(K)}(n) - Q_{\mu|\nu}^{(K)}(n)|\Big] \le \mathbb{E}\Big[\sum_{n=0}^{K}|Q_{\hat{\mu}_N|\hat{\nu}_M}^{(K)}(n) - Q_{\hat{\mu}_N|\nu}^{(K)}(n)|\Big] + \mathbb{E}\Big[\sum_{n=0}^{K}|Q_{\hat{\mu}_N|\nu}^{(K)}(n) - Q_{\mu|\nu}^{(K)}(n)|\Big]
$$
$$
\le 2K\sqrt{\frac{\pi}{2M}} + \frac{K+1}{2\sqrt{N}} \le (K+1)\sqrt{2\pi}\left(\frac{1}{\sqrt{N}} + \frac{1}{\sqrt{M}}\right),
$$
(18)

where we used $\frac{1}{2} \le \sqrt{2\pi}$ to simplify the constants. $\qquad\square$

### B.7. Proof of the concentration bound

*Proof.* Let $K \in \mathbb{N}$ and define the functional

$$
F(X_1, \ldots, X_N, Y_1, \ldots, Y_M) := D_{f,\hat{\nu}_M}^{(K)}(\hat{\mu}_N),
$$

where the empirical measures $\hat{\mu}_N$ and $\hat{\nu}_M$ are given by (5). We will apply *McDiarmid's bounded differences inequality* (see (Boucheron et al., 2013, Sec 6)) to $F$. We first quantify how much $F$ can change when we replace one observation $X_i$ in the sample from $\mu$, while keeping all other data points fixed. Consider two datasets that differ only in the $i$-th sample from $\mu$:

$$
(X_1, \ldots, X_i, \ldots, X_N, Y_1, \ldots, Y_M), \quad (X_1, \ldots, X_i', \ldots, X_N, Y_1, \ldots, Y_M),
$$

and fix $Y_1, \ldots, Y_M$. Thus $\hat{\nu}_M$ is the same in both cases, while the empirical measure of $\mu$ changes from $\hat{\mu}_N$ to

$$
\hat{\mu}_N' = \hat{\mu}_N - \frac{1}{N}\delta_{X_i} + \frac{1}{N}\delta_{X_i'}.
$$

Fix $\hat{\nu}_M$ and $K \in \mathbb{N}$. For each $n \in [K]$ define $\varphi_n$ by (15). Again,

$$
Q_{\hat{\mu}_N|\hat{\nu}_M}^{(K)}(n) = \frac{1}{N}\sum_{k=1}^{N}\varphi_n(X_k), \qquad Q_{\hat{\mu}_N'|\hat{\nu}_M}^{(K)}(n) = \frac{1}{N}\sum_{k=1}^{N}\varphi_n(X_k'),
$$

where $X_k' = X_k$ for $k \ne i$ and $X_i' = X_i'$. Hence

$$
Q_{\hat{\mu}_N'|\hat{\nu}_M}^{(K)}(n) - Q_{\hat{\mu}_N|\hat{\nu}_M}^{(K)}(n) = \frac{1}{N}\big(\varphi_n(X_i') - \varphi_n(X_i)\big).
$$

Since $0 \le \varphi_n \le 1$, we obtain

$$
\big\|Q_{\hat{\mu}_N'|\hat{\nu}_M}^{(K)} - Q_{\hat{\mu}_N|\hat{\nu}_M}^{(K)}\big\|_1 \le \frac{K+1}{N}.
$$

Applying Equation (16) with $(\hat{\mu}_N' \mid \hat{\nu}_M)$ instead of $(\mu \mid \nu)$, we obtain

$$
\big|F(X_1, \ldots, X_i', \ldots, X_N, Y_1, \ldots, Y_M) - F(X_1, \ldots, X_i, \ldots, X_N, Y_1, \ldots, Y_M)\big|
$$
$$
= \big|D_{f,\hat{\nu}_M}^{(K)}(\hat{\mu}_N') - D_{f,\hat{\nu}_M}^{(K)}(\hat{\mu}_N)\big| \le L_f\big\|Q_{\hat{\mu}_N'|\hat{\nu}_M}^{(K)} - Q_{\hat{\mu}_N|\hat{\nu}_M}^{(K)}\big\|_1 \le L_f\frac{K+1}{N}.
$$

Now, consider two datasets that differ only in $Y_j$:

$$
(X_1, \ldots, X_N, Y_1, \ldots, Y_j, \ldots, Y_M), \quad (X_1, \ldots, X_N, Y_1, \ldots, Y_j', \ldots, Y_M),
$$

so that $\hat{\mu}_N$ is fixed, while the empirical measure of $\nu$ changes from $\hat{\nu}_M$ to $\hat{\nu}_M' = \hat{\nu}_M - \frac{1}{M}\delta_{Y_j} + \frac{1}{M}\delta_{Y_j'}$. Changing a single atom in an empirical measure of size $M$ changes the CDF by at most $1/M$, that is,

$$
\sup_{x \in \mathbb{R}}|R_{\hat{\nu}_M'}(x) - R_{\hat{\nu}_M}(x)| \le \frac{1}{M}.
$$

As in the proof of Lemma B.5, we obtain

$$
\left\| Q_{\hat{\mu}_N|\hat{\nu}'_M}^{(K)} - Q_{\hat{\mu}_N|\hat{\nu}_M}^{(K)} \right\|_1 = \sum_{n=0}^{K} \left| \mathbb{E}_{\hat{\mu}_N}\left[ b_{n,K}(R_{\hat{\nu}'_M}) \right] - \mathbb{E}_{\hat{\mu}_N}\left[ b_{n,K}(R_{\hat{\nu}_M}) \right] \right| \le \mathbb{E}_{\hat{\mu}_N}\left[ \sum_{n=0}^{K} \left| b_{n,K}(R_{\hat{\nu}'_M}) - b_{n,K}(R_{\hat{\nu}_M}) \right| \right]
$$

$$
\le 2K\, \mathbb{E}_{\hat{\mu}_N}\left[ |R_{\hat{\nu}'_M} - R_{\hat{\nu}_M}| \right] \le 2K \sup_{x\in\mathbb{R}} |R_{\hat{\nu}'_M}(x) - R_{\hat{\nu}_M}(x)| \le \frac{2K}{M}.
$$

Applying Equation (16) with $(\hat{\mu}_N, \hat{\nu}'_M)$ instead of $(\mu \mid \nu)$, we obtain

$$
\left| F(X_1, \ldots, X_N, Y_1, \ldots, Y'_j, \ldots, Y_M) - F(X_1, \ldots, X_N, Y_1, \ldots, Y_j, \ldots, Y_M) \right|
$$

$$
= \left| D_{f,\hat{\nu}'_M}^{(K)}(\hat{\mu}_N) - D_{f,\hat{\nu}_M}^{(K)}(\hat{\mu}_N) \right| \le L_f \left\| Q_{\hat{\mu}_N|\hat{\nu}'_M}^{(K)} - Q_{\hat{\mu}_N|\hat{\nu}_M}^{(K)} \right\|_1 \le L_f \frac{2K}{M}.
$$

We have shown that $F$ satisfies the bounded differences condition with

$$
c_i = L_f \frac{K+1}{N} \quad (i = 1, \ldots, N), \qquad c_{N+j} = L_f \frac{2K}{M} \quad (j = 1, \ldots, M).
$$

Hence, using $K^2 \le (K+1)^2$,

$$
\sum_{i=1}^{N+M} c_i^2 = \sum_{i=1}^{N} \left( L_f \frac{K+1}{N} \right)^2 + \sum_{j=1}^{M} \left( L_f \frac{2K}{M} \right)^2 \le 4 L_f^2 (K+1)^2 \left( \frac{1}{N} + \frac{1}{M} \right),
$$

By McDiarmid's inequality (Theorem 6.2 in Boucheron et al. (2013)), for any $t > 0$,

$$
\mathbb{P}\Big( |F - \mathbb{E}[F]| \ge t \Big) \le 2\exp\left( -\frac{2t^2}{\sum_{i=1}^{N+M} c_i^2} \right) \le 2\exp\left( -\frac{t^2}{2 L_f^2 (K+1)^2 \left( \frac{1}{N} + \frac{1}{M} \right)} \right).
$$

Let $\delta > 0$ and choose

$$
t = L_f(K+1)\sqrt{2\log(2/\delta)\left( \frac{1}{N} + \frac{1}{M} \right)}.
$$

Then

$$
\mathbb{P}\Big( |F - \mathbb{E}[F]| \ge t \Big) \le \delta.
$$

Since $F = D_{f,\hat{\nu}_M}^{(K)}(\hat{\mu}_N)$, this is exactly the desired bound: with probability at least $1 - \delta$,

$$
\left| D_{f,\hat{\nu}_M}^{(K)}(\hat{\mu}_N) - \mathbb{E}\left[ D_{f,\hat{\nu}_M}^{(K)}(\hat{\mu}_N) \right] \right| \le L_f(K+1)\sqrt{2\log(2/\delta)\left( \frac{1}{N} + \frac{1}{M} \right)}. \qquad \square
$$

### B.8. Proof of the asymptotic normality Theorem 3.3

**Theorem 3.3** (Asymptotic normality, sliced one-sample case). *Fix $K \in \mathbb{N}$ and $\mu, \nu \in \mathcal{P}(\mathbb{R}^d)$ with $\mu \ne \nu$ and $D_{f,\nu}^{(K)}(\mu) > 0$, and let $\hat{\mu}_N$ be the empirical approximation from (5). If $f \in \mathcal{C}^1([0, K+1])$ and $f'$ is Lipschitz, then there exists a constant $\tau_K^2 \ge 0$ such that, in distribution,*

$$
\sqrt{N}\Big( \mathbf{D}_{f,\nu}^{(K)}(\hat{\mu}_N) - \mathbf{D}_{f,\nu}^{(K)}(\mu) \Big) \xrightarrow[N\to\infty]{d} \mathcal{N}(0, \tau_K^2).
$$

*Proof.* 1. For each direction $s \in \mathbb{S}^{d-1}$ and $n \in [K]$, define $\varphi_{n,s} : \mathbb{R}^d \to [0,1]$ as the probability that a sample $x$ has rank $n$ among $K$ draws $\tilde{Y}_1, \ldots, \tilde{Y}_K \overset{\text{i.i.d.}}{\sim} \nu$ when both are projected along $s$, i.e.,

$$
\varphi_{n,s}(x) := \mathbb{P}\Big( \#\{j \in \{1, \ldots, K\} : s^\top \tilde{Y}_j \le s^\top x\} = n \Big).
$$

As in the univariate case (compare (11)) one checks that for any $\mu \in \mathcal{P}(\mathbb{R}^d)$ and $s \in \mathbb{S}^{d-1}$,

$$Q_{\mu_s|\nu_s}^{(K)}(n) = \mathbb{E}_\mu[\varphi_{n,s}] \qquad \text{and} \qquad Q_{(\hat{\mu}_N)_s|\nu_s}^{(K)}(n) = \frac{1}{N}\sum_{i=1}^N \varphi_{n,s}(X_i), \tag{19}$$

where the empirical version $\hat{\mu}_N$ is defined in (5). Define the Hilbert space $\mathcal{H} := L^2\big(\mathbb{S}^{d-1} \times [K], \sigma \otimes U_K\big)$ with inner product

$$\langle h, g \rangle_\mathcal{H} := \int_{\mathbb{S}^{d-1}} \frac{1}{K+1}\sum_{n=0}^K h(s,n)\, g(s,n)\, \mathrm{d}\sigma(s), \qquad g, h \in \mathcal{H}.$$

For $x \in \mathbb{R}^d$, define the random element $\Phi(x) \in \mathcal{H}$ by $\Phi(x)(s,n) := \varphi_{n,s}(x)$. Then $\|\Phi(x)\|_\mathcal{H} \leq 1$ because $0 \leq \varphi_{n,s}(x) \leq 1$. Let

$$T := \mathbb{E}_\mu[\Phi] \in \mathcal{H}, \qquad T_N := \frac{1}{N}\sum_{i=1}^N \Phi(X_i) \in \mathcal{H}.$$

By the identities (19), we have

$$T(s,n) = \mathbb{E}_\mu[\varphi_{n,s}] = Q_{\mu_s|\nu_s}^{(K)}(n), \qquad T_N(s,n) = Q_{(\hat{\mu}_N)_s|\nu_s}^{(K)}(n).$$

The random elements $(\Phi(X_k))_{k \in \mathbb{N}}$ are i.i.d. in $\mathcal{H}$ with $\mathbb{E}_\mu[\|\Phi\|_\mathcal{H}^2] \leq 1$. Hence, the standard central limit theorem in separable Hilbert spaces applies, and we obtain

$$\sqrt{N}\big(T_N - T\big) \xrightarrow{d} \mathcal{G} \quad \text{in } \mathcal{H},$$

where $\mathcal{G}$ is a mean-zero Gaussian element in $\mathcal{H}$ with covariance operator determined by $\Phi$.

2. We now write the rank-statistic $f$-divergence and its approximation by samples as $\Psi(T)$ and $\Psi(T_N)$ for some $\Psi$ and use the delta-method.

Indeed, for

$$\Psi \colon \mathcal{H} \to \mathbb{R}, \qquad h \mapsto \int_{\mathbb{S}^{d-1}} \frac{1}{K+1}\sum_{n=0}^K f\big((K+1)h(s,n)\big)\, \mathrm{d}\sigma(s)$$

we have

$$D_{f,\nu}^{(K)}(\mu) = \Psi(T) = \int_{\mathbb{S}^{d-1}} \frac{1}{K+1}\sum_{n=0}^K f\Big((K+1)\, Q_{\mu_s|\nu_s}^{(K)}(n)\Big)\, \mathrm{d}\sigma(s) \qquad \text{and} \qquad D_{f,\nu}^{(K)}(\hat{\mu}_N) = \Psi(T_N).$$

We now show that $\Psi$ is differentiable. Since $f \in \mathcal{C}^1([0, K+1])$ and $f'$ is Lipschitz, $\Psi$ is Fréchet differentiable at $T$ with Lipschitz continuous derivative

$$D\Psi(T) \colon \mathcal{H} \to \mathbb{R}, \qquad h \mapsto \int_{\mathbb{S}^{d-1}} \sum_{n=0}^K f'\big((K+1)T(s,n)\big)\, h(s,n)\, \mathrm{d}\sigma(s).$$

In particular, at $T(s,n) = Q_{\mu_s|\nu_s}^{(K)}(n)$ this becomes

$$D\Psi(T)[h] = \int_{\mathbb{S}^{d-1}} \sum_{n=0}^K f'\big((K+1)Q_{\mu_s|\nu_s}^{(K)}(n)\big)\, h(s,n)\, \mathrm{d}\sigma(s),$$

which defines a bounded linear functional on $\mathcal{H}$.

By the Hilbert-space delta method (van der Vaart & Wellner, 1996, Thm. 3.9.4), the combination of the CLT for $T_N$ and the Fréchet differentiability of $\Psi$ at $T$ implies

$$\sqrt{N}\big(\Psi(T_N) - \Psi(T)\big) \xrightarrow{d} \mathcal{N}\big(0, \tau_K^2\big),$$

with asymptotic variance

$$\tau_K^2 = \mathbb{V}\big(D\Psi(T)[\Phi(X_1) - T]\big) \geq 0.$$

We have $\tau_K^2 > 0$ if $\int_{\mathbb{S}^{d-1}} \sum_{n=0}^K f'\big((K+1)Q_{\mu_s|\nu_s}^{(K)}(n)\big) b_{n,K}(R_{\nu_s}(s^\top x))\, \mathrm{d}\sigma(s)$ is not constant on $\mathrm{supp}(\mu)$.

Recalling that $\Psi(T) = D_{f,\nu}^{(K)}(\mu)$ and $\Psi(T_N) = D_{f,\nu}^{(K)}(\hat{\mu}_N)$, we obtain the desired conclusion. $\qquad \square$

# C. Experiments

## C.1. Neural vs. rank-statistic divergence estimation across dimensions

This appendix provides the full experimental details for the benchmark in Section 4 comparing the proposed rank-statistic estimator to the neural KL-divergence estimator of Sreekumar et al. (2021) on the truncated-Gaussian vs. uniform setup.

**Distributions and supports.** For each dimension $d \in \{2, 5, 10\}$, the target distribution $\mu$ is the standard Gaussian $\mathcal{N}(0, I_d)$ truncated (and renormalized) to an axis-aligned box $X$, and the reference distribution is the uniform measure $\nu = \mathrm{Unif}(X)$. Concretely,
$$X_2 = [0.1, 2] \times [-1, 0], \qquad X_5 = [0.1, 2] \times [-1, 0] \times [2, 3] \times [-2, -1.5] \times [-1, 1],$$
and for $d = 10$ we use the product support $X_{10} = X_5 \times X_5$. Sampling from $\nu$ is done by drawing each coordinate independently and uniformly over the corresponding interval. Sampling from $\mu$ is done by accept-reject: draw from $\mathcal{N}(0, I_d)$ until the sample falls in $X$.

**Sample sizes and repetitions.** For each $n \in 10^4 \cdot \{1, 2, 4, 8, 16, 32, 64, 128, 256, 512\}$ we draw $n$ i.i.d. samples from $\mu$ and $n$ i.i.d. samples from $\nu$, and repeat the whole procedure over $R = 10$ independent random seeds. Figure 2 reports the mean and $\pm 1$ standard deviation bands over these runs (for both estimators).

**Analytic KL reference.** In this benchmark, the "ground-truth" $\mathrm{KL}(\mu\|\nu)$ shown as a dashed line in Figure 2 is computed analytically (no additional Monte Carlo layer). Let $\tilde{\mu} = \mathcal{N}(0, I_d)$ denote the untruncated Gaussian density and let $X = \prod_{j=1}^{d} [a_j, b_j]$ be the truncation box. The truncated density is $p_\mu(x) = \tilde{p}(x) \mathbf{1}\{x \in X\}/Z$, where $Z = \int_X \tilde{p}(x)\, dx$ is the truncation mass, and $p_\nu(x) = 1/\mathrm{Vol}(X)$ for $x \in X$. Using $\mathrm{KL}(\mu\|\nu) = \mathbb{E}_\mu[\log p_\mu(X) - \log p_\nu(X)]$ and separability of the standard Gaussian, $Z$ factorizes as $Z = \prod_{j=1}^{d}(\Phi(b_j) - \Phi(a_j))$, where $\Phi$ denotes the CDF of the standard normal distribution (and $\varphi$ its density). The remaining expectation reduces to the sum of one-dimensional truncated moments $\mathbb{E}_\mu[X_j^2]$ (available in closed form via $\varphi$ and $\Phi$). This yields an exact value for each $d$ and box $X$; the implementation follows these standard identities.

**Neural baseline (full protocol).** We compare against the neural KL-divergence estimator of Sreekumar et al. (2021) and follow their protocol exactly. For each sample size $n$, the network width is set to $k = \lceil n^{1/5} \rceil$ and the model is trained for 200 epochs with Adam, using learning rate $10^{-2}$ and a single decay to $10^{-3}$ after 100 epochs. Minibatches have size $10^{-3}n$. Results are averaged over the same $R = 10$ seeds used for the rank-based estimator. (All remaining hyperparameters and architectural details are as in Sreekumar et al. (2021).)

**Rank-statistic estimator settings.** In contrast, the rank-statistic estimator requires no iterative optimization: once the samples are fixed, the estimate is fully determined by the rank resolution $K$. In this benchmark, we exploit the product structure of the distributions and use the axis-corrected estimator described in the main text: we compute the one-dimensional degree-$K$ rank-statistic KL terms along each coordinate axis and sum them over coordinates. Thus, no random projection parameter $L$ is used for this experiment. Unless stated otherwise, all randomness in the rank estimator comes solely from the sampled data.

## C.2. Univariate empirical convergence and the influence of resolution $K$

This appendix provides additional implementation details for the one-dimensional benchmarks reported in Section 4.2. For each configuration we draw $n_\mu$ i.i.d. samples from $\mu$ and $n_\nu$ i.i.d. samples from $\nu$, construct the Bernstein rank histogram $Q_{\mu|\nu}^{(K)}$, and compute the corresponding discrete $f$-divergence $D_{f,\nu}^{(K)}(\mu)$. We repeat each setting for $R = 10$ seeds and report mean±std.

**Distributions and supports.** We study four representative mismatch families that collectively capture shifts in location and scale, departures from unimodality, and tail mismatch:

- Location shift (Gaussian mean). We set $\mu = \mathcal{N}(0, 1)$ and $\nu = \mathcal{N}(\Delta, 1)$ with $\Delta \in \{0, 0.5, 1, 2\}$, and report *JS*, *KL*, and *TV*.

- Scale change (Gaussian variance). We take $\mu = \mathcal{N}(0, 1)$ and $\nu = \mathcal{N}(0, \sigma)$ for $\sigma \in \{1, 1.2, 1.5, 2\}$, and report *KL*, squared Hellinger (*Hell²*), and *TV*.

- Multimodality (mixture vs. unimodal). To probe sensitivity to multiple modes, we compare the symmetric mixture $\mu = \frac{1}{2}\mathcal{N}(-\Delta, 1) + \frac{1}{2}\mathcal{N}(+\Delta, 1)$ against $\nu = \mathcal{N}(0, 1)$ over the same $\Delta$ values, and report *JS*, *KL*, and *TV*.

- Tail mismatch (heavy-tailed vs. Gaussian). Finally, we compare $\mu = \text{Laplace}(0, 1)$ to $\nu = \mathcal{N}(0, 1)$ and report *JS*, *KL*, and *TV*.

Unless stated otherwise we use $n_\mu = n_\nu = 10{,}000$, and we evaluate $K \in \{32, 64, 128, 256, 512\}$.

**Reference.** Whenever a closed form is available, we use it as ground truth (in particular, Gaussian-Gaussian KL, squared Hellinger, and TV for Gaussian mean/scale changes). For Jensen-Shannon (JS), we compute a high-accuracy reference from

$$\text{JS}(\mu, \nu) = \frac{1}{2}\, \mathbb{E}_{X \sim \mu}\left[\log \frac{2\, p_\mu(X)}{p_\mu(X) + p_\nu(X)}\right] + \frac{1}{2}\, \mathbb{E}_{Y \sim \nu}\left[\log \frac{2\, p_\nu(Y)}{p_\mu(Y) + p_\nu(Y)}\right],$$

and evaluate the resulting one-dimensional expectations by numerical quadrature with a tight tolerance. All evaluations are carried out in the log domain to avoid numerical issues in the tails. In the mixture-vs-Gaussian setting, the mixture density $p_\mu$ is computed exactly as the average of its two Gaussian components inside the same procedure.

For settings where our reference is not available in closed form in our implementation, namely, mixture-vs-Gaussian KL and TV, as well as $\text{Laplace}(0, 1)$ vs. $\mathcal{N}(0, 1)$ for JS/KL/TV, we compute a single high-sample Monte Carlo reference once and reuse it across all $R$ runs. Specifically, we draw $n_{\text{ref}} = 10^7$ i.i.d. samples from each distribution and plug them into the corresponding expectation identity (JS as above, and analogously for KL/TV).

For each configuration and each $K$, we report the estimate $\widehat{D}_{f,\nu}^{(K)}(\mu)$ (mean±std over $R$ seeds) and the ratio $\widehat{D}_{f,\nu}^{(K)}(\mu)/D_{f,\nu}(\mu)$ to summarize the finite-$K$ approximation gap.

**Results** Taken together, Table 3 and Figures 7-9 show that increasing the rank resolution $K$ systematically closes the finite-$K$ gap: the estimate/reference ratio moves toward 1 across all mismatch families and all reported divergences (JS, KL, Hell², and the added TV), with near-unbiased behavior already for moderate $K$ in the smooth Gaussian cases (mean/scale shifts), while multimodality and tail mismatch require larger $K$ to reduce bias and typically exhibit larger finite-sample variability. In particular, TV is already accurate at moderate $K$ in the Gaussian settings (after the $\frac{1}{2}$ conversion from the $\ell_1$ form returned by our implementation), whereas KL under heavy tails remains substantially underestimated even at the largest $K$ considered, reflecting the increased difficulty of capturing tail contributions with finite rank resolution. At fixed $K = 256$, the estimator remains stable across a range of shift magnitudes and scales, as summarized in Figure 8. The $K$-sweeps in Figure 7 and, especially, the log-log plot in Figure 9 further clarify the convergence hierarchy by visualizing the absolute ratio error $|1 - \text{Ratio}|$ versus $K$: the Gaussian scale-change case decays the fastest (tracking a near-optimal $\mathcal{O}(K^{-1})$ slope), the Gaussian mean-shift case converges more slowly (consistent with reduced tail regularity), and the Laplace-vs-Gaussian heavy-tail mismatch improves the slowest, remaining closest to the shallowest guide slope. This separation aligns with the qualitative ordering suggested by the regularity-based rate discussion in the main text and motivates the larger-$K$ settings used in the tail-mismatch experiments.

### C.3. Sliced rank-statistic $f$-divergences: empirical convergence

This appendix provides the full experimental protocol underlying Section 4.3. For each configuration (dimension $d$ and distribution pair $(\mu, \nu)$), we draw $n_\mu$ i.i.d. samples from $\mu$ and $n_\nu$ i.i.d. samples from $\nu$, compute the sliced rank-$f$ estimate using $L$ random directions and rank order $K$, and repeat the procedure for $R$ independent runs, reporting mean±std. Unless stated otherwise we use $K = 64$, $L = 128$, $n_\mu = n_\nu = 10{,}000$, and $R = 10$. Random directions are sampled uniformly on $\mathbb{S}^{d-1}$.

**Distribution families.** We consider the following multivariate mismatch families (in dimension $d$): (i) Gaussian mean shifts, $\mu = \mathcal{N}(0, I_d)$ and $\nu = \mathcal{N}(\Delta e_1, I_d)$ with $\Delta \in \{0, 0.5, 1.0\}$; (ii) Gaussian scale changes, $\mu = \mathcal{N}(0, I_d)$ and $\nu = \mathcal{N}(0, \sigma^2 I_d)$ with $\sigma \in \{1.0, 1.2, 1.5, 2.0\}$; (iii) anisotropic Gaussian covariance mismatch, $\mu = \mathcal{N}(0, I_d)$ and $\nu = \mathcal{N}(0, \text{diag}(1, \ldots, 2))$; and (iv) non-Gaussian comparisons, including factorized Laplace vs. Gaussian for JS, Student-$t$ vs. Gaussian for KL, and a symmetric two-component Gaussian mixture vs. Gaussian for JS.

| Family | Scen. | Param. | Ratio $D_{f,\nu}^{(K)}(\mu)$ for $K =$ | | | | |
| --- | --- | --- | --- | --- | --- | --- | --- |
| | | | 32 | 64 | 128 | 256 | 512 |
| Mean shift | JS | $\Delta = 0.5$ | $0.933 \pm 0.040$ | $0.968 \pm 0.041$ | $0.989 \pm 0.042$ | $1.003 \pm 0.042$ | $1.013 \pm 0.042$ |
| | JS | $\Delta = 1.0$ | $0.928 \pm 0.033$ | $0.961 \pm 0.034$ | $0.981 \pm 0.035$ | $0.992 \pm 0.035$ | $0.999 \pm 0.035$ |
| | JS | $\Delta = 2.0$ | $0.930 \pm 0.008$ | $0.962 \pm 0.008$ | $0.981 \pm 0.009$ | $0.991 \pm 0.009$ | $0.997 \pm 0.009$ |
| | KL | $\Delta = 0.5$ | $0.946 \pm 0.060$ | $0.987 \pm 0.063$ | $1.013 \pm 0.065$ | $1.030 \pm 0.066$ | $1.044 \pm 0.068$ |
| | KL | $\Delta = 1.0$ | $0.880 \pm 0.024$ | $0.924 \pm 0.025$ | $0.952 \pm 0.026$ | $0.969 \pm 0.027$ | $0.980 \pm 0.027$ |
| | KL | $\Delta = 2.0$ | $0.775 \pm 0.010$ | $0.844 \pm 0.012$ | $0.895 \pm 0.013$ | $0.933 \pm 0.015$ | $0.959 \pm 0.016$ |
| | TV | $\Delta = 0.5$ | $0.979 \pm 0.026$ | $0.991 \pm 0.027$ | $0.998 \pm 0.027$ | $1.001 \pm 0.027$ | $1.003 \pm 0.027$ |
| | TV | $\Delta = 1.0$ | $0.974 \pm 0.011$ | $0.985 \pm 0.011$ | $0.991 \pm 0.011$ | $0.994 \pm 0.011$ | $0.996 \pm 0.011$ |
| | TV | $\Delta = 2.0$ | $0.977 \pm 0.002$ | $0.989 \pm 0.002$ | $0.996 \pm 0.002$ | $0.999 \pm 0.003$ | $1.001 \pm 0.003$ |
| Scale change | KL | $\sigma = 1.2$ | $0.743 \pm 0.063$ | $0.841 \pm 0.070$ | $0.908 \pm 0.072$ | $0.954 \pm 0.072$ | $0.991 \pm 0.072$ |
| | KL | $\sigma = 1.5$ | $0.779 \pm 0.027$ | $0.872 \pm 0.029$ | $0.927 \pm 0.030$ | $0.958 \pm 0.031$ | $0.977 \pm 0.031$ |
| | KL | $\sigma = 2.0$ | $0.803 \pm 0.018$ | $0.898 \pm 0.020$ | $0.953 \pm 0.021$ | $0.982 \pm 0.022$ | $0.998 \pm 0.022$ |
| | Hell$^2$ | $\sigma = 1.2$ | $0.741 \pm 0.077$ | $0.853 \pm 0.089$ | $0.931 \pm 0.098$ | $0.986 \pm 0.106$ | $1.029 \pm 0.111$ |
| | Hell$^2$ | $\sigma = 1.5$ | $0.735 \pm 0.035$ | $0.842 \pm 0.039$ | $0.908 \pm 0.041$ | $0.948 \pm 0.042$ | $0.973 \pm 0.042$ |
| | Hell$^2$ | $\sigma = 2.0$ | $0.744 \pm 0.014$ | $0.858 \pm 0.014$ | $0.926 \pm 0.014$ | $0.965 \pm 0.013$ | $0.987 \pm 0.012$ |
| | TV | $\sigma = 1.2$ | $0.934 \pm 0.033$ | $0.970 \pm 0.036$ | $0.990 \pm 0.039$ | $1.001 \pm 0.040$ | $1.008 \pm 0.041$ |
| | TV | $\sigma = 1.5$ | $0.907 \pm 0.014$ | $0.948 \pm 0.015$ | $0.970 \pm 0.017$ | $0.982 \pm 0.018$ | $0.989 \pm 0.018$ |
| | TV | $\sigma = 2.0$ | $0.898 \pm 0.009$ | $0.947 \pm 0.010$ | $0.974 \pm 0.010$ | $0.988 \pm 0.010$ | $0.995 \pm 0.010$ |
| Multimodal | JS | $\Delta = 0.5$ | $0.746 \pm 0.157$ | $0.849 \pm 0.176$ | $0.926 \pm 0.189$ | $0.994 \pm 0.196$ | $1.068 \pm 0.199$ |
| | JS | $\Delta = 1.0$ | $0.769 \pm 0.038$ | $0.849 \pm 0.040$ | $0.898 \pm 0.041$ | $0.929 \pm 0.041$ | $0.948 \pm 0.042$ |
| | JS | $\Delta = 2.0$ | $0.846 \pm 0.019$ | $0.912 \pm 0.020$ | $0.950 \pm 0.021$ | $0.972 \pm 0.021$ | $0.985 \pm 0.021$ |
| | KL | $\Delta = 0.5$ | $0.742 \pm 0.125$ | $0.853 \pm 0.147$ | $0.936 \pm 0.165$ | $1.000 \pm 0.178$ | $1.054 \pm 0.187$ |
| | KL | $\Delta = 1.0$ | $0.766 \pm 0.032$ | $0.864 \pm 0.036$ | $0.930 \pm 0.039$ | $0.971 \pm 0.041$ | $0.998 \pm 0.041$ |
| | KL | $\Delta = 2.0$ | $0.669 \pm 0.009$ | $0.765 \pm 0.010$ | $0.837 \pm 0.012$ | $0.889 \pm 0.013$ | $0.926 \pm 0.013$ |
| | TV | $\Delta = 0.5$ | $0.935 \pm 0.066$ | $0.969 \pm 0.070$ | $0.990 \pm 0.074$ | $1.005 \pm 0.076$ | $1.017 \pm 0.078$ |
| | TV | $\Delta = 1.0$ | $0.947 \pm 0.023$ | $0.975 \pm 0.023$ | $0.990 \pm 0.023$ | $0.998 \pm 0.023$ | $1.003 \pm 0.024$ |
| | TV | $\Delta = 2.0$ | $0.942 \pm 0.005$ | $0.969 \pm 0.005$ | $0.982 \pm 0.005$ | $0.989 \pm 0.006$ | $0.993 \pm 0.006$ |
| Heavy tails | JS | – | $0.488 \pm 0.028$ | $0.651 \pm 0.036$ | $0.778 \pm 0.041$ | $0.869 \pm 0.044$ | $0.933 \pm 0.046$ |
| | KL | – | $0.210 \pm 0.012$ | $0.299 \pm 0.017$ | $0.383 \pm 0.022$ | $0.458 \pm 0.025$ | $0.524 \pm 0.028$ |
| | TV | – | $0.824 \pm 0.014$ | $0.907 \pm 0.017$ | $0.955 \pm 0.020$ | $0.981 \pm 0.023$ | $0.996 \pm 0.025$ |

*Table 3.* 1D divergence estimation benchmarks (10 runs). We report the ratio estimate/reference (mean $\pm$ std) for various $K$ values.

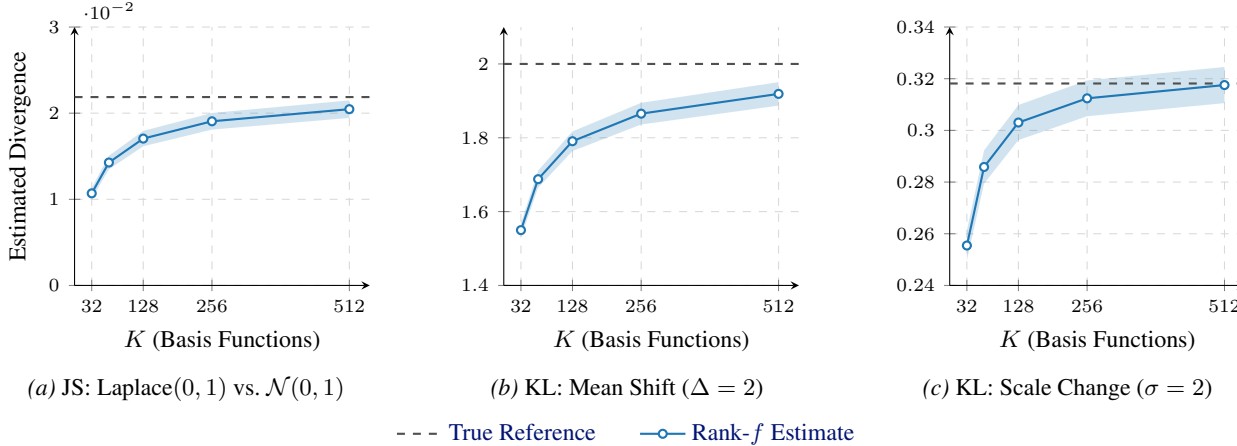

*(a)* JS: Laplace$(0, 1)$ vs. $\mathcal{N}(0, 1)$  *(b)* KL: Mean Shift $(\Delta = 2)$  *(c)* KL: Scale Change $(\sigma = 2)$

- - - True Reference  ⊸ Rank-$f$ Estimate

*Figure 7.* Convergence of the Rank-$f$ estimator as the number of basis functions $K$ increases. The estimator (blue) consistently converges to the true analytic or Monte Carlo reference (dashed gray) across different divergence types and scenarios.

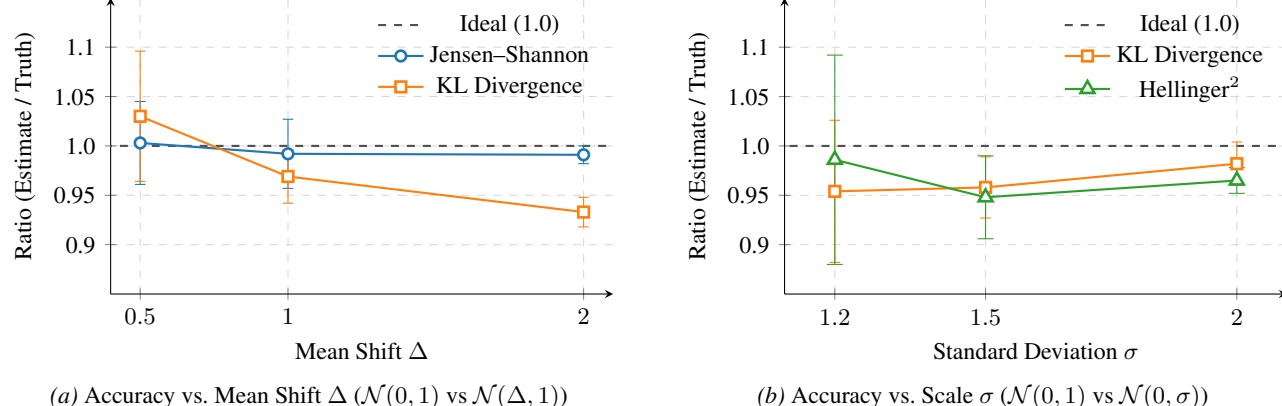

*(a) Accuracy vs. Mean Shift $\Delta$ ($\mathcal{N}(0,1)$ vs $\mathcal{N}(\Delta,1)$)*    *(b) Accuracy vs. Scale $\sigma$ ($\mathcal{N}(0,1)$ vs $\mathcal{N}(0,\sigma)$)*

*Figure 8.* Robustness of the Rank-$f$ estimator at fixed $K = 256$. The plots show the ratio of the estimated divergence to the ground truth (closer to 1.0 is better) as the shift ($\Delta$) or scale ($\sigma$) increases.

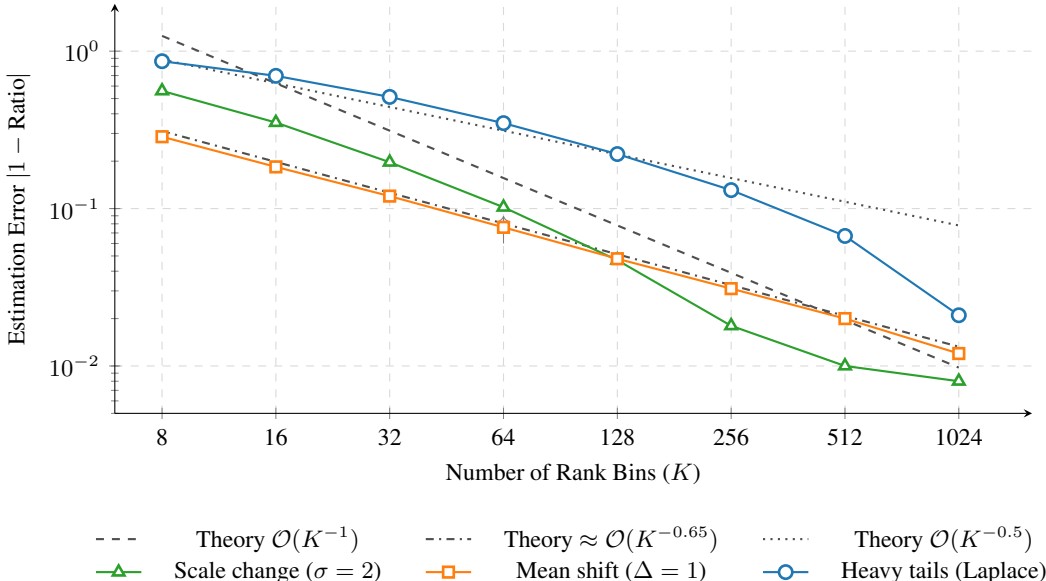

*Figure 9.* Convergence rate analysis across different tail behaviors. We compare the empirical error against theoretical slopes. The Scale Change (Green) is bounded and achieves the optimal $\mathcal{O}(K^{-1})$ rate. The Mean Shift (Orange) suffers from an unbounded density ratio at the tail, degrading convergence to $\approx \mathcal{O}(K^{-0.65})$. The Heavy Tail (Blue) case is the most difficult, bounded by the Hölder continuity limit of $\mathcal{O}(K^{-0.5})$.

**Scaling and reported ratios.** Alongside the sliced estimate $D_{f,\nu}^{(K)}(\mu)$, we also report the simple normalization $d \times D_{f,\nu}^{(K)}(\mu)$ and summarize accuracy via the ratio $(d \times \text{sliced})/\text{true}$. This scaling is not intended to be exact in general; it is a lightweight calibration that keeps ratios on a comparable scale across dimensions.

**Reference ("ground-truth") divergences.** Reference divergences are computed as follows, depending on whether closed forms are available:

- **Gaussian-Gaussian (analytic references).** For $\mu = \mathcal{N}(\mu_0, \Sigma_0)$ and $\nu = \mathcal{N}(\mu_1, \Sigma_1)$, the reference $\text{KL}(\mu\|\nu)$ and squared Hellinger $\text{H}^2(\mu, \nu)$ are evaluated in closed form. This avoids an additional numerical approximation layer, so discrepancies can be attributed to the rank-statistic estimator rather than to the reference computation.

- **Gaussian Jensen-Shannon (deterministic proxy).** The multivariate Jensen-Shannon divergence between two Gaussians does not admit a simple closed form because the mixture $\frac{1}{2}\mu + \frac{1}{2}\nu$ is not Gaussian. To keep the reference deterministic (and avoid injecting extra Monte Carlo variance), we approximate the mixture by a single Gaussian with matched mean and covariance (moment matching), and define the reference as $\frac{1}{2}\text{KL}(\mu\|M) + \frac{1}{2}\text{KL}(\nu\|M)$ for that matched Gaussian $M$.

- **Non-Gaussian pairs (Monte Carlo from known log-densities).** When at least one distribution is non-Gaussian (e.g., Laplace vs. Gaussian, Student-$t$ vs. Gaussian, or a Gaussian mixture vs. Gaussian), a closed-form multivariate reference is typically unavailable. In these cases, the reference divergence is computed by Monte Carlo from its expectation form using the *known* log-densities (e.g., $\text{KL}(\mu\|\nu) = \mathbb{E}_\mu[\log p_\mu(X) - \log p_\nu(X)]$, and similarly for JS via expectations under $\mu$ and $\nu$). All evaluations are performed in the log domain using log-sum-exp to ensure numerical stability.

Figure 3 and Table 4 provide complementary views of the same phenomenon. Figure 3 isolates the Gaussian mean-shift setting and shows that the normalized quantity $(d \times \text{sliced})/\text{true}$ remains close to the ideal value 1 across dimensions, with moderate, dimension-dependent deviations that are consistent with a mismatch between the sliced functional and the full multivariate divergence (and with the crudeness of the $d \times (\cdot)$ calibration). Table 4 summarizes this behavior across a broader set of distribution pairs: for Gaussian-Gaussian benchmarks (KL and Hellinger$^2$) the ratios typically stay near 1, while JS experiments that rely on the Gaussian-proxy reference and non-Gaussian misspecification cases exhibit larger and more variable departures, especially as $d$ grows, highlighting that the sliced estimator is best interpreted as a stable, sample-based surrogate whose absolute scale can drift from the multivariate reference in challenging regimes. Finally, Figure 10 illustrates how increasing the rank resolution $K$ systematically reduces the finite-$K$ approximation gap in representative cases, with ratios approaching the ideal baseline as $K$ increases.

| Setting | $f$-divergence | Parameter | Dimension $d$ | | | | |
| --- | --- | --- | --- | --- | --- | --- | --- |
| | | | 2 | 5 | 10 | 20 | 50 |
| Mean Shift | KL | $\Delta = 0.5$ | $1.015 \pm 0.067$ | $1.098 \pm 0.052$ | $1.188 \pm 0.072$ | $0.977 \pm 0.060$ | $1.304 \pm 0.044$ |
| | KL | $\Delta = 1.0$ | $0.991 \pm 0.030$ | $1.087 \pm 0.032$ | $1.170 \pm 0.031$ | $0.899 \pm 0.028$ | $1.113 \pm 0.042$ |
| | Hellinger$^2$ | $\Delta = 0.5$ | $1.005 \pm 0.062$ | $0.935 \pm 0.048$ | $0.973 \pm 0.049$ | $1.002 \pm 0.037$ | $1.035 \pm 0.060$ |
| | Hellinger$^2$ | $\Delta = 1.0$ | $0.972 \pm 0.033$ | $0.931 \pm 0.022$ | $0.965 \pm 0.048$ | $0.943 \pm 0.042$ | $0.851 \pm 0.021$ |
| | JS (Gaussian) | $\Delta = 0.5$ | $1.007 \pm 0.076$ | $0.899 \pm 0.040$ | $0.887 \pm 0.031$ | $1.006 \pm 0.060$ | $1.234 \pm 0.038$ |
| | JS (Gaussian) | $\Delta = 1.0$ | $0.979 \pm 0.043$ | $0.892 \pm 0.013$ | $0.895 \pm 0.035$ | $0.951 \pm 0.041$ | $1.144 \pm 0.031$ |
| Scale and Covariance | JS (Scale) | $\sigma = 1.2$ | $0.891 \pm 0.052$ | $0.936 \pm 0.048$ | $0.913 \pm 0.030$ | $0.851 \pm 0.016$ | $0.846 \pm 0.016$ |
| | JS (Scale) | $\sigma = 1.5$ | $0.856 \pm 0.027$ | $0.856 \pm 0.015$ | $0.858 \pm 0.013$ | $0.797 \pm 0.007$ | $0.803 \pm 0.006$ |
| | JS (Scale) | $\sigma = 2.0$ | $0.776 \pm 0.014$ | $0.775 \pm 0.010$ | $0.785 \pm 0.005$ | $0.734 \pm 0.005$ | $0.734 \pm 0.003$ |
| | JS (Anisotropic) | — | $0.837 \pm 0.037$ | $0.761 \pm 0.040$ | $0.791 \pm 0.032$ | $0.726 \pm 0.015$ | $0.736 \pm 0.011$ |
| Model Misspecification | JS (Laplace vs. Gaussian) | — | $0.767 \pm 0.015$ | $0.861 \pm 0.016$ | $1.052 \pm 0.017$ | $1.227 \pm 0.015$ | $1.968 \pm 0.014$ |
| | KL ($t$-dist vs. Gaussian) | $df = 3$ | $0.164 \pm 0.023$ | $0.218 \pm 0.017$ | $0.267 \pm 0.015$ | $0.268 \pm 0.010$ | $0.318 \pm 0.012$ |
| | JS (GMM vs. Gaussian) | $\Delta = 1.0$ | $0.923 \pm 0.006$ | $0.759 \pm 0.009$ | $0.753 \pm 0.007$ | $1.012 \pm 0.009$ | $2.247 \pm 0.016$ |

*Table 4.* Ratio summary across dimensions ($d$). Values report mean $\pm$ standard deviation over 10 runs.

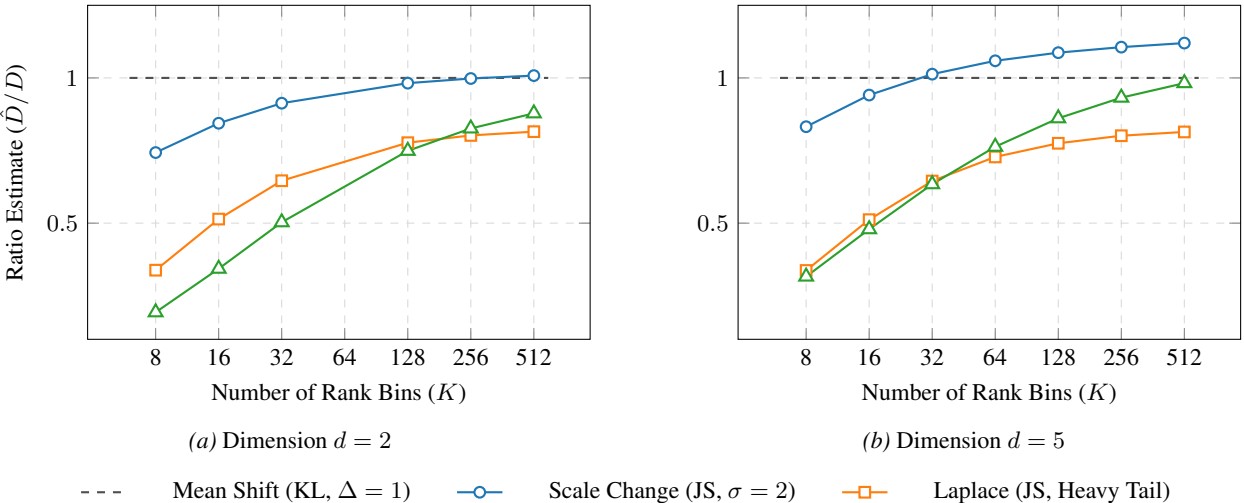

*(a)* Dimension $d = 2$                                               *(b)* Dimension $d = 5$

- - -     Mean Shift (KL, $\Delta = 1$)        Scale Change (JS, $\sigma = 2$)        Laplace (JS, Heavy Tail)

*Figure 10.* Convergence of ratio estimates ($\hat{D}/D$) versus number of rank bins ($K$) for dimensions $d = 2$ (left) and $d = 5$ (right).

## C.4. Generative transport dynamics for rank-statistic $f$-divergences

We next provide pseudocode for the sliced rank-proximal transport update used in Section 4.4.

---

**Algorithm 1** Sliced rank-proximal transport (one outer step)

---

**Input:** particles $(x_i)_{i=1}^N \subset \mathbb{R}^d$, reference samples $(y_j)_{j=1}^M \subset \mathbb{R}^d$, slices $L$, rank order $K$, temperature $\tau$, trust $\eta$, step size $\varepsilon$, $f$-generator $f(\cdot)$.
**Output:** updated particles $(x_i)_{i=1}^N$.
Draw directions $s_1, \ldots, s_L \in \mathbb{S}^{d-1}$ (optionally include antithetic pairs $\pm s_\ell$)
Initialize $\Delta x_i \leftarrow 0 \in \mathbb{R}^d$ for all $i = 1, \ldots, N$
**for** $\ell = 1$ **to** $L$ **do**
    Project: $x_i^{(\ell)} \leftarrow \langle x_i, s_\ell \rangle, y_j^{(\ell)} \leftarrow \langle y_j, s_\ell \rangle$
    Soft ranks: $v_{0,i}^{(\ell)} \leftarrow \widehat{R}_{\nu^{(\ell)},\tau}(x_i^{(\ell)}) \in (0,1)$ for $i = 1, \ldots, N$; write $\mathbf{v}_0^{(\ell)} = (v_{0,1}^{(\ell)}, \ldots, v_{0,N}^{(\ell)})$
    Prox in rank space (approx. by SGD/ULA/MALA):

$$\mathbf{v}_1^{(\ell)} \approx \underset{\mathbf{v} \in (0,1)^N}{\arg\min} \left\{ D_f(\widehat{Q}^{(K)}(\mathbf{v}) \| U_K) + \frac{1}{2\eta} \|\mathbf{v} - \mathbf{v}_0^{(\ell)}\|_2^2 \right\}$$

    Quantile match: $z_i^{(\ell)} \leftarrow Q_{\widehat{\nu}^{(\ell)}}(v_{1,i}^{(\ell)})$, $\delta_i^{(\ell)} \leftarrow z_i^{(\ell)} - x_i^{(\ell)}$
    Accumulate: $\Delta x_i \leftarrow \Delta x_i + \delta_i^{(\ell)} s_\ell$
**end for**
Update: $x_i \leftarrow x_i + \varepsilon \frac{d}{L} \Delta x_i$    for all $i$
**return** $(x_i)_{i=1}^N$

---

## C.5. Generative experiments on image datasets

In this appendix, we provide implementation details for the image-generation experiments. Section C.5.1 gives the full experimental setup for the CO-RPT experiment discussed in Section 4.4.2. Section C.5.2 reports an additional feature-space rank-TV training experiment on CIFAR-10, which is included as supplementary evidence that rank-based objectives can provide a useful training signal for neural generators.

### C.5.1. CO-RPT ON CIFAR-10

**Center-outward rank-proximal transport (CO-RPT)**    Algorithm 2 implements a slice-free variant of rank-proximal transport based on a center-outward decomposition. Starting from particles $X = \{x_i\}_{i=1}^N$ and reference samples $Y = \{y_j\}_{j=1}^M$, we first recenter the configuration using the target mean $\bar{y} = \frac{1}{M} \sum_{j=1}^M y_j$, i.e., $\tilde{x}_i = x_i - \bar{y}$ and $\tilde{y}_j = y_j - \bar{y}$. Optionally, we apply a whitening transform (e.g. ZCA fitted on $\tilde{Y}$) so that the target is approximately isotropic; this reduces anisotropy and makes the radial/angular decomposition more stable, and the inverse transform is applied at the end.

We then decompose each point into its radius and direction: $r_i^x = \|\tilde{x}_i\|$, $u_i^x = \tilde{x}_i / r_i^x$ and $r_j^y = \|\tilde{y}_j\|$, $u_j^y = \tilde{y}_j / r_j^y$. The transport step is built by updating radii through a one-dimensional rank-proximal refinement, and updating directions through a simple matching on the unit sphere. Concretely, we compute soft radial ranks $v_{0,i} \approx \widehat{R}_{r^y,\tau}(r_i^x) \in [0,1]$ using a smoothed empirical CDF of the target radii, where the temperature $\tau$ controls how sharp the rank assignment is. We refine these ranks by approximately solving the proximal objective

$$\mathbf{v}_1 \approx \underset{\mathbf{v} \in (0,1)^N}{\arg\min} \left\{ D_f(\widehat{Q}^{(K)}(\mathbf{v}) \| U_K) + \frac{1}{2\eta} \|\mathbf{v} - \mathbf{v}_0\|_2^2 \right\},$$

where $\widehat{Q}^{(K)}(\mathbf{v})$ is the Bernstein-smoothed histogram over $[K]$ induced by $\mathbf{v}$, $D_f(\cdot \| U_K)$ measures deviation from uniformity, and $\eta > 0$ acts as a trust region that prevents overly aggressive rank updates. In practice, we use only a few inner iterations (e.g., SGD or a simple extragradient update). The refined ranks are mapped back to the radial axis via the empirical quantile of the target radii, $r_i^\star = Q_{\widehat{r}^y}(v_{1,i})$, ensuring that uniform ranks would reproduce the target radial distribution.

To align the angular structure, we match each particle direction $u_i^x$ to a nearby target direction using cosine similarity, i.e. $j^\star(i) \in \arg\max_j \langle u_i^x, u_j^y \rangle$, and blend toward it with weight $\beta \in [0,1]$: $u_i^\star = \text{normalize}((1-\beta)u_i^x + \beta u_{j^\star(i)}^y)$. Setting

$\beta = 0$ yields a purely radial update, while larger $\beta$ accelerates angular adaptation. Combining the transported radius and direction gives the center-outward proposal $\tilde{x}_i^\star = r_i^\star u_i^\star$, and we take an outer step $\tilde{x}_i^+ = \tilde{x}_i + \varepsilon(\tilde{x}_i^\star - \tilde{x}_i)$ with step size $\varepsilon > 0$. Optionally, we clip the increment to a maximum norm $c$ to avoid rare large jumps. Finally, we undo the optional whitening and add back the center $\bar{y}$ to obtain the updated particles $X^+ = \{x_i^+\}_{i=1}^N$.

Overall, CO-RPT replaces the multi-slice quantile matching of (10) by a single univariate rank-prox update on radii together with a lightweight spherical matching for directions. This yields a geometrically interpretable, fully sample-based update that explicitly controls the radial marginal through the rank objective, while encouraging directional alignment through nearest-neighbor coupling on $\mathbb{S}^{d-1}$.

---

**Algorithm 2** Center-outward rank-proximal transport (CO-RPT) update

---

**Input:** particles $X = \{x_i\}_{i=1}^N$, targets $Y = \{y_j\}_{j=1}^M$, $K, f, \tau, \eta$, step $\varepsilon$, angular blend $\beta$ (optional cap $c$).
**Output:** updated particles $X^+$
(Optional) center/whiten: $\tilde{x}_i, \tilde{y}_j$.
Compute radii/directions: $r_i^x = \|\tilde{x}_i\|$, $u_i^x = \tilde{x}_i/r_i^x$ and $r_j^y = \|\tilde{y}_j\|$, $u_j^y = \tilde{y}_j/r_j^y$.
Soft radial ranks: $v_{0,i} \approx \widehat{R}_{r^y,\tau}(r_i^x) \in (0,1)$ for $i = 1, \ldots, N$; write $\mathbf{v}_0 = (v_{0,1}, \ldots, v_{0,N})$.
Prox in rank space: $\mathbf{v}_1 \approx \arg\min_{\mathbf{v} \in (0,1)^N} \left\{ D_f(\widehat{Q}^{(K)}(\mathbf{v}) \| U_K) + \frac{1}{2\eta} \|\mathbf{v} - \mathbf{v}_0\|_2^2 \right\}$.
Target radii: $r_i^\star \leftarrow Q_{\widehat{r}^y}(v_{1,i})$ for $i = 1, \ldots, N$.
Angular match: $j^\star(i) \in \arg\max_j \langle u_i^x, u_j^y \rangle$, $u_i^\star \leftarrow \text{normalize}\big((1 - \beta)u_i^x + \beta u_{j^\star(i)}^y\big)$.
Proposal: $\tilde{x}_i^\star \leftarrow r_i^\star u_i^\star$, $\Delta_i \leftarrow \tilde{x}_i^\star - \tilde{x}_i$ (optional: $\Delta_i \leftarrow \Delta_i \cdot \min\{1, c/\|\Delta_i\|\}$).
Update: $\tilde{x}_i^+ \leftarrow \tilde{x}_i + \varepsilon \, \Delta_i$ and uncenter/unwhiten to get $x_i^+$.
**return** $X^+ = \{x_i^+\}_{i=1}^N$

---

**Experimental setup and evaluation.** We ran the proposed center-outward rank-proximal transport (CO-RPT) directly in pixel space on CIFAR-10. We randomly selected $M = 1000$ images from the CIFAR-10 training set, upsampled them to $64 \times 64$ using bicubic interpolation with antialiasing, and mapped pixel intensities to $[0, 1]$; each image was then flattened to $\mathbb{R}^{3HW}$ with $H = W = 64$. We used the Jensen–Shannon generator ($f = \text{JS}$) with trust-region parameter $\eta = 0.5$ and 3 inner extragradient steps per outer iteration. We initialized $N = M$ particles from a Gaussian matched to the mean and marginal standard deviation of the whitened target features, and iterated CO-RPT for $T = 20000$ outer steps. We linearly annealed the rank resolution from $K = 64$ to $K = 160$, the rank-smoothing temperature from $\tau = 0.30$ to $\tau = 0.07$, and the outer step size from $\varepsilon = 0.16$ to $\varepsilon = 0.10$, while clipping per-particle updates to a maximum norm of $c = 0.30$. We also used a small angular blending parameter $\beta_{\text{angle}} = 0.01$.

### C.5.2. FEATURE-SPACE RANK-TV TRAINING ON CIFAR-10

We also evaluate the rank-based total variation objective as a feature-space training signal on CIFAR-10. Since CIFAR-10 images are natively $32 \times 32$, we resize them to $64 \times 64$ using bicubic interpolation with antialiasing and normalize pixel values to $[-1, 1]$. We use a DCGAN-style generator and a spectrally normalized convolutional discriminator. The discriminator is trained with the standard hinge loss, while the generator is trained to match real and generated samples in the feature space of the discriminator using the sliced rank-TV objective. We also consider a hybrid variant with a small adversarial generator loss. An exponential moving average of the generator weights is used for sampling and evaluation.

The goal of this experiment is not to obtain state-of-the-art CIFAR-10 generation results, but to test whether the proposed rank-based discrepancy provides a usable training signal in a nontrivial image setting. We observe a steady improvement in FID during training: in our longest run, the FID decreases from roughly 380 at early iterations to about 41 after 245,000 generator updates. This suggests that the feature-space rank-TV objective can progressively improve image quality and distributional matching.

The method can be applied after any fixed encoder, such as Inception, VGG, DINOv2, or a task-specific representation, yielding a rank divergence between feature pushforwards. Hence, the discrepancy is representation-dependent, as with FID or precision–recall metrics in learned feature spaces. Unlike FID, the rank statistic does not impose a Gaussian approximation in feature space; however, slicing may still miss informative directions when only finitely many projections are used.

## C.6. Addressing mode collapse on MNIST

Building on the pretraining approach of de Frutos et al. (2024b), who employ a rank-based total-variation divergence to reduce mode collapse in GANs, we study whether the same strategy carries over to a wider class of rank-statistic $f$-divergences. Concretely, we incorporate the sliced rank-statistic $f$-divergences objective as a pretraining signal for the DCGAN generator (Radford et al., 2015) and evaluate the resulting pipeline on MNIST. To measure both realism and coverage, we follow Sajjadi et al. (2018) and report precision (as a proxy for fidelity) and recall (as a proxy for diversity). All models are trained for 40 epochs with batch size 128. For the pretrained variants, we first run 20 epochs under the sliced rank-statistic $f$-divergences objective and then continue with 40 additional epochs of standard DCGAN training.

In Table 5, we compare *rank-statistic $f$-divergence* pretraining variants (TV, KL, JS, and $\text{Hell}^2$) with standard DCGAN training, *rank-statistic $f$-divergence*+DCGAN fine-tuning, and stronger multi-discriminator baselines (Durugkar et al., 2016; Choi & Han, 2022). Focusing on precision and recall, the standalone *rank-statistic $f$-divergence* models already achieve strong recall on MNIST: JS and $\text{Hell}^2$ are the most recall-oriented, reaching 97.0% and 98.7% recall for $m=50$, respectively, while TV and KL yield a more balanced behavior (e.g., 95.0% and 91.1% recall for $m=50$). When we pretrain with a *rank-statistic $f$-divergence* and then fine-tune with the adversarial loss, precision increases substantially relative to vanilla DCGAN: TV+DCGAN and KL+DCGAN reach 95.0% and 96.2% precision, respectively (vs. 93.85% for DCGAN), while maintaining competitive recall (around 92.8% and 90.5%). Overall, these results highlight a clear trade-off between the choice of $f$ (more recall-oriented for JS/$\text{Hell}^2$) and the benefit of adversarial fine-tuning for improving precision without collapsing recall.

Our estimator targets scalar $f$-divergences rather than precision–recall curves. However, precision–recall divergences form a family of $f$-divergence-type trade-off quantities (Simon et al., 2019; Siry et al., 2023; Vérine et al., 2023); choosing a corresponding generator yields a rank-histogram approximation of the associated scalar PR divergence, while varying the PR parameter would produce a discretized PR trade-off curve. This differs from (Simon et al., 2019), who estimate the PR curve directly.

| Dataset | Method | F-score | | P&R | |
|---------|--------|---------|---------|-----|-----|
| | | $F_{1/8}$ ↑ | $F_8$ ↑ | Precision ↑ | Recall ↑ |
| **MNIST** | TV (m=20) | 88.09 ± 0.32 | 93.91 ± 0.72 | 88.01 ± 0.52 | 94.25 ± 0.91 |
| | TV (m=50) | 88.89 ± 0.31 | 94.90 ± 0.71 | 88.80 ± 0.50 | 95.08 ± 0.94 |
| | KL (m=20) | 90.50 ± 0.43 | 90.21 ± 0.62 | 90.50 ± 0.47 | 90.18 ± 0.91 |
| | KL (m=50) | 91.59 ± 0.46 | 91.11 ± 0.68 | 91.62 ± 0.47 | 91.11 ± 0.88 |
| | JS (m=20) | 86.64 ± 0.56 | 96.13 ± 0.76 | 86.48 ± 0.40 | 96.32 ± 0.92 |
| | JS (m=50) | 87.73 ± 0.32 | 96.84 ± 0.75 | 87.60 ± 0.49 | 97.09 ± 0.91 |
| | $\text{Hell}^2$ (m=20) | 83.00 ± 0.35 | 97.92 ± 0.74 | 82.81 ± 0.50 | 98.23 ± 0.87 |
| | $\text{Hell}^2$ (m=50) | 83.83 ± 0.35 | 98.43 ± 0.69 | 83.62 ± 0.50 | 98.71 ± 0.90 |
| | DCGAN | 93.54 ± 0.64 | 75.66 ± 1.46 | 93.85 ± 1.45 | 75.43 ± 2.56 |
| | TV + DCGAN | 94.97 ± 0.35 | 92.83 ± 0.72 | 95.00 ± 0.53 | 92.80 ± 0.85 |
| | KL + DCGAN | 96.11 ± 0.36 | 90.58 ± 0.70 | 96.20 ± 0.46 | 90.50 ± 0.83 |
| | JS + DCGAN | 94.71 ± 0.42 | 95.19 ± 0.73 | 94.67 ± 0.42 | 95.21 ± 0.87 |
| | $\text{Hell}^2$ + DCGAN | 93.84 ± 0.35 | 96.75 ± 0.75 | 93.82 ± 0.45 | 96.81 ± 0.85 |
| | GMAN | 97.78 ± 0.40 | 96.52 ± 0.57 | 97.80 ± 0.71 | 96.53 ± 0.89 |
| | **MCL-GAN** | **98.20 ± 0.19** | **98.00 ± 0.25** | **98.20 ± 0.30** | **98.00 ± 0.40** |

*Table 5.* Quantitative results on MNIST ($28 \times 28$), reporting $F_{1/8}$ and $F_8$ (the $\beta$-weighted harmonic means of precision and recall), precision, and recall (mean±std, %). Results are grouped by divergence type for enhanced scannability.

## C.7. Two-sample testing with rank chi-square statistics

We include a small two-sample testing experiment to illustrate that the proposed rank-statistic construction also yields a natural nonparametric test. Given two independent samples

$$X_1, \ldots, X_N \sim \mu, \qquad Y_1, \ldots, Y_M \sim \nu,$$

we test

$$H_0 : \mu = \nu \qquad \text{against} \qquad H_1 : \mu \neq \nu.$$

For a fixed rank resolution $K$, we compute the rank histogram of the $X$-sample with respect to the $Y$-sample and compare it to the discrete uniform distribution on $\{0, \ldots, K\}$. More precisely, denoting by $\widehat{Q}_{N,M}^{(K)}$ the empirical rank histogram, we use the Pearson/Cressie-Read-type statistic (Cressie & Read, 1984)

$$T_K = \sum_{k=0}^{K} \frac{\left( \widehat{Q}_{N,M}^{(K)}(k) - \frac{1}{K+1} \right)^2}{\frac{1}{K+1}}.$$

Equivalently, up to a deterministic multiplicative factor, this is the empirical rank-statistic $f$-divergence associated with the chi-square generator

$$f_{\chi^2}(t) = \frac{1}{2}(t-1)^2.$$

Thus, in the univariate case, the rank chi-square test is exactly the proposed rank-statistic divergence estimator specialized to the $\chi^2$ entropy.

For higher-dimensional data, we use the corresponding sliced version based on random projections. We draw $L$ random projection directions $s_1, \ldots, s_L \in \mathbb{S}^{d-1}$ and compute the univariate rank chi-square statistic on each projected pair

$$\{s_\ell^\top X_i\}_{i=1}^N, \qquad \{s_\ell^\top Y_j\}_{j=1}^M.$$

The final statistic is obtained by averaging the projected discrepancies,

$$T_{K,L} = \frac{1}{L} \sum_{\ell=1}^{L} T_K^{(\ell)}.$$

When $d = 1$, there is only one possible projection, and the random-subspace statistic reduces to the univariate rank-statistic chi-square divergence described above.

Since the null distribution of the statistic may depend on the sample sizes and on the finite-sample rank construction, we calibrate the test by permutation. Specifically, we repeatedly permute the pooled sample

$$\{X_1, \ldots, X_N, Y_1, \ldots, Y_M\},$$

split it into two groups of sizes $N$ and $M$, recompute the statistic, and use the resulting empirical null distribution to obtain a $p$-value. We reject $H_0$ at level $\alpha = 0.05$ whenever this permutation $p$-value is smaller than $\alpha$.

We evaluate the resulting test on a collection of controlled synthetic two-sample benchmarks designed to probe different types of distributional mismatch. The benchmark families include Gaussian location shifts, Gaussian scale changes, Laplace and Student-$t$ shape changes, Gaussian mixture alternatives, and dependence alternatives generated from $t$-, Clayton-, and Gumbel-copula models. These settings cover mean, scale, tail, multimodal, and dependence changes. We compare the sliced rank $\chi^2$ test against standard two-sample baselines, namely Hotelling's $T^2$, sliced optimal transport, MMD, and a tuned MMD variant.

We first assess calibration under the null. Figure 11 reports the empirical rejection probability at nominal level $\alpha = 0.05$ across dimensions $d \in \{2, 4, 10\}$. The results show that the permutation-calibrated sliced rank $\chi^2$ statistic remains close to the target type-I error across the considered families, with deviations compatible with the Monte Carlo variability of the experiment.

We then evaluate power under controlled alternatives. Figures 12 and 13 report empirical power as a function of average runtime per repetition in dimension $d = 4$. The sliced rank $\chi^2$ statistic provides a competitive efficiency–power tradeoff on

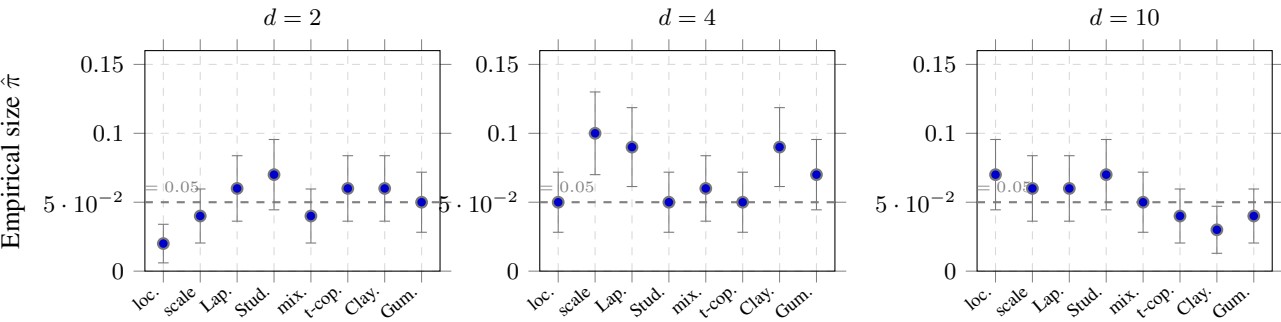

*Figure 11.* Calibration of the rank-statistic $\chi^2$-divergence two-sample test under the null. The statistic corresponds to the sliced rank $f$-divergence with the chi-square generator $f_{\chi^2}(t) = \frac{1}{2}(t-1)^2$, calibrated by permutation. We report empirical rejection probabilities at nominal level $\alpha = 0.05$ across distributional families and dimensions. Error bars denote $\hat{\pi} \pm \mathrm{SE}$ over $R = 100$ repetitions. All experiments use $N = M = 2000$, $L = 64$, $K = 4$, and $B = 500$ permutations. The dashed horizontal line indicates the nominal level.

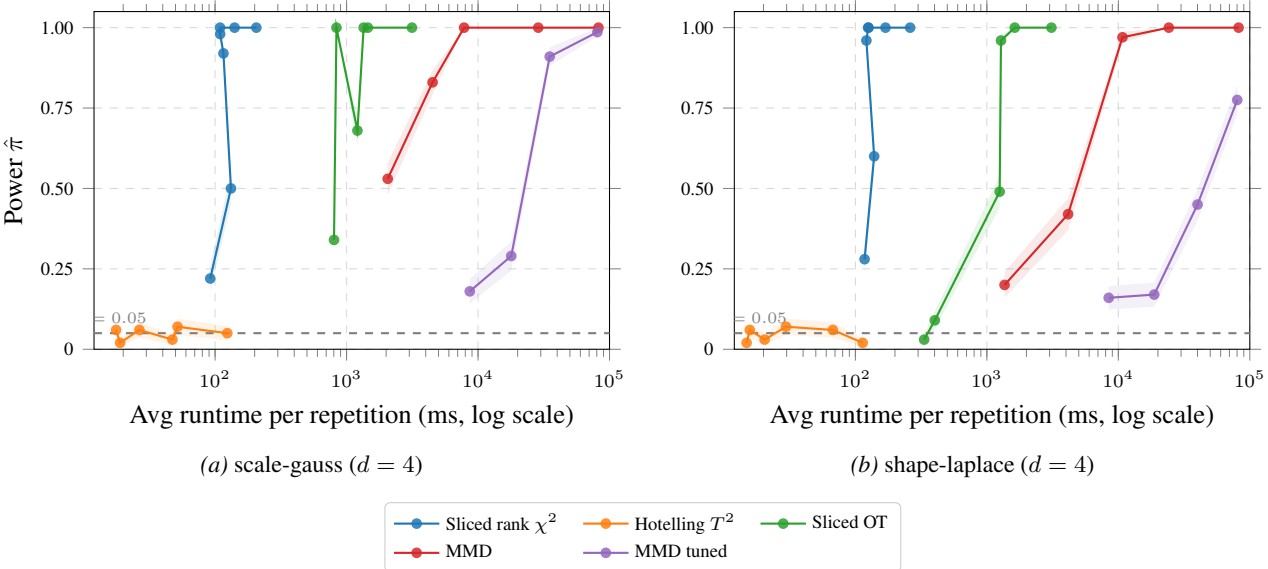

*Figure 12.* Efficiency–power tradeoff for scale and shape alternatives in dimension $d = 4$. We compare the sliced rank-statistic $\chi^2$-divergence test against Hotelling's $T^2$, sliced optimal transport, MMD, and tuned MMD. Curves show empirical power as a function of average runtime per repetition, and shaded regions denote $\hat{\pi} \pm \mathrm{SE}$. All tests are calibrated at nominal level $\alpha = 0.05$.

scale and shape alternatives, reaching high power at substantially lower runtime than the more computationally expensive MMD and sliced optimal-transport baselines in these settings. For dependence alternatives, its behavior is more problem-dependent: it performs strongly on the Gumbel-copula alternative, while the Clayton-copula alternative is less favorable for this particular random-projection configuration.

Overall, these experiments indicate that the permutation-calibrated rank-statistic construction behaves as a calibrated two-sample test in the considered settings, and that, in the one-dimensional case, the rank chi-square subspace statistic coincides with our proposed estimator for the chi-square $f$-divergence.

### C.8. CIFAR-10-C benchmark

We also evaluate the same sliced rank $f$-divergence statistic on a high-dimensional image benchmark. We use clean CIFAR-10 test images as the reference distribution $\nu$, and CIFAR-10-C (Hendrycks & Dietterich, 2019) images as corrupted alternatives $\mu_{c,s}$, indexed by corruption type $c$ and severity level $s$. Images are flattened into pixel-space vectors in $[0,1]^{3072}$.

For each trial, we draw samples from $\nu$ and $\mu_{c,s}$, sample $L$ random one-dimensional projection directions, and compute the univariate rank $f$-divergence on each projected pair. As in the synthetic experiments, we use the chi-square generator

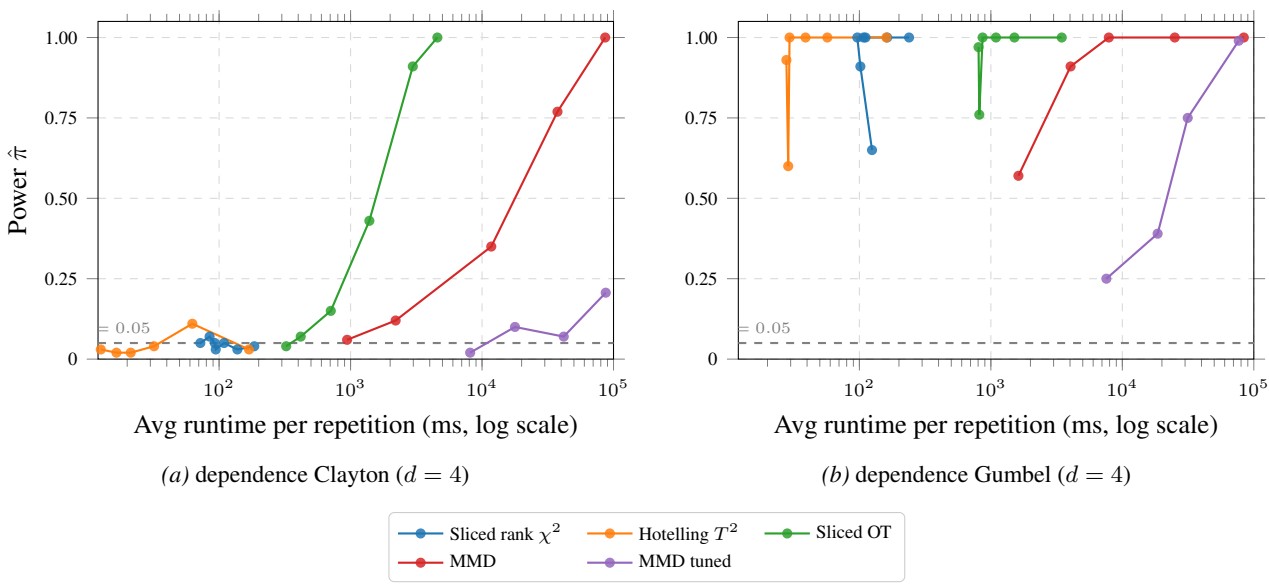

*Figure 13.* Efficiency–power tradeoff for dependence alternatives in dimension $d = 4$. We compare the sliced rank-statistic $\chi^2$-divergence test against Hotelling's $T^2$, sliced optimal transport, MMD, and tuned MMD. Curves show empirical power as a function of average runtime per repetition. All tests are calibrated at nominal level $\alpha = 0.05$.

$f_{\chi^2}(t) = \frac{1}{2}(t - 1)^2$, and average the resulting one-dimensional statistics over projections. The statistic is calibrated by the same permutation procedure described above, and empirical power is estimated from the rejection frequency over repeated trials.

We compare against sliced optimal transport (SOT), ridge-regularized Hotelling's $T^2$ (Hotelling, 1931), Gaussian-kernel maximum mean discrepancy (MMD) (Gretton et al., 2012), and a classifier two-sample test (C2ST) (Lopez-Paz & Oquab, 2017). This experiment tests whether the proposed sliced rank $f$-divergence detects realistic high-dimensional distribution shifts induced by image corruptions.

The results are summarized in Figure 14. The proposed rank-based statistic gains power as the matched sample size $N = M$ increases and achieves high empirical power with substantially lower runtime than the classifier-based baseline. The runtime comparison highlights the favorable power–runtime trade-off of the rank statistic in this pixel-space benchmark. The right panel reports the updated C2ST run at $N = M = 100$, showing that snow corruptions are detected more reliably at high severities, whereas JPEG compression and defocus blur are less clearly separated at this sample size.

## D. Limitations and future work

The proposed rank-statistic construction reduces divergence estimation to operations on ranks and histograms, yielding a fully sample-based surrogate that avoids explicit density-ratio modeling. At the same time, several limitations remain, many of which are shared by other projection-averaging objectives. In particular, the multivariate variant is defined by averaging a univariate discrepancy over random one-dimensional projections. While projection families can characterize distributions in the limit, any finite number of directions $L$ can miss informative orientations, especially when the discrepancy is concentrated in a low-dimensional subspace, encoded in rare but important directions, or dominated by higher-order dependence patterns. Similar phenomena are documented for sliced objectives in optimal transport and generative modeling (Kolouri et al., 2019). A practical implication is that performance can depend non-trivially on $L$ and on how directions are sampled, and diagnosing "missed directions" may be difficult without problem-specific insight.

A second limitation is that the surrogate is inherently discretized through the resolution parameter $K$. Although the theoretical analysis establishes monotonicity and consistency as $K \to \infty$, the choice of a finite parameter $K$ induces approximation error and may distort the geometry of the objective. This is particularly relevant when the divergence is used as a learning signal: the discretization can alter local sensitivity and may emphasize coarse distributional differences over fine structure. Developing principled, data-dependent rules for selecting $K$ (and, in the sliced case, jointly selecting $(K, L)$)

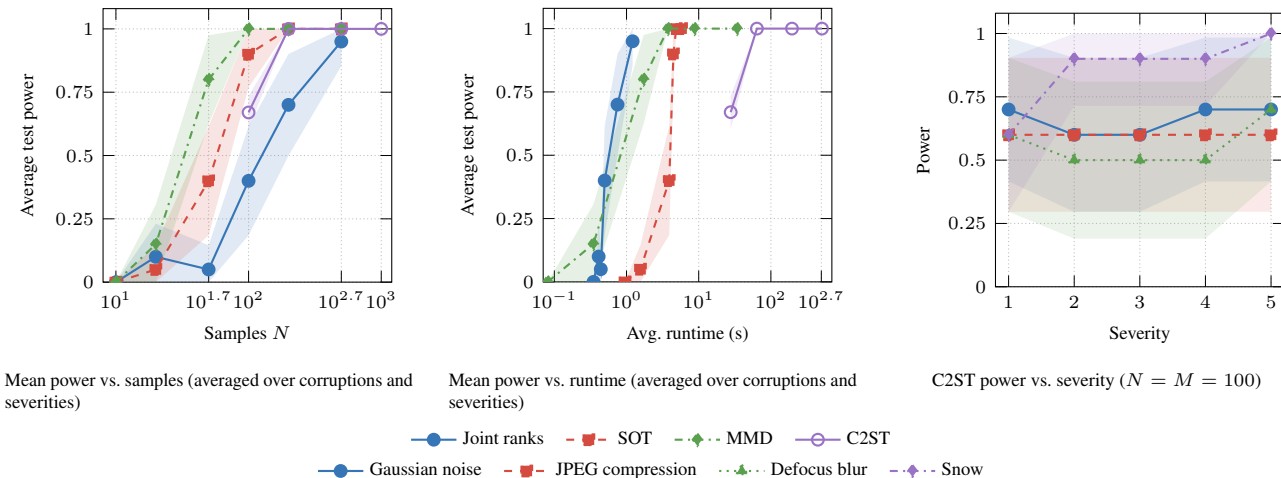

*Figure 14.* Empirical power for two-sample testing between CIFAR-10 and CIFAR-10-C at level $\alpha = 0.01$. Left: average power versus matched sample size $N = M$. Middle: average power versus average runtime. Right: C2ST power versus corruption severity for $N = M = 100$. Averages are computed over the considered corruption types and severities when applicable, and shaded bands show 95% binomial confidence intervals. The C2ST results are based on 10 independent trials for each corruption–severity pair.

remains an open problem. Promising directions include selection via held-out calibration, adaptive schedules that increase $K$ over training, and criteria based on stability of estimates across nearby resolutions.

Third, the sliced rank-statistic construction inherits an accuracy-compute trade-off from Monte Carlo integration on the sphere. In high dimensions, each projection reduces the problem to a 1D rank histogram, but capturing direction-dependent mismatch may require many directions $L$. Increasing $L$ improves coverage of informative orientations and typically reduces Monte Carlo variability through averaging, yet the overall cost grows roughly linearly in $L$ (and also increases with the rank resolution $K$ through the histogram/Bernstein evaluation). At realistic scales, this can make the sliced surrogate expensive, whereas using too few directions risks missing informative projections and yielding overly optimistic (or misleading) discrepancy estimates–a limitation shared by other sliced objectives (Kolouri et al., 2019).

Several extensions could increase the information per projection beyond i.i.d. random directions. One option is to replace purely random directions with structured ensembles that reduce redundancy (e.g., near-orthogonal directions) or with low-discrepancy point sets on the sphere, which can lower projection variance at fixed $L$ compared to standard Monte Carlo (Sobol, 1967; Dick & Pillichshammer, 2010). Another direction is data-dependent slicing: rather than sampling $s$ uniformly, directions can be biased toward projections with the largest (rank-based) discrepancy, connecting to max-sliced and projection-pursuit ideas (Deshpande et al., 2019; Paty & Cuturi, 2019). More generally, one could adapt learned slicing to the rank-statistic setting by choosing projection directions in a learned feature space, or by learning a small set of directions jointly with the generator so that each slice is maximally informative (Kolouri et al., 2019). Recent analyses of sliced distances and direction sampling also suggest that carefully designed projection sets can achieve greater statistical efficiency and stronger practical guarantees (Nietert et al., 2022).

Fourth, it would be valuable to place the sliced rank-statistic $f$-divergence in a more formal "flow" framework, in the same spirit as sliced-Wasserstein flows (Liutkus et al., 2019). Concretely, one can view the sliced rank objective as defining a functional on probability measures whose descent induces transport dynamics: at each time step, projected one-dimensional rank corrections define a drift field that moves particles toward the data distribution, while optional entropy/noise terms control dispersion and prevent collapse. A theory along these lines could clarify the accuracy-compute trade-off introduced by slicing (finite $L$ directions) and discretization (finite $K$), and could enable finite-time guarantees for particle discretizations that explicitly track how the error depends on $(m, K)$ and step sizes, mirroring the role of Monte Carlo slicing and time discretization in flow-based analyses (Liutkus et al., 2019).

In parallel, these transport dynamics suggest an amortized alternative: instead of running particle updates at test time, a generator could be trained to emulate one (or a few) steps of rank-proximal transport, or to directly map base noise to samples that minimize the sliced rank divergence. Such amortization could dramatically reduce the per-iteration cost at image scale.

Finally, empirical evaluation can be broadened along several axes. The current experiments emphasize synthetic settings and representative implicit learning tasks, but more diverse benchmarks (including larger-scale image generation, text/sequence data, and domain adaptation scenarios, and time-series forecasting) would better delineate when rank-statistic divergences outperform classical and neural alternatives. In addition, ablations that isolate the effects of $K$, $L$, projection sampling, and batching would help translate theoretical guarantees into robust practitioner guidance. Extensions to conditional divergences, two-sample testing, and settings with nuisance variables (e.g., covariate shift) are also natural, since rank constructions are compatible with sample-based pipelines and may be combined with representation learning.

## E. Computational Cost and Memory Footprint

We briefly discuss the computational cost of the sliced rank-statistic $f$-divergence estimators. For each projection direction, both the rank-statistic estimator and sliced Wasserstein first project the samples $X = \{x_i\}_{i=1}^N \subset \mathbb{R}^d$ and $Y = \{y_j\}_{j=1}^M \subset \mathbb{R}^d$ onto one dimension, which costs

$$\mathcal{O}((N + M)d).$$

Thus, for a fixed number of samples and slices, both methods have the same explicit linear dependence on the ambient dimension $d$.

For evaluation and two-sample testing, we use a non-differentiable hard-rank implementation. In each slice, we sort the projected reference samples and evaluate the empirical CDF of the reference distribution at the projected samples from $\mu$:

$$\widehat{F}_\nu(x_i) = \frac{1}{M}\#\{j : y_j \le x_i\}.$$

The resulting empirical CDF values are used to construct the degree-$K$ rank histogram and the corresponding discrete $f$-divergence. This gives the per-slice cost

$$\mathcal{O}\big((N + M)d + M \log M + N \log M + NK\big),$$

and therefore

$$\mathcal{O}\big(L((N + M)d + M \log M + N \log M + NK)\big)$$

over $L$ projections. The corresponding memory footprint is mild: apart from the input samples, it stores the projected samples, the rank/CDF values, and the rank histogram, giving approximately

$$\mathcal{O}\big((N + M)d + N + M + K\big).$$

For differentiable training, we use a soft-rank variant in which the empirical CDF is smoothed by logistic comparisons and the rank histogram is represented with a Bernstein basis of degree $K$. This yields a per-slice cost

$$\mathcal{O}\big((N + M)d + NM + NK\big),$$

where the $NM$ term comes from the pairwise soft-CDF computation. Its peak memory is approximately

$$\mathcal{O}\big((N + M)d + NM + NK\big),$$

since the implementation stores a pairwise comparison matrix of size $N \times M$ and a Bernstein basis matrix of size $N \times (K+1)$. When slices are processed sequentially, increasing $L$ mainly increases runtime rather than peak memory.

For comparison, a standard sliced Wasserstein estimator has per-slice cost

$$\mathcal{O}\big((N + M)d + N \log N + M \log M\big),$$

due to sorting the one-dimensional projected samples. Hence, both sliced Wasserstein and the hard-rank rank-statistic estimator have the same asymptotic dependence on the ambient dimension $d$, namely linear scaling through the projection step. The hard-rank estimator adds an extra one-dimensional term $NK$ for evaluating the rank-statistic $f$-divergence at resolution $K$. The soft-rank variant is more expensive because it pays the pairwise $NM$ cost, but this overhead is independent of the ambient dimension.

**Empirical time-to-quality comparison.** We also report an empirical time-to-quality comparison on a synthetic generative modeling benchmark in dimensions $d \in \{2, 4, 10\}$. For $d = 2$, the target distribution is a noisy two-moons distribution. For larger $d$, we keep the same two-dimensional nonlinear backbone and augment it with additional smooth nonlinear coordinates, followed by coordinate-wise standardization.

All methods train the same fully connected generator $G_\theta : \mathbb{R}^{16} \to \mathbb{R}^d$, with three hidden layers of width 128 and SiLU activations. We use Adam with learning rate $2 \cdot 10^{-3}$ and gradient clipping at 5. Sliced rank $f$-divergence (Rank), sliced Wasserstein (SWD), and maximum mean discrepancy (MMD) use mini-batches of size 256, while the full assignment optimal transport (OT) baseline uses mini-batches of size 96, due to the cubic cost of the Hungarian solver. The rank objective uses $L = 64$ random projections, Bernstein degree $K = 32$, the JS generator $f$, and soft-rank temperature scale 0.05. Sliced Wasserstein uses $L = 64$ projections and $p = 2$. MMD uses a multi-scale RBF kernel with bandwidths given by fixed multiples of the median heuristic. Evaluation uses $n_{\text{eval}} = 2048$ real and generated samples; rank-JS and SWD evaluations use 128 random projections.

As an external stopping criterion, we use a classifier-based estimate of the Jensen–Shannon divergence. At each evaluation checkpoint, a post-hoc binary classifier with two hidden layers of width 128 is trained for 300 steps to distinguish generated from real samples, using a $70/30$ train/validation split. This classifier is used only for evaluation and early stopping, not for training the generators.

Because a fixed classifier-JS threshold becomes more stringent as dimension increases, we use dimension-adapted targets:

$$\widehat{\text{JS}}_{\text{clf}} \leq 0.05 \quad (d = 2), \qquad \widehat{\text{JS}}_{\text{clf}} \leq 0.15 \quad (d = 4), \qquad \widehat{\text{JS}}_{\text{clf}} \leq 0.30 \quad (d = 10).$$

The resulting time-to-target comparison is summarized in Table 6 and visualized in Figures 15 and 16. Table 6 reports, for each method and dimension, whether the prescribed classifier-JS target is reached, together with the corresponding optimization step, wall-clock time, and final value of $\widehat{\text{JS}}_{\text{clf}}$. Figure 15 provides a direct visual comparison of the runtime required to reach the dimension-adapted target, while Figure 16 shows the final classifier-based JS estimate attained by each method.

| Dimension | Method | Reached target | Step | Time (s) | Final $\widehat{\text{JS}}_{\text{clf}}$ |
|---|---|---|---|---|---|
| $d = 2$ | Rank | yes | 600 | 91.3 | 0.0487 |
| | SWD | yes | 800 | 147.0 | 0.0444 |
| | MMD | no | – | – | 0.1415 |
| | OT | yes | 2500 | 266.9 | 0.0465 |
| $d = 4$ | Rank | yes | 1100 | 131.6 | 0.1410 |
| | SWD | yes | 1500 | 117.6 | 0.1101 |
| | MMD | no | – | – | 0.3942 |
| | OT | no | – | – | 0.2845 |
| $d = 10$ | Rank | yes | 600 | 59.5 | 0.2941 |
| | SWD | yes | 1700 | 120.5 | 0.3000 |
| | MMD | no | – | – | 0.4280 |
| | OT | no | – | – | 0.4856 |

*Table 6.* Time-to-target comparison using the external classifier-based JS criterion. A dash indicates that the method did not reach the target within the 3000-step budget.

Overall, these results show that the rank objective is competitive in wall-clock time under an external classifier-based stopping criterion. In $d = 2$, rank reaches the target faster than all baselines. In $d = 4$, both rank and sliced Wasserstein reach the adapted target; rank requires fewer optimization steps, while sliced Wasserstein is slightly faster in wall-clock time and attains a lower final classifier-JS value. In $d = 10$, rank again reaches the target substantially faster than sliced Wasserstein, whereas MMD and full assignment OT do not reach the target within the training budget. These results suggest that the finite-projection rank objective provides a competitive time-to-quality tradeoff, while also motivating the use of complementary evaluation metrics such as classifier-JS, sliced Wasserstein, MMD, and rank-JS.

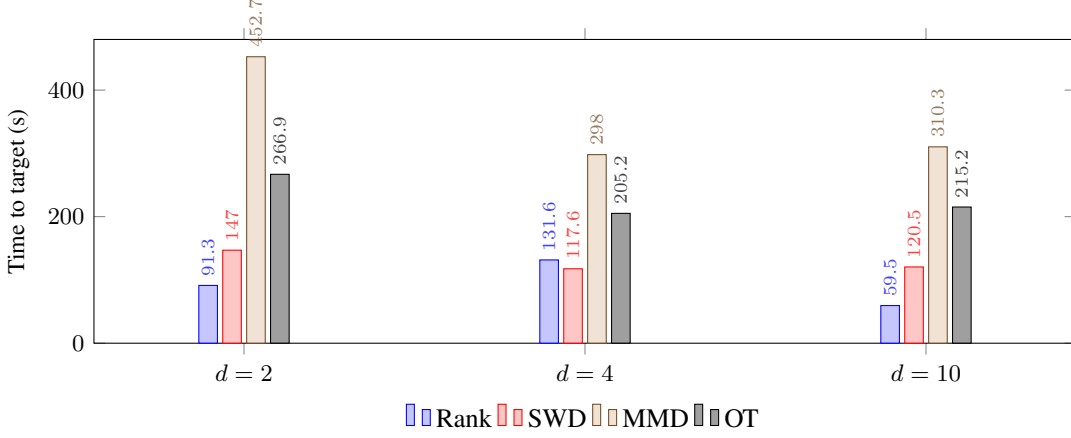

*Figure 15.* Time-to-target comparison under the external classifier-based JS stopping criterion. The targets are dimension-adapted: $\widehat{\mathrm{JS}}_{\mathrm{clf}} \leq 0.05$ for $d = 2$, $\leq 0.15$ for $d = 4$, and $\leq 0.30$ for $d = 10$. For methods that do not reach the target, the reported value corresponds to the total runtime under the 3000-step budget. In particular, MMD does not reach the target in any dimension, and OT does not reach it in $d = 4, 10$.

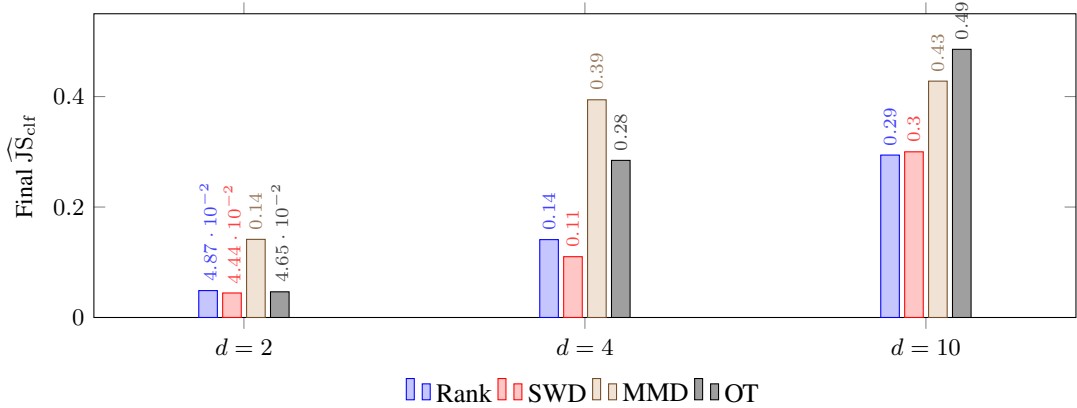

*Figure 16.* Final classifier-based JS estimate for each method and dimension. The dimension-adapted stopping thresholds are $\widehat{\mathrm{JS}}_{\mathrm{clf}} \leq 0.05$ for $d = 2$, $\leq 0.15$ for $d = 4$, and $\leq 0.30$ for $d = 10$. Lower is better.

## F. Experimental Setup

All experiments were performed on a MacBook Pro running macOS 13.2.1, equipped with an Apple M1 Pro CPU and 16 GB of RAM. When GPU acceleration was required, we used a single NVIDIA TITAN Xp with 12 GB of VRAM. Detailed hyperparameter settings for each experiment are provided in the corresponding sections. The code is available at https://github.com/josemanuel22/rsfdiv.

