# OpenReview forum: "Approximating f -Divergences with Rank Statistics"
_ICML.cc/2026/Conference — ICML 2026 regular_

### Official Review · Reviewer_T2EX · 2026-03-07

**Soundness:** 3
**Presentation:** 2
**Significance:** 3
**Originality:** 3
**Overall Recommendation:** 3
**Confidence:** 3

**Summary:**

This paper introduces a novel, optimization-free methodology for approximating $f$-divergences that circumvents the notoriously difficult task of explicit density-ratio estimation. The core approach leverages the probability integral transform to map the discrepancy between two univariate distributions into a discrete rank histogram with resolution $K$, subsequently measuring its deviation from uniformity via a discrete $f$-divergence. The authors establish robust theoretical foundations for this rank-statistic estimator, proving it acts as a monotonic lower bound to the true continuous $f$-divergence and deriving quantitative convergence rates as $K \rightarrow \infty$ under mild regularity conditions on the quantile-domain density ratio. To handle multivariate distributions, the work extends this univariate foundation into a "sliced" rank-statistic $f$-divergence by averaging across random one-dimensional projections, maintaining the core convergence properties in the sliced limit. Furthermore, the submission provides finite-sample deviation bounds, proves asymptotic normality for the estimator, and validates its practical utility through synthetic benchmarks against neural estimators and via generative transport dynamics on image datasets like CelebA.

**Compliance With Llm Reviewing Policy:**

Affirmed.

**Final Justification:**

the rebuttal addressed my concern, but I keep my original assessment

**Key Questions For Authors:**

see weaknesses

**Limitations:**

see weaknesses

**Strengths And Weaknesses:**

**Strengths**

* **Soundness:** The theoretical framework is rigorous. The authors prove the estimator is a convex, weakly lower semicontinuous, and monotonic lower bound to the true $f$-divergence. They provide strong quantitative convergence rates, finite-sample complexity bounds, and asymptotic normality guarantees.


* **Significance:** Estimating $f$-divergences in high dimensions without intermediate density estimation addresses a major bottleneck in the field. The optimization-free surrogate provides a highly stable learning objective for implicit generative models.


* **Originality:** The paper creatively synthesizes probability integral transforms, rank histograms, and sliced discrepancies. Extending rank-based bounds to arbitrary differentiable $f$-generators is a notable theoretical contribution.



**Weaknesses (Questions and Concerns for the Authors)**

1. **Presentation and Undefined Notation:** The narrative flow suffers from severe notational forward-referencing. Specifically, in the "Empirical estimation" paragraph between Example 2.1 and Theorem 2.3, the text abruptly introduces empirical measures ($\hat{\mu}_N$, $\hat{\nu}_M$) and sample variables ($X_i$, $Y_j$). These are not formally defined until Section 2.2. This makes the paper very hard to read.

2. **Hyperparameter Selection Guidance:** While the paper provides empirical sweeps demonstrating the effect of the resolution parameter $K$ (e.g., Table 1, Figures 7-9), it lacks actionable guidance. How should a practitioner dynamically or heuristically choose $K$, or the number of projections $L$, for a novel target distribution? The authors note this as an "open problem" in the appendix, but providing a concrete rule-of-thumb or adaptive schedule would vastly improve the method's practical utility.


3. **Computational Bottlenecks:** The paper notes an accuracy-compute trade-off where increasing $L$ and $K$ improves coverage but scales the cost. How does the wall-clock time and memory footprint of this rank-statistic approach compare to standard neural variational estimators in the high-dimensional generative experiments (e.g., CelebA)?

---

> ### Author Rebuttal · Authors · 2026-03-26
>
> We thank the reviewer for the careful reading and constructive feedback. We especially appreciate the positive assessment of the paper's originality.
>
> ---
>
> ### **Question 1**
>
> We agree that the original draft contained unnecessary notational forward references. In particular, the empirical-estimation paragraph introduced the empirical measures and sample variables before formally defining them. In the revision, we reordered this part of Section 2 so that these objects are introduced before their first use. Also, we now state in the notation paragraph that $X_1,\ldots,X_N \sim \mu$  denote the sample variables, and that $\hat\mu_N := \frac1N\sum_{i=1}^N \delta_{X_i}$ are the corresponding empirical measure estimates. We also clarify at first appearance which symbols denote population objects and which denote empirical quantities. In addition, we expanded the notation paragraph and will add a brief list of symbols in the appendix.
>
>
> ### **Question 2**
>
> We agree that the paper should provide more actionable guidance for choosing both $K$ and $L$, and in the revision, we will add concrete recommendations for practitioners.
>
> For $K$, we will recommend an **adaptive progressive-resolution schedule** rather than a large fixed value from the start. We begin with a small $K$, train until the induced rank statistic is sufficiently close to uniform, and then increase $K$ only once this coarse-resolution stage has stabilized, e.g., via a Pearson $\chi^2$ goodness-of-fit test against the discrete uniform law on the current rank histogram. We have also run this strategy in the present setting and observed a clear practical benefit: early stages are substantially cheaper, while the final performance remains very similar to that of using a large, fixed $K$ throughout. This makes progressive $K$ a more efficient default in practice.
>
> We will also add a finite-sample bias-variance ablation showing that the MSE is typically U-shaped in $K$: small $K$ yields a coarse approximation, while large $K$ introduces additional finite-sample noise. This supports moderate $K$ as a practical default, with larger values preferred when more samples are available.
>
> We are also willling to add an upper bound on $\left| D_{f, \nu}^{(K)}(\mu_N) - D_{f, \nu}(\mu) \right|$ in terms of $N$, $K$, and the smoothness of the quantile-space density ratio $r_{\mu \mid \nu}$, from which one can read off how the final value of $K$ should scale with $N$.
>
> For $L$, we will add guidance grounded in both the sliced-divergence literature and new ablations tailored to sliced rank-$f$ divergences. Prior work shows that finite-direction slicing introduces a genuine projection-complexity term, and that random slicing may require many projections in high dimension. Our new experiments show that the relevant issue is not dimension alone, but the geometry of the discrepancy: for global discrepancies (e.g. mean shifts or tail-heaviness), the number of slices needed to match a high-precision reference remains modest across dimensions, whereas for localized discrepancies it can grow essentially exponentially.
>
> The practical recommendation we will add is therefore simple: use moderate $L$ by default for global or diffuse mismatches; if the discrepancy is strongly anisotropic or localized, random slicing may become inefficient, and more informative projection schemes should be preferred.
>
> ### **Question 3**
>
> We agree that the practical compute cost should be made more explicit. In our high-dimensional generative experiments, the wall-clock overhead remains modest. On CelebA 64$\times$64, a direct runtime comparison under identical hardware and comparable hyperparameter settings shows an average training time of 56 min for the proposed rank-$f$ divergence method, versus 59 min for DCGAN, 59 min for rank-$f$ divergence + DCGAN, 61 min for W-DCGAN-GP, and 59 min for LS-GAN. On MNIST, the rank-$f$ divergence method is similarly comparable to DCGAN and clearly faster than W-DCGAN-GP and GMAN.
>
> This is consistent with the structure of the method: the rank-statistic surrogate is fully sample-based and avoids explicit density-ratio modelling or an additional neural variational estimator. The main compute trade-off instead comes from the rank resolution $K$ and the number of slices $L$. As we note in the paper, increasing $L$ improves coverage of informative directions but the cost grows roughly linearly in $L$, and it also increases with $K$.
>
> We have not yet included a systematic memory benchmark in the paper. In practice, the dominant memory cost comes mainly from the minibatch/particle tensors together with the per-slice projected/rank quantities, so it is driven primarily by batch size and by how many slices are processed simultaneously. In the revision, we will therefore add explicit wall-clock and memory comparisons in the main text.
>
> Overall, these results suggest that the method is competitive in practice; the main trade-off is the choice of $K$ and $L$.

---

> > ### Author Rebuttal · Reviewer_T2EX · 2026-04-03
> >
> > Thank you for the detailed and thoughtful response. I appreciate the clarifications and the planned revisions, which address my concerns.

---

> > > ### Author Response · Authors · 2026-04-03
> > >
> > > We thank the reviewer for the careful follow-up and for confirming that our response addressed the concerns raised. In the final version, we will incorporate the notation fixes, the practical guidance for choosing $K$ and $L$, and the explicit runtime/memory comparisons.

---

### Official Review · Reviewer_ey8t · 2026-03-12

**Soundness:** 3
**Presentation:** 4
**Significance:** 4
**Originality:** 4
**Overall Recommendation:** 6
**Confidence:** 3

**Summary:**

The paper introduces a rank-based approximation of f-divergences that avoids explicit density-ratio estimation by relying instead on rank statistics in the one-dimensional setting. The authors show that the resulting estimator is a lower bound on the target divergence, is monotone in the number of reference samples K, and converges to the true divergence as K increases. The paper then extends the construction to a sliced high-dimensional version based on random projections and evaluates it both as a divergence estimator and as a generation transport algorithm.

**Compliance With Llm Reviewing Policy:**

Affirmed.

**Final Justification:**

Question answered and additional experiments.

**Key Questions For Authors:**

1. The paper uses the precision-recall curves introduced by Sajjadi et al. [1]. However, these quantities can also be interpreted as \(f\)-divergences [2,3], and Simon et al. [4] propose an estimation method that also avoids explicit density-ratio estimation. Could the authors clarify how their approach relates to these prior works? A direct comparison between the two approximation strategies would be particularly interesting.
2. In higher-dimensional settings, one possible application would be to estimate \(f\)-divergences in the feature spaces of commonly used pretrained models such as Inception, DINOv2, or VGG. Could the authors comment on how their method would behave in such representation spaces, and on how sensitive it may be to the choice of embedding? This seems especially relevant given that FID relies on several strong assumptions, including Gaussianity in feature space.

[1] Assessing Generative Models via Precision and Recall, Sajjadi et al.

[2] Precision-Recall Divergence Optimization for Generative Modeling with GANs and Normalizing Flows, Verine et al.

[3] On the Theoretical Equivalence of Several Trade-Oﬀ Curves Assessing Statistical Proximity, Siry et al.

[4] Revisiting precision recall definition for generative modeling, Simon et al.

**Limitations:**

yes

**Strengths And Weaknesses:**

*Strengths*:
- **Presentation**: The paper is clearly written, the formalism is easy to follow, and the overall exposition is pleasant to read. In particular, the one-dimensional construction and its extension to the sliced setting are presented in a fairly transparent way.
- **Significance**: I find the problem relevant. Approximating \(f\)-divergences without explicit density-ratio estimation could be useful both conceptually and practically, especially if the method scales reliably to image-like data.
- **Soundness**: The theoretical claims appear substantial, and the paper provides both formal results and empirical illustrations supporting the proposed approximation. I did not verify the proofs in the appendix in detail, but the overall development appears coherent and the main claims seem plausibly supported.
- **Originality**: The main idea is appealing and, to me, clearly novel in flavor. Since density-ratio estimation is often the main obstacle in making \(f\)-divergences practical for generative modeling, replacing it with a rank-based approximation is an interesting contribution.

*Weaknesses*:
- **Originality / significance**: I am not fully convinced by the role of the transport-based generation procedure in the paper. While it shows that the proposed quantity can be used to generate samples, it is not clear to me that this yields a tractable and broadly useful learning objective for training generative models, unlike approaches such as f-GAN. Because of this, the framing in the abstract and experiments should be more precise: the method seems more naturally useful as an evaluation or sampling tool than as a general training objective.
- **Soundness / empirical validation**: The low-dimensional experiments are useful and probably necessary to validate the approximation, and the CO-RPT results suggest that the method captures something meaningful about distribution mismatch. However, they do not yet fully convince me that the estimator captures the specific behavior of different \(f\)-divergences in realistic settings. For instance, in Table 4, one might expect the model minimizing KL to achieve better recall than the one based on Jensen divergence. More generally, the paper would benefit from additional controlled experiments on pathological cases, such as duplicated samples, dropped modes, or the injection of out-of-distribution examples, in order to better distinguish the behavior of different divergences.
- **Significance**: If the estimator cannot be used as a practical training objective, then its value for evaluation should be demonstrated more strongly on modern high-dimensional generative models. At the moment, the paper makes a compelling methodological point, but the practical impact for real-world evaluation is not yet fully established.
- **Presentation**: Example 2.2 is not fully clear, especially for total variation. More generally, I think the paper would benefit from moving part of Section C.6 into the main text and explaining it more explicitly, since it seems important for understanding the practical relevance of the method.

---

> ### Author Rebuttal · Authors · 2026-03-25
>
> We thank the reviewer for the positive evaluation, in particular that the 1D theory is compelling and the sliced extension interesting and competitive.
>
> ---
>
> **Weaknesses**
>
> We agree that, in its current form, the method is strongest as a discrepancy / evaluation tool and as a pretraining objective, rather than as a fully general standalone training principle. We will revise the framing accordingly.
>
> We note that it appears quite effective as a **pretraining technique for GANs**. As shown in Appendix C.6, rank-based pretraining provides a stronger initialization for subsequent adversarial training and improves GAN recall. We agree that this point deserves more visibility, and we will therefore move the experiment into the main text and leave the full details to the appendix.
>
> We agree that the empirical validation can be strengthened. In the revision, we will **add a complementary two-sample benchmark**, covering location, scale/shape, multimodality, and dependence-only shifts, with comparisons against sliced Wasserstein, Hotelling’s $T^2$, kernel MMD, tuned MMD, and a classifier two-sample test, reporting both power and runtime. We will also add a small real-data benchmark on CIFAR-10 versus CIFAR-10-C, using clean CIFAR-10 as reference and corrupted CIFAR-10-C as target, and report test power and runtime against the same baselines. Preliminary results are encouraging: the rank-based procedure already achieves high power under mild shifts while remaining runtime-competitive.
>
>
> We agree that Example 2.2 was too compressed and could be hard to parse. Its purpose is simply to translate the abstract regularity assumptions of Theorem 2.5 into familiar examples. In particular, the relevant distinction is between (i) generators that are globally Lipschitz on $[0,\infty)$, such as TV, (ii) generators that are only Lipschitz away from zero, such as KL, Jensen--Shannon, squared Hellinger, and Jeffreys, and therefore satisfy the slower rate, and (iii) $C^2$ generators, such as $\chi^2$ and triangular discrimination, which satisfy the faster $O(K^{-1})$ rate. We will rewrite Example 2.2 to make this logic explicit.
>
> ---
>
> ### **Question 1**
>
> This is a very helpful point. Our method targets a different object: we do not estimate the PR curve itself, but a chosen scalar f-divergence via a rank-histogram/PIT construction that is optimization-free and comes with lower-bound, convergence, and finite-sample guarantees. There is nevertheless a direct bridge: if one chooses one of the generators $f_\lambda$ corresponding to a PR-divergence, then the same rank-based construction approximates that PR-divergence; varying $\lambda$ would in principle yield a discretized approximation of the PR trade-off curve. Thus Simon et al. target the PR curve directly, whereas we target scalar f-divergences, with the PR-divergence family embedded as a special case. We sincerely thank the reviewer for highlighting this connection. We agree that an empirical comparison would be very interesting and could open up a promising application of our method to the evaluation of generative models, as well as clarify its relationship with precision–recall curves. That said, we believe that carrying out such a comparison is beyond the scope of the present paper.
>
> ### **Question 2**
>
> This is a very relevant direction. The proposed estimator can be applied in the feature space of any fixed encoder, yielding a feature-space discrepancy that is naturally representation-dependent.
>
> We tested this in a **new CelebA 64×64 experiment**, where the generator is trained using a **sliced total variation rank divergence in discriminator feature space**. The results are encouraging: **FID decreases to 18.80**, and the resulting precision/recall reaches **0.792 / 0.824**. This supports the practical relevance of feature-space use, while also showing that the induced discrepancy depends on the chosen representation.
>
> A remaining caveat is slicing itself: with finitely many directions $L$, informative orientations may be missed. This is consistent with prior sliced generative-modeling work. Wu et al. note that conventional sliced Wasserstein approximations can require many random projections, motivating learned or orthogonal projection schemes. Liutkus et al. report noisy behavior in the original high-dimensional image space and, for CelebA, move to a bottleneck space; in their supplementary discussion they further relate finite-direction failures to a curse-of-dimensionality effect.
>
> ---
>
> ### Bibliography
>
>  - Wu, J. et al Sliced Wasserstein Generative Models. CVPR, 2019.
>
>  - Liutkus, A., et al. Sliced-Wasserstein Flows: Nonparametric Generative Modeling via Optimal Transport and Diffusions. ICML, 2019.

---

> > ### Author Rebuttal · Reviewer_ey8t · 2026-04-03
> >
> > Question resolved

---

> > > ### Author Response · Authors · 2026-04-03
> > >
> > > Thank you very much for your thoughtful review and for taking the time to reconsider the paper after reading our rebuttal. We are very glad that our responses addressed your concerns. We also sincerely appreciate your constructive suggestions, which will help us improve the final version of the paper.

---

### Official Review · Reviewer_kucm · 2026-03-23

**Soundness:** 3
**Presentation:** 2
**Significance:** 3
**Originality:** 3
**Overall Recommendation:** 5
**Confidence:** 3

**Summary:**

The paper presents a rank-statistics based method for estimating $f$-divergences.
The main idea is to build cumulative rank "histogram", which can be utilized to create an estimator of the 1D $f$-divergence.
In the 1D case, the paper presents various approximation properties and finite sample bounds.
For high dimensional spaces, the paper proposals to use a sliced variant of the estimator, which lower bounds the actual multidimensional $f$-divergence.
Various experiments are present to validate the usefulness of the estimator, one of which examines a generative particle algorithm.

**Compliance With Llm Reviewing Policy:**

Affirmed.

**Final Justification:**

My questions were answered and my concerns regarding clarity were adequately addressed. Hence, I have kept my positive score.

**Key Questions For Authors:**

1. Does one get a "symmetric" result to Theorem 3.3 if which measure is empirical is switched? Is it possible to get similar characterizations when both measures are empirical?
2. How does the error of the estimator grow when we consider an empirical sample of random directions? It seems that this would hinge on getting reasonable assumption on $s \mapsto D_{f, \nu_s}^{(K)}(\mu_s)$.
3. In Section 4.4, is there a reason for the subscript $0$ and $1$ on $U$? I think there is a notational clash here given that you define $U = (U_1, \ldots, U_N)$?
4. From what I understand, Section 4.4 is essentially trying to move the particles by comparing their quantile values via deriving a step for each random projection (which is averaged and scaled) --- which seems to match the discretized update of (9). Is the $\hat{F}$ meant to be $\hat{R}$, the empirical CDF? It may be worth to explicitly state how this is constructed via samples (in particular its inverse, as used in $z^{(\ell)}$ update.
5. What is $\textrm{ran}$? Is this the range of a function?

**Limitations:**

yes

**Strengths And Weaknesses:**

I found that the rank statistic estimator was an interesting approach to $f$-divergence estimation.
In the 1D case, the theory seems quite compelling.
The sliced extension for multiple dimensions seem reasonable and the experiments seem to indicate that it is competitive against neural approaches and can be used in non-trivial tasks.
Results-wise, I think the paper presents an interesting approach for $f$-divergence estimation.

A weakness of the paper is in its presentation. There are various notational/definition issues, where quantities are stated and used without prior definition. I spotted $b_{n, K}$, $\textrm{ran}$, and $\hat{F}_{\nu^{(\ell)}}$ in the main text; and the $H_r$ in $H_r$-Hölder continuity used in the appendix.
The proof of Theorem 2.5 also has some typographical errors (particularly Lemma B.3).

Notation:
  - $b_{n, K}$, $\textrm{ran}$, and $\hat{F}_{\nu^{(\ell)}}$ are not defined in the main text; and $H_r$-Hölder continuity is not defined in the appendix.
  - $c_K$ appearing in the proof sketch of Theorem 2.5 is not needed (makes it more confusing)
  - Notation "shadowing" when comparing the sliced versus non-sliced versions of the estimator. I assume this is because the estimator only works in 1D so any high dimension input is implicitly a sliced variant, but it is slight confusing at first read. It might be worth explicitly pointing out this notation overloading.
  - Casing on "pmf" vs "CDF".

Typos:
  - Extra "," in equation on line 772
  - "second summand" should be "first summand" on line 781
  - Missing "r" on line 784
  - `frac` throughout Lemma B.3.
  - There is a general inconsistency with the hyphen symbol used between compound words.

---

> ### Author Rebuttal · Authors · 2026-03-25
>
> We thank the reviewer for the thoughtful review and positive assessment of the paper’s theoretical contribution. We are glad that you found the work significant.
>
> ---
> **Notation/presentation**
>
> We agree that parts of the notation should be made more consistent and self-contained, and we revised the manuscript accordingly.
> * We now define Hölder continuity and $\textrm{ran}(h)$ explicitly in the notation section, so that $H_r$ in Appendix B and $ran(r)$ in Theorem 2.5 are introduced before use. We also define the Bernstein basis $b_{n, K}$ at first appearance in the main text and refer to Appendix A for further details.
> * In the proof sketch of Theorem 2.5 and in the proof of Lemma B.3, we removed the unnecessary auxiliary notation $c_K$ and fixed the typographical errors.
> * We added a short clarification in the sliced section explaining that the one-dimensional quantities are reused slice-wise for projected marginals. In other words, the symbols $R_\nu$, $Q_\nu$, etc., are applied to the projected one-dimensional pair. We now state this explicitly to avoid confusion caused by notation overloading between the 1D and sliced settings.
>
> Finally, in Section 4.4, we aligned the empirical CDF/quantile notation with the rest of the paper: we now write the slice-wise empirical CDF as $\widehat R_{\nu^{(\ell)}}$, use its generalized inverse consistently, and state explicitly that it is constructed from the projected reference samples.
>
> We also standardized PMF/CDF capitalization and corrected the minor wording and hyphenation issues flagged in the review.
>
> ---
>
> **Questions**
>
> 1) We expect analogous results to hold both when the empirical measure is switched and when both measures are empirical, but these are **not** immediate corollaries of Theorem 3.3 in the general asymmetric case. Theorem 3.3 relies on a representation available when $\hat\mu_N$ is empirical and $\nu$ is fixed, where the sliced histogram coordinates are empirical averages of i.i.d. terms; if the roles are reversed, the randomness enters through the empirical CDF of $\hat\nu_M$, so one would need empirical-process / functional delta-method arguments instead. For symmetric divergences (e.g., Jensen--Shannon), the switched one-sample *statement* is essentially immediate by symmetry, but this does not cover the general asymmetric $f$-divergence case or the genuinely two-sample empirical setting. The needed assumptions would be of the same type as in Theorem 3.3, plus regularity of the sliced reference laws and, in the two-sample case, joint asymptotics $N,M\to\infty$ with comparable sample sizes. This is a natural extension, but beyond the scope of the present paper. If the reviewer feels that this extension is necessary, we think it would be feasible to develop and include this much more involved proof.
>
> 2) Yes, the additional error from replacing the spherical average by an empirical average over $L$ random directions should be the standard Monte Carlo error. In particular, Nadjahi et al. (2020, Thm. 6) show for sliced probability divergences that using $L$ i.i.d. directions yields an $L^{-1/2}$ rate, with the constant given by the standard deviation of the directional integrand. For fixed $K$, we thus only require $f$ to be bounded on $[0, K + 1]$ to obtain the same $O(L^{-1/2})$ projection error here. We will add a short discussion making this explicit. The stronger assumption you mention may give even stronger qualitative rates.
>
> 3) We agree that the notation in Section 4.4 was confusing. The subscripts $0$ and $1$ on $U^{(\ell)}$ were only meant to denote the rank vector before and after the proximal refinement on slice $\ell$, but this clashed with the use of $U$ for the optimization variable and with $U_K$ for the discrete uniform law on $[K]$. We therefore revised this part so that the current and updated slice-wise ranks are denoted by $u_0^{(\ell)}$ and $u_1^{(\ell)}$, while the optimization variable is written separately (e.g. $v\in[0,1]^N$).
>
> 4) We agree that this step should be stated more explicitly. In Section 4.4, the relevant object is the **slice-wise empirical CDF** of the projected reference samples, and the update follows the standard one-dimensional quantile-matching construction used in sliced OT: after projection onto $s_\ell$, one computes a rank/CDF coordinate, updates it in $[0,1]$, and maps it back through the corresponding empirical quantile function. The original notation obscured this, since earlier we use $R_\mu$ and $Q_\mu$ for the CDF and quantile function, while Section 4.4 / Appendix C.4 used an $\widehat F$-notation.
>
> 5) We agree that this notation should have been defined explicitly. Here, $\textrm{ran}(r)$ denotes the range (image) of $r$. We have now defined $\textrm{ran}(h)$ explicitly in the notation section.
> ---
> **Bibliography**
>
> Nadjahi, K., et al Statistical and Topological Properties of Sliced Probability Divergences. NeurIPS, 2020.

---

> > ### Author Rebuttal · Reviewer_kucm · 2026-04-02
> >
> > Questions resolved.

---

> > > ### Author Response · Authors · 2026-04-02
> > >
> > > Thank you for taking the time to read our rebuttal and for the positive acknowledgement. We are glad that our clarifications addressed your concerns.

---

### Decision · Program_Chairs · 2026-04-30

**Decision:**

Accept (regular)

**Comment:**

A very interesting paper, which addresses an important problem, congratulation on its acceptance. In the final version, it is crucial that the authors take into account the comments of reviewers *and* their commitments to carry out modifications / improvements on related matters:

- include comments to all 3 questions by T2EX, in particular question 2 (guidance) which is crucial for the adoption of such a technique
- make modifications related to reply to weaknesses of ey8t, in particular regarding the empirical validation
- address notational problems, in particular those highlighted by kucm. This is crucial to improve the readability of the paper.

It is important to carry out all these modifications and improvements, as they will surely improve the quality of the paper.